# Learning Large-Scale Competitive Team Behaviors with Mean-Field Interactions and Online Opponent Modeling

## Abstract

While multi-agent reinforcement learning (MARL) has been proven effective across both collaborative and competitive tasks, existing algorithms often struggle to scale to large populations of agents. Recent advancements in mean-field (MF) theory provide scalable solutions by approximating population interactions as a continuum, yet most existing frameworks focus exclusively on either fully cooperative or purely competitive settings. To bridge this gap, we introduce MF-MAPPO, a mean-field extension of PPO designed for zero-sum team games that integrate intra-team cooperation with inter-team competition. MF-MAPPO employs a shared actor and a minimally informed critic per team and is trained directly on finite-population simulators, thereby enabling deployment to realistic scenarios with thousands of agents. We further show that MF-MAPPO naturally extends to partially observable settings through a simple gradient-regularized training scheme. Our evaluation utilizes large-scale benchmark scenarios using our own testing simulation platform for MF team games (`MFEnv`), including offense–defense battlefield tasks as well as variants of population-based rock-paper-scissors games that admit analytical solutions, for benchmarking. Across these benchmarks, MF-MAPPO outperforms existing methods and exhibits complex, heterogeneous behaviors, demonstrating the effectiveness of combining mean-field theory and MARL techniques at scale.

## 1 Introduction

Existing state-of-the-art MARL algorithms built upon MADDPG, MAAC and MA-PPO (Lowe et al., 2017; Yu et al., 2022), face severe scalability challenges as the number of agents grows, primarily due to the well-known *curse of dimensionality*. A promising remedy is offered by mean-field theory, which approximates large-scale agent–environment interactions in the infinite-population limit (Huang et al., 2006). Two major areas of mean-field research are mean-field games (MFGs) (Huang et al., 2006; Lasry & Lions, 2007; Sen & Caines, 2019; Laurière et al., 2022), which focus on non-cooperative agents, and mean-field control (MFC) problems (Bensoussan et al., 2013; Gu et al., 2021), which study fully cooperative scenarios. In contrast, mixed collaborative–competitive scenarios that arise in many real-world domains, such as team sports (Gaviria Alzate et al., 2025) and social dilemmas (Leibo et al., 2017), remain relatively under-explored. To address this gap, we propose Mean-Field Multi-Agent Proximal Policy Optimization (MF-MAPPO), the first PPO-based *learning* algorithm tailored for mixed cooperative–competitive mean-field settings. Guided by existing theoretical results (Guan et al., 2024a), MF-MAPPO scales to hundreds or thousands of agents while preserving convergence guarantees and remaining agnostic to individual identities or private observations.

**Mean-Field Teams.** The single-team problem was explored in Arabneydi & Mahajan (2014), where agents share a common team reward (MFC). In contrast, Mahajan & Nayyar (2015) established optimality for finite-population games *only* in the LQG setting, while Sanjari et al. (2022) analyzed a two-team setting with continuous states and actions—unlike our finite state-space formulation that directly admits the familiar MDP-type structure. Similarly, *multi-population* MFGs (MP-MFGs) have been studied in the past but often restrict agent dynamics and policies to be independent of both other agents and other population distributions, see Perolat et al. (2021) and references therein. More

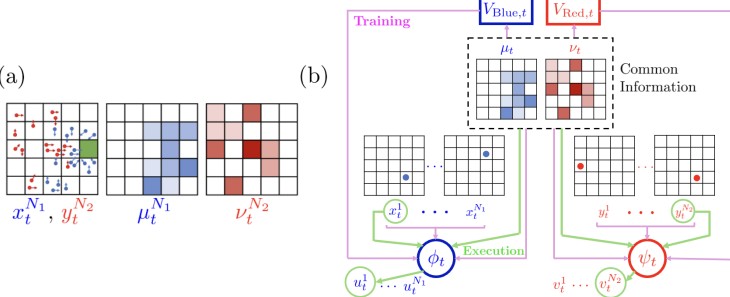

Figure 1: (a) Battlefield as a ZS-MFTG (b) Overview of the architecture of MF-MAPPO.

recently, Guan et al. (2024a) introduced *zero-sum mean-field team games* (ZS-MFTGs), modeling large-population teams that compete while cooperating internally. A common-information decomposition (Nayyar et al., 2013) enables team-size-independent learning with identical optimal team policies using MF feedback, unlike aforementioned MP-MFGs (open-loop MF policies). However, computing such policies numerically, especially in large state-action spaces using dynamic programming, is costly. By contrast, MF-MAPPO leverages shared actor–critic networks per team and uses only commonly accessible information (exact/estimated), ensuring tractability and scalability.

**Mean-Field Theory and Learning.** Recent advances in mean-field learning span Q-function–based methods, such as MF-Q and MF-AC (Yang et al., 2020) and DDPG-MFTG (Shao et al., 2024b), to value-function–based methods like Dec-POMFPPO (Cui et al., 2024) (MFC only). While DDPG-MFTG incorporates team games, it is restricted to simple grid worlds, unlike our focus on tightly coupled collaborative–competitive domains (ZS-MFTGs). We adopt it as a baseline and show that MF-MAPPO consistently outperforms it in stability and performance. Other related works include PMD-TD for MFGs (Yardim & He, 2024) and GAN-based ECA-Net for continuous-space attack–defense games (Wang et al., 2022), both differing in scope and structure. Moreover, Yang et al. (2020) define mean fields over neighboring actions rather than the full state space. MF-MAPPO extends PPO (Schulman et al., 2017) to competitive MFTGs, using team distributions as critic inputs for scalability, a shared actor–critic per team with a single buffer for efficiency, and simultaneous team training to avoid the inefficiencies of iterative best-response methods (Lanctot et al., 2017; Smith et al., 2021). Unlike prior MF methods that rely on infinite-population oracles (Shao et al., 2024a; Perolat et al., 2021; Carmona et al., 2021), MF-MAPPO is trained directly in finite-population simulators, making it suitable for realistic deployment. Finally, to standardize evaluation in large-scale MFTGs, we create novel MFTG benchmark environments (Constrained Rock–Paper–Scissors, Battlefield) going beyond existing ones that either focus only on MFGs (Guo et al., 2023) or omit MF coupling altogether (Terry et al., 2021; Zheng et al., 2018).

**Mean-Field Estimation and Opponent Modeling.** To enable reactive behavior to opponents' actions, most existing MF approaches assume centralized or exact knowledge of the opponent's MF, which is rarely practical. While Cui et al. (2024) considers partial observability, it is limited to MFC. Estimation methods such as kernel density estimation (Inoue et al., 2021) or normalizing flows (Perrin et al., 2021) struggle with discrete spaces and limited agent-visibility. Communication-based methods (Benjamin & Abate, 2025b;a) address these constraints but assume uniform estimates for unobserved states and rely on perfect multi-round communication while being restricted to fully cooperative or competitive regimes; noisy variants may even produce invalid distributions. We instead propose Dynamic-Projected Consensus (D-PC), a constrained consensus algorithm that ensures validity, exponential convergence, and bounded deviations when paired with a *gradient-regularized* MF-MAPPO policy. Gradient regularization naturally stabilizes MF-MAPPO in partially observable MFTGs. Experiments show that D-PC matches baseline performance and even outperforms them under limited communication, especially critical in adversarial settings requiring rapid adaptation (Richards et al., 2012), enhancing robustness and fault tolerance. To our knowledge, this is the first use of MF estimation for opponent modeling in competitive team settings.

**Our contributions.** The main contributions of our work can be summarized as: 1) MF-MAPPO, a scalable shared-actor–critic algorithm for large-scale MFTGs; 2) novel MFTG benchmarking environments (`MFEnv`) for validating MARL scalability; 3) A gradient-regularized extension of MF-MAPPO coupled with a decentralized mean-field estimation framework D-PC, with theoretical

performance guarantees in partially observable MFTGs; 4) comprehensive numerical experiments demonstrating MF-MAPPO and D-PC's superior performance and efficiency over existing baselines.

## 2 PROBLEM FORMULATION

### 2.1 ZERO-SUM MEAN-FIELD TEAM GAME

The zero-sum mean-field team game models a discrete-time stochastic game between two large teams of agents (Guan et al., 2024a). The Blue and Red teams consist of $N_1$ and $N_2$ *identical* agents for each team, with the total number of agents being $N = N_1 + N_2$. Let $X_{i,t}^{N_1} \in \mathcal{X}$ and $U_{i,t}^{N_1} \in \mathcal{U}$ represent the state and action of Blue agent $i \in [N_1]$ at time $t$. Here, $\mathcal{X}$ and $\mathcal{U}$ are the finite state and action spaces of the Blue team. Similarly, $Y_{j,t}^{N_2} \in \mathcal{Y}$ and $V_{j,t}^{N_2} \in \mathcal{V}$ denote the state and action of Red agent $j \in [N_2]$ at time $t$. The joint state-action variables for the Blue and Red teams are denoted as $(\mathbf{X}_t^{N_1}, \mathbf{U}_t^{N_1})$ and $(\mathbf{Y}_t^{N_2}, \mathbf{V}_t^{N_2})$, respectively. We denote the space of probability measures over a set $E$ as $\mathcal{P}(E)$. Below, $\mathrm{d}_{\mathrm{TV}}(\mu, \mu')$ represents the total variation between $\mu, \mu' \in \mathcal{P}(E)$.

**Definition 1.** *The empirical distributions (ED) for the Blue and Red teams are defined as*

$$\mathcal{M}_t^{N_1}(x) = \frac{1}{N_1} \sum_{i=1}^{N_1} \mathbb{1}_x(X_{i,t}^{N_1}), \ x \in \mathcal{X}, \ \text{and} \ \mathcal{N}_t^{N_2}(y) = \frac{1}{N_2} \sum_{j=1}^{N_2} \mathbb{1}_y(Y_{j,t}^{N_2}), \ y \in \mathcal{Y}, \quad (1)$$

where $\mathbb{1}_a(b) = 1$ if $a = b$ and $0$ otherwise. Specifically, $\mathcal{M}_t^{N_1}(x)$ gives the fraction of Blue agents at state $x$ and, similarly, for $\mathcal{N}_t^{N_2}(y)$. We use $\mathcal{M}_t^{N_1} = \mathrm{Emp}_\mu(\mathbf{X}_t^{N_1})$ and $\mathcal{N}_t^{N_2} = \mathrm{Emp}_\nu(\mathbf{Y}_t^{N_2})$ to denote the EDs computed from the given joint states. Note that the $\mathrm{Emp}$ operators remove agent index information, so one *cannot* determine the state of a specific Blue agent $i$ from $\mathcal{M}_t^{N_1}$.

We consider weakly-coupled dynamics where the dynamics of each individual agent is coupled with other agents through the EDs (Huang et al., 2006; Sanjari et al., 2022). For Blue agent $i$, its stochastic transition is governed by the transition kernel $f_t : \mathcal{X} \times \mathcal{U} \times \mathcal{P}(\mathcal{X}) \times \mathcal{P}(\mathcal{Y}) \to \mathcal{P}(\mathcal{X})$:

$$\mathbb{P}(X_{i,t+1}^{N_1} = x_{i,t+1}^{N_1} | U_{i,t}^{N_1} = u_{i,t}^{N_1}, \mathbf{X}_t^{N_1} = \mathbf{x}_t^{N_1}, \mathbf{Y}_t^{N_2} = \mathbf{y}_t^{N_2}) = f_t(x_{i,t+1}^{N_1} | x_{i,t}^{N_1}, u_{i,t}^{N_1}, \mu_t^{N_1}, \nu_t^{N_2}), \quad (2)$$

where $\mu_t^{N_1} = \mathrm{Emp}_\mu(\mathbf{x}_t^{N_1})$ and $\nu_t^{N_2} = \mathrm{Emp}_\nu(\mathbf{y}_t^{N_2})$. Similarly, the dynamics of Red agent $j$ is governed by the transition kernel $g_t : \mathcal{Y} \times \mathcal{V} \times \mathcal{P}(\mathcal{X}) \times \mathcal{P}(\mathcal{Y}) \to \mathcal{P}(\mathcal{Y})$. All agents in the Blue team receive an identical weakly-coupled team reward, i.e., $r_t \triangleq r_t(\mu_t, \nu_t) : \mathcal{P}(\mathcal{X}) \times \mathcal{P}(\mathcal{Y}) \to \mathbb{R}$. The Red agents receive $-r_t(\mu_t, \nu_t)$ as their rewards (zero-sum). We assume that the Blue team is *maximizing* while the Red team is *minimizing* and $r_t \in [-R_{\max}, R_{\max}]$ for all $t$.

**Assumption 1** (Lipschitz Model). *For all $x \in \mathcal{X}, u \in \mathcal{U}$, $\mu, \mu' \in \mathcal{P}(\mathcal{X})$, $\nu, \nu' \in \mathcal{P}(\mathcal{Y})$ and all $t$, there exist constants $L_f, L_r > 0$ such that $\sum_{x' \in \mathcal{X}} |f_t(x'|x, u, \mu, \nu) - f_t(x'|x, u, \mu', \nu')| \leq L_f(\mathrm{d}_{\mathrm{TV}}(\mu, \mu') + \mathrm{d}_{\mathrm{TV}}(\nu, \nu'))$ and $|r_t(\mu, \nu) - r_t(\mu', \nu')| \leq L_r(\mathrm{d}_{\mathrm{TV}}(\mu, \mu') + \mathrm{d}_{\mathrm{TV}}(\nu, \nu'))$. A similar assumption also holds for $g_t$.*

Lipschitz continuity is commonly assumed (Huang et al., 2006; Gu et al., 2021), and at minimum uniform continuity is required; see Cui et al. (2024) for counterexamples.

The first grid in Figure 1(a) depicts the individual agents' local positions, with the target marked by the green cell. The subsequent grids illustrate the state distributions $\mu_t^{N_1}$ and $\nu_t^{N_2}$ of both teams. The agent interactions depend only on $\mu_t^{N_1}$ and $\nu_t^{N_2}$ (weakly-coupled) as described in (2).

We consider a mean-field sharing information structure (Arabneydi & Mahajan, 2015), where each agent's decision depends on its own state and the two team EDs. We start with assuming full observation of mean-fields and later relax this assumption. Specifically, the Blue and Red agents seek to construct mixed Markov policies $\phi_{i,t} : \mathcal{U} \times \mathcal{X} \times \mathcal{P}(\mathcal{X}) \times \mathcal{P}(\mathcal{Y}) \to [0, 1]$, and $\psi_{j,t} : \mathcal{V} \times \mathcal{Y} \times \mathcal{P}(\mathcal{X}) \times \mathcal{P}(\mathcal{Y}) \to [0, 1]$, where the Blue policy $\phi_{i,t}(u|x_{i,t}^{N_1}, \mu_t^{N_1}, \nu_t^{N_2})$ dictates the probability that Blue agent $i$ selects action $u$ given its state $x_{i,t}^{N_1}$ and the observed/estimated team EDs $\mu_t^{N_1}$ and $\nu_t^{N_2}$. Note that each agent's individual state is its private information.

Let $\Phi_t$ ($\Psi_t$) denote the set of individual Blue (Red) policies at time $t$. We define the Blue team policy $\phi_t^{N_1} = \{\phi_{i,t}\}_{i=1}^{N_1}$ as the collection of the $N_1$ Blue agent individual policies, and denote the set of Blue team policies as $\Phi_t^{N_1} = \times_{N_1} \Phi_t$. Similarly, the Red team policy is denoted as $\psi_t^{N_2} \in \Psi_t^{N_2} = \times_{N_2} \Psi_t$.

**Definition 2** (Identical team policy). *The Blue team policy $\phi_t^{N_1} = (\phi_{1,t}^{N_1}, \ldots, \phi_{N_1,t}^{N_1})$ is identical, if $\phi_{i_1,t} = \phi_{i_2,t}$ for all times $t$ and all $i_1, i_2 \in [N_1]$. $\Phi$ represents the set of identical Blue team policies.*

The definition extends naturally to the Red team, and $\Psi$ denotes the set of identical Red team policies. The expected cumulative reward defines the performance of the team policy pair $(\phi^{N_1}, \psi^{N_2})$:

$$J^{N,\phi^{N_1},\psi^{N_2}}\left(\mathbf{x}_0^{N_1}, \mathbf{y}_0^{N_2}\right) = \mathbb{E}_{\phi^{N_1},\psi^{N_2}}\Big[\sum_{t=0}^{T} r_t(\mathcal{M}_t^{N_1}, \mathcal{N}_t^{N_2})\Big|\mathbf{x}_0^{N_1}, \mathbf{y}_0^{N_2}\Big]. \tag{3}$$

When the Blue team considers its worst-case performance, we have the following max-min optimization problem:

$$\underline{J}^{N*}(\mathbf{x}_0^{N_1}, \mathbf{y}_0^{N_2}) = \max_{\phi^{N_1} \in \Phi^{N_1}} \min_{\psi^{N_2} \in \Psi^{N_2}} J^{N,\phi^{N_1},\psi^{N_2}}(\mathbf{x}_0^{N_1}, \mathbf{y}_0^{N_2}), \tag{4}$$

where $\underline{J}^{N*}$ is the lower game value for the *finite-population* game. Similarly, the minimizing Red team considers a min-max optimization problem, which leads to the upper game value. Note that we allow both teams to follow *non-identical* team policies in (4).

## 2.2 INFINITE-POPULATION SOLUTION

To reduce the complexity of team policy optimization domains in (4), the authors of Guan et al. (2024a) examined team behaviors under *identical team policies* at the *infinite-population* limit. It was shown that the team joint states can be represented using the team population distribution, which coincides with the state distribution of a *typical agent* referred to as the mean-fields ($\mu_t$ and $\nu_t$ for the Blue and Red teams, respectively). They also proved that MFs induced by identical team policies in an infinite-population game closely approximate the EDs induced by *non-identical* team policies in the corresponding finite-population game, which justifies the simplification of the optimization domain in (4) to identical team policies (also see Theorem 1). Furthermore, there is a one-to-one correspondence between infinite-population coordination policies $(\alpha, \beta)$ and local *identical* team policies $(\phi, \psi) \in \Phi \times \Psi$. The performance of $(\phi, \psi)$ in the equivalent zero-sum coordinator game is measured by

$$J_\infty^{\alpha,\beta}(\mu_0, \nu_0) \equiv J_\infty^{\phi,\psi}(\mu_0, \nu_0) = \sum_{t=0}^{T} r_t(\mu_t, \nu_t), \tag{5}$$

where $\mu_t$ and $\nu_t$ follow a *deterministic* dynamics (Guan et al., 2024a) similar to the state distribution propagation of a controlled Markov chain. The worst-case performance of the Blue team in this infinite-population game is then given by the lower game value $\underline{J}_\infty^*(\mu_0, \nu_0) = \max_{\phi \in \Phi} \min_{\psi \in \Psi} J_\infty^{\phi,\psi}(\mu_0, \nu_0)$, where the optimization domain is restricted to *identical team policies*. Guan et al. (2024a) establishes guarantees that identical team policies resulting from the solution of this equivalent zero-sum *coordinator* game are still $\epsilon$-optimal for the original max-min optimization problem in (4) where $\epsilon = \mathcal{O}(1/\sqrt{\underline{N}})$ and $\underline{N} = \min\{N_1, N_2\}$.

The infinite-population limit of large-population games offers several theoretical advantages, such as representing the population by a typical agent and deterministic dynamics that reduce (3) to the non-stochastic optimization of (5). Previous works (Shao et al., 2024a; Perolat et al., 2021; Carmona et al., 2021) depend on infinite-population oracles to obtain mean-field trajectories $(\mu_t, \nu_t)$ in order to compute (5). This is rather unrealistic, since in practice, only finite-population simulations and local states $(\mathbf{x}_t^{N_1}, \mathbf{y}_t^{N_2})$ with actions $(\mathbf{u}_t^{N_1}, \mathbf{v}_t^{N_2})$ are available/observable. Moreover, a single coordinator policy $\alpha(\beta)$ defines a distribution over actions for each state conditioned on the mean-field, causing its dimensionality to scale with the joint state–action space (e.g., DDPG-MFTG), leading to high computational cost and degraded empirical performance (see Section 5). In summary, the infinite-population model is both impractical (due to oracle dependency) and computationally intractable (due to policy size). Thus, we turn to finite-population simulators and derive guarantees of optimality, scalability, and convergence of the policy gradient to the infinite-population ZS-MFTG.

The next result quantifies the level of suboptimality for the Blue team when it deploys the optimal identical policy learned directly from the solution of *finite*-population ZS-MFTG.

**Theorem 1.** *The value of the optimal identical Blue team policy $\phi^*$ obtained from the* finite popula­tion game *is within $\epsilon$ of the finite-population lower game value defined in* (4). *Formally, for all joint states $\mathbf{x}^{N_1}$ and $\mathbf{y}^{N_2}$,*

$$\min_{\psi^{N_2}} J^{N,\phi^*,\psi^{N_2}}(\mathbf{x}^{N_1}, \mathbf{y}^{N_2}) \geq \underline{J}^{N*}(\mathbf{x}^{N_1}, \mathbf{y}^{N_2}) - \mathcal{O}\Big(\frac{1}{\sqrt{\underline{N}}}\Big), \text{ where } \underline{N} = \min\{N_1, N_2\}. \quad (6)$$

Theorem 1 provides a principled justification for learning identical finite-population team policies in competitive–collaborative team games even when being exploited by non-identical opponent team strategies. Its motivation and proof build on the performance guarantees of the ZS-MFTG in the infinite-population limit, i.e., the coordinator game. Moreover, the error vanishes as $N_1, N_2 \to \infty$, thereby recovering the well-studied infinite-population MF formulation (Huang et al., 2006). We detail this finite-population training paradigm in the next section.

## 3 MEAN-FIELD MULTI-AGENT PROXIMAL POLICY OPTIMIZATION

Motivated by Theorem 1, we present an algorithm to learn the *finite-population* optimal identical team policy. We build our algorithm based on the proximal policy optimization (PPO) framework due to its simplicity and effectiveness. While PPO has shown promising performance in cooper­ative tasks including MFC problems (Yu et al., 2022; Cui et al., 2024), its application in mixed competitive-collaborative scenarios is less studied. In the sequel, we introduce our key contribution: MF-MAPPO. We initialize two pairs of actor-critic networks, one for each team, deployed to learn the identical policy used by each team, see Figure 1(b). Specifically, we introduce a *minimally-informed critic* network by exploiting the MF information structure. The key point here is that we only require commonly accessible information for the critic network in order to learn the value func­tion (Proposition 1). Further, the private information available to each agent only *individually* enters the actor during training. This results in neural networks that scale well with the number of agents. We present the team actor-critic networks from the Blue team's perspective, and due to symmetry results extend naturally to the Red team.

**Minimally-Informed Critic.** The MF-MAPPO critic network of the Blue team evaluates the value function $V_{\text{Blue}}(\mu, \nu)$, which depends only on the common information (MFs)—assumed to be avail­able at the time of training—and is *independent* of the joint agent states and actions. We use the parameter vector $\zeta_{\text{Blue}}$ to parameterize the critic network while minimizing the MSE loss

$$L_{\text{critic}}(\zeta_{\text{Blue}}) = \frac{1}{|B|} \sum_{k=1}^{|B|} \Big(V_{\text{Blue}}(\mu_k, \nu_k | \zeta_{\text{Blue}}) - \hat{R}_{\text{Blue},k}\Big)^2, \quad (7)$$

where $B$ refers to the mini-batch size and $\hat{R}_{\text{Blue},k}$ is the discounted reward-to-go for sample $k$. The following proposition results from weakly-coupled team rewards and the use of identical team policies-justifying the deployment of a minimally-informed critic network with only MF inputs.

**Proposition 1.** *Let $\mu_t^{N_1}$, and $\nu_t^{N_2}$ denote the EDs of a finite-population game obtained from iden­tical Blue and Red team policies $\phi_t \in \Phi_t$ and $\psi_t \in \Psi_t$, respectively. The team reward structure admits a critic that depends only on $\mu_t^{N_1}$ and $\nu_t^{N_2}$. Specifically, for each Blue team agent $i \in [N_1]$, the individual critic value function $V_{i,t}^{N_1,\phi_t}(x_{i,t}, \mu_t^{N_1}, \nu_t^{N_2})$ satisfies $V_{i,t}^{N_1,\phi_t}(x_{i,t}, \mu_t^{N_1}, \nu_t^{N_2}) = V_{\text{Blue},t}^{N_1,\phi_t}(\mu_t^{N_1}, \nu_t^{N_2})$, where $V_{\text{Blue},t}^{N_1,\phi_t}(\mu_t, \nu_t)$ is the team-level critic.*

Importantly, it reduces the learning problem to one critic network per team. Specifically, the shared team reward structure along with the assumption of homogeneous agents in each team enables us to evaluate the performance of a team's agent using the minimally-informed critic—even if the individual agent has additional local observations such as their actions and private states.

**Shared-Team Actor.** As discussed in earlier sections, the coordinator game is a useful theoretical construct but has limited practical value for real-world deployment since it relies on an infinite-population oracle for training and produces policies whose size scales poorly with the state-action space. We therefore directly train finite-population identical local policies, which preserve the mean-field structure while reducing complexity and improving tractability, with guarantees in Theorem 1. Not only is the approach computationally tractable in terms of the size of the policy, but is also more realistic in terms of sampling training data. We use a single actor network per team to learn identical

team policies. The actor optimizes a PPO-based objective with a decaying entropy bonus (Schulman et al., 2017; Huang et al., 2022), which promotes exploration and stabilizes learning in mean-field settings (Cui & Koeppl, 2022; Guan et al., 2022). Permutation invariance and identical team policies further allow a single replay buffer per team, reducing memory costs and simplifying experience collection. The PPO-based objective function of the Blue actor is given by:

$$L(\theta_{\text{Blue}}) = \frac{1}{|B|} \sum_{k=1}^{|B|} \Big[ \min\Big( g_k(\theta_{\text{Blue}}) A_k, \text{clip}_{[1-\epsilon, 1+\epsilon]}(g_k(\theta_{\text{Blue}})) A_k \Big) + \omega S(\phi_{\theta_{\text{Blue}}}(x_k, \mu_k, \nu_k)) \Big], \quad (8)$$

where, $g(\theta) = \frac{\phi_\theta(u|x,\mu,\nu)}{\phi_{\theta^{\text{old}}}(u|x,\mu,\nu)}$, $A_k$ is the generalized advantage function estimate (Schulman et al., 2018) and the tunable parameter $\omega$ weighs the contribution of the entropy term $S(\phi_\theta(x, \mu, \nu))$ and decays as training progresses.

### 3.1 THEORETICAL GUARANTEES

As described in Section 2, the theoretical benefits of MFTGs at the infinite-population limit remain of significant interest. Indeed, the following theorem shows that policy gradients obtained through finite-population training (using a finite-population simulator) converge to their infinite-population counterparts as the population size grows.

**Theorem 2.** *The approximate policy gradient of the infinite-population Blue (Red) team coordinator policy $\alpha$ ($\beta$) computed using local policies from the finite-population ZS-MFTG via MF-MAPPO ($\hat{J}^{N_1}(\alpha_\theta)$) uniformly tends to the true policy gradient as the population size increases, i.e., $\|\nabla_\theta J_\infty(\alpha_\theta) - \nabla_\theta \hat{J}^{N_1}(\alpha_\theta)\|_2 \to 0$ as $(N_1, N_2) \to \infty$, where $\| \cdot \|_2$ is the 2-norm.*

The results extend to the Red team. We next demonstrate the scalability of MF-MAPPO as a direct consequence of Theorem 1 and the infinite-population coordinator game, by showing that, under certain conditions, the learned team policies generalize to varying population sizes $(\bar{N}_1, \bar{N}_2)$ while maintaining performance guarantees. Theorem 3 allows MF-MAPPO to be trained on a smaller population and deployed to larger teams without additional tuning, significantly reducing computational costs while maintaining performance consistency and generalizability across population sizes.

**Theorem 3.** *Let $\mathcal{G}_1$ denote the finite-population game where the agents utilize the identical team policies $\phi_t^*$ and $\psi_t^*$ derived from MF-MAPPO trained on $\mathcal{G}_1$ and let the finite-population game $\mathcal{G}_2$ with the same state-action space, dynamics, and rewards, but with population sizes $\bar{N}_1$ and $\bar{N}_2$ such that $\bar{N}_1/\bar{N}_2 = N_1/N_2$ and $\min(\bar{N}_1, \bar{N}_2) \geq \min(N_1, N_2)$. Then, $(\phi_t^*, \psi_t^*)$ remain $\epsilon$-optimal for $\mathcal{G}_2$.*

## 4 MF-MAPPO FOR PARTIALLY OBSERVABLE MFTGS

To have strategies reactive to opponent's unexpected behaviors, one needs feedback on opponent's MF-distribution, which in practice, is often unavailable through direct means. We consider a partially observable ZS-MFTG, relevant to domains like competitive sports/battlefield, where decentralized decision-making relies on estimating the opponent's state distribution. The two main challenges are: 1) the sensitivity of MF policies $(\phi, \psi)$ to the MF $(\mu_t, \nu_t)$ feedback and 2) constructing valid, performance-preserving MF estimates that can serve as inputs to the trained MF policies.

To address the first challenge, we introduce a gradient penalty to the MF-MAPPO objective (8), enforcing Lipschitz continuity in the mean-field and ensuring robustness: small estimation errors induce only minor changes in actions distributions. The following proposition formalizes this idea.

**Proposition 2.** *If the log-probability of the Blue team policy is bounded such that for all $x \in \mathcal{X}, u \in \mathcal{U}, \mu \in \mathcal{P}(\mathcal{X}), \nu \in \mathcal{P}(\mathcal{Y})$, $\|\nabla_\eta \log \phi(u \mid x, \mu, \nu)\|_2 \leq L_\phi/2|\mathcal{U}|$, where the gradient is taken with respect to $\eta \in \mathcal{P}(\mathcal{X}) \times \mathcal{P}(\mathcal{Y}) \triangleq [\mu^\mathsf{T}, \nu^\mathsf{T}]^\mathsf{T}$ and $L_\phi > 0$, then $\phi(u \mid x, \mu, \nu)$ is Lipschitz continuous with Lipschitz constant $L_\phi$, i.e.,*

$$\sum_u |\phi_t(u|x, \hat{\mu}, \hat{\nu}) - \phi_t(u|x, \hat{\mu}', \hat{\nu}')| \leq L_\phi \big( \mathrm{d}_{\text{TV}}(\hat{\mu}, \hat{\mu}') + \mathrm{d}_{\text{TV}}(\hat{\nu}, \hat{\nu}') \big) \quad \forall \ x \in \mathcal{X}. \quad (9)$$

Similarly, we can define Lipschitz continuous policies for the Red team with constant $L_\psi$. This idea of penalizing the gradient of the policies was introduced in robotics to promote smooth and stable policies in order to aid sim-to-real transfer (Chen et al., 2024; Shin et al., 2025).

To address the second challenge, we require a filter that can estimate the opponent distribution at every time-step for each agent (e.g., $i \in [N_1]$ obtains an estimate of the Red team distribution at time $t$ given by $\hat{\nu}_{i,t}^{N_2}$) with accuracy guaranteed within a bounded tolerance ensuring agent actions and overall performance (3) remain within acceptable limits. Note that we formulate the problem from the Blue team's perspective. The results extend naturally to the Red team's perspective. Let the full-information and estimated MF trajectories be $\{\mathcal{M}^{N_1}, \mathcal{N}^{N_2}\}$ and $\{\hat{\mathcal{M}}^{N_1}, \hat{\mathcal{N}}^{N_2}\}$, respectively. We measure estimator performance for gradient-regularized MF-MAPPO (GR-MF-MAPPO) team policies $(\phi_t^*, \psi_t^*)$ via the cumulative regret between fully and partially observable MF rewards as:

$$\Delta J(\phi_t^*, \psi_t^*) = \mathbb{E}_{\phi^*, \psi^*} \left[ \left| \sum_{t=0}^{T} r_t(\mathcal{M}_t^{N_1}, \mathcal{N}_t^{N_2}) - \sum_{t=0}^{T} r_t(\hat{\mathcal{M}}_t^{N_1}, \hat{\mathcal{N}}_t^{N_2}) \right| \right]. \tag{10}$$

In fact, any $\epsilon$-accurate estimator can be utilized during the deployment of GR-MF-MAPPO.

**Proposition 3.** *Consider a given $\epsilon$-accurate estimator, i.e., $\mathrm{d}_{\mathrm{TV}}\left(\hat{\nu}_{i,t}^{N_2}, \hat{\nu}_t^{N_2}\right) < \epsilon$, for all $i, t$, where $\hat{\nu}_t^{N_2}$ is the true opponent MF at time $t$ and $\epsilon > 0$. For the identical team-policy pair $(\phi_t^*, \psi_t^*)$ obtained via gradient-regularized MF-MAPPO and deployed using this estimator, the cumulative regret satisfies $\Delta J(\phi_t^*, \psi_t^*) \leq K\epsilon + \mathcal{O}(1/\sqrt{\underline{N}})$ for some constant $K > 0$.*

We emphasize that MF-MAPPO is modular and policy inputs can be swapped with different estimates (using estimation/prediction algorithms) and still have good performance. Gradient regularization is key to ensure that minor errors in estimation do not result in extreme changes in MF trajectories and performance.

In lieu of Proposition 3, we propose a communication network-based decentralized estimation filter, namely, Dynamic-Projected Consensus (D-PC). It is an extension of the control-theoretic constrained consensus problem (Nedić & Liu, 2016) and addresses the shortcomings of the estimation algorithm proposed in Benjamin & Abate (2025b), namely, estimation under limited communication rounds, and ensuring valid estimates in the presence of errors. Following the connected graph topology used in MFGs (Benjamin & Abate, 2025b), we define a state-based visibility graph $\mathcal{G}_t^{\mathrm{viz}}$ and team-based communication graphs $\mathcal{G}_{\mathrm{Blue},t}^{\mathrm{com}}$ and $\mathcal{G}_{\mathrm{Red},t}^{\mathrm{com}}$. We also define the projection operator $\Omega_{\mathcal{R}(x)}[\eta] \triangleq \arg\min_{\omega \in \mathcal{R}(x)} \|\eta - \omega\|_2$ for $\eta, \omega \in \mathbb{R}^{|\mathcal{Y}|}$ where $\mathcal{R}(x)$ is a closed and convex set. We assume that all agents in the same state receive the same information, so naturally they have the same estimate, i.e., all Blue agents at state $x \in \mathcal{X}$ at time $t$ have the same estimate $\hat{\nu}_{x,t}^{N_2}$ of the Red team MF. We consider two time scales: $t$ for system dynamics (2) and $\tau$ for communication rounds. At time $t$, and $\tau = 0$, each Blue agent at state $x \in \mathcal{X}$ holds a belief $\hat{\nu}_{x,t}^{\tau=0}$ consistent with $\mathcal{G}_t^{\mathrm{viz}}$. At communication round $\tau$, agents share estimates $\hat{\nu}_{x,t}^{\tau-1}$ with neighbors defined by $\mathcal{G}_{\mathrm{Blue},t}^{\mathrm{com}}$ and perform for $R_{\mathrm{com}}$ communication rounds: a) weighted-average consensus; and b) a projection onto a closed and convex constraint set $\mathcal{R}(x)$. The set $\mathcal{R}(x)$ combines the Red team's MF components known with certainty by the Blue agents at state $x$—i.e., the observable states given by $\mathcal{G}_t^{\mathrm{viz}}$—with those that must be estimated. $\mathcal{R}(x)$ guarantees that operations such as information aggregation, averaging, or distributed communication do not alter the parts of the distribution that are known with certainty. See Appendix A for detailed definitions.

**Theorem 4.** *D-PC satisfies Proposition 3 with $\epsilon = \mathcal{O}\left(e^{-cR_{\mathrm{com}}}\right)$ with $c > 0$.*

## 5 NUMERICAL EXPERIMENTS

In this section, we evaluate MF-MAPPO across large-population scenarios using our custom-made benchmark simulation platform, `MFEnv`. Built as an extension of Gymnasium (Towers et al., 2024), `MFEnv` is developed specifically to facilitate research in MFTGs, supporting both finite-agent simulations and oracle-based infinite-population models. Unlike existing toolkits, `MFEnv` includes aggregate reward metrics, policy-versus-policy evaluation, and flexible APIs for custom *mean-field environments* that adhere to MF dynamics, rewards and information structures. We showcase MF-MAPPO's efficacy on three representative environments (1) a constrained-action variant of the classical rock–paper–scissors game (Raghavan, 1994), enabling validation against analytically computed equilibria, (2) a complex battlefield setting where Blue and Red teams engage in attack–defense tasks with higher-dimensional state and action spaces, requiring sophisticated team-level coordination (3) a ZS-MFTG between a virus and a population modeling a SIS-framework. Additional results and environments (not limited to ZS-MFTGs) in Appendix E.

Table 1: Performance comparison for cRPS

| Algorithm | Critic Input | Average Reward | NE Attained? | Training Time |
|-----------|--------------|----------------|--------------|---------------|
| MF-IPPO | $\{(x_{i,t}^{N_1}, \mu_t)\}_{i=1}^{N_1}$ | -0.340 | ✓ | 2:45:11 |
| MAPPO-PS | $\{(x_{i,t}^{N_1}, \mu_t, \nu_t)\}_{i=1}^{N_1}$ | -0.313 | ✓ | 2:47:54 |
| MAPPO-CC | $(\mathbf{x}_t^{N_1}, \mathbf{y}_t^{N_2}, \mu_t, \nu_t)$ | -0.342 | ✓ | 3:20:46 |
| DDPG-MFTG | $(\phi_t, \mu_t, \nu_t)$ | 3.774 | ✗ | 60:49:35 |
| MF-MAPPO | $(\mu_t, \nu_t)$ | **-0.331** | ✓ | **2:17:15** |

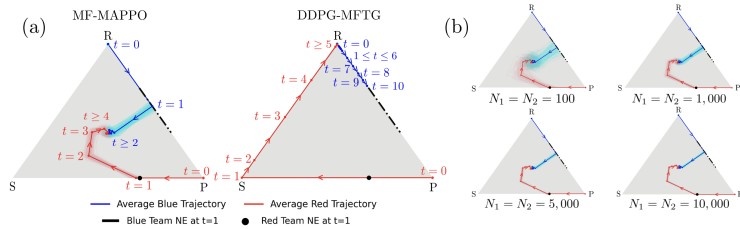

Figure 2: (a) 150 initializations of $\mu_{t=0} = [1,0,0]^\mathsf{T}$ and $\nu_{t=0} = [0,1,0]^\mathsf{T}$ for cRPS; $N_1 = N_2 = 1,000$ (b) Deploying MF-MAPPO trained on $N_1 = N_2 = 1,000$ to varying team sizes.

To the best of our knowledge, DDPG-MFTG (Shao et al., 2024a) is the only existing algorithm explicitly designed for mixed collaborative-competitive mean-field team problems. Accordingly, we mainly benchmark against DDPG-MFTG as well as several "house-made" mean-field adaptations of established MARL methods: namely, independent PPO (MF-IPPO) and two variants of MAPPO—parameter sharing (MAPPO-PS) and centralized critic (MAPPO-CC) (Yu et al., 2022). Centralized Q-learning methods such as QMix (Rashid et al., 2020), MADDPG (Lowe et al., 2017), and FACMAC (Peng et al., 2021) are not suitable baselines in our setting because their centralized action-value functions depend on the joint action and full system state, making them intractable in large-population regimes. Moreover, these approaches yield deterministic local policies, which tend to perform poorly since mixed strategies produce the heterogeneous team behaviors essential for mission success, as shown in Figure 4, making such comparisons inherently unfair.

**Constrained Rock-Paper-Scissors (cRPS).** The state space of each individual agent is $\mathcal{S} = \{\texttt{R}, \texttt{P}, \texttt{S}\}$, representing rock, paper, and scissors, respectively. We consider a non-trivial restriction of the action space to $\mathcal{A} = \{\texttt{CW}, \texttt{Stay}\}$ allowing agents to either move clockwise ($\texttt{R} \to \texttt{P}$, $\texttt{P} \to \texttt{S}$, $\texttt{S} \to \texttt{R}$) or remain idle, respectively. We assume deterministic transitions, where each action leads to a unique next state *deterministically*. At each time step $t$, the Blue (Red) team receives a team reward $r_t(\mu_t, \nu_t) = \mu_t^\mathsf{T} A \nu_t$ $(-\mu_t^\mathsf{T} A \nu_t)$ where $A$ is the standard RPS payoff matrix. Table 1 reports performance comparisons. MF-MAPPO achieves rewards closest to the analytical Nash value $(-1/3)$. Its minimally-informed critic preserves local-state privacy while substantially reducing critic dimensionality, resulting in a faster and more efficient algorithm under identical operating conditions. Figure 2(a) compares trajectories from MF-MAPPO and DDPG-MFTG. We see that MF-MAPPO successfully reaches the equilibrium distribution. By contrast, DDPG-MFTG diverges, relying on a mean-field oracle, which is valid only in the infinite-population limit, and "central players" that map mean-field distributions to deterministic policies without clipping or regularization, making it unstable. Unlike multi-agent DDPG extensions (e.g., MADDPG), which consider other teams' local policies, DDPG-MFTG conditions only on its own, limiting inter-team awareness. Figure 2(b) shows MF-MAPPO's scalability, where larger populations reduce noise and variance, aligning with Theorem 3.

**Battlefield Game.** To fully test the capability of MF-MAPPO on a more complex scenario, we propose a grid-based battlefield game where an individual agent's dynamics is highly coupled with both teams' distributions. The Blue agents aim is to reach their targets without being deactivated, while the Red agents learn to guard them. Deactivation occurs when one of the opponents holds a numerical advantage within a cell, incentivizing both teams to aggregate to reduce the risk of being deactivated by a numerically superior opponent. The Blue team's reward depends on the fraction of agents active at the target, the Red team's reward follows from the zero-sum structure, and (to avoid degeneracy) the Red agents are restricted from entering the target. All experiments use $N_1 = N_2 = 100$ agents on grids with varied targets (lilac) and obstacles (black).

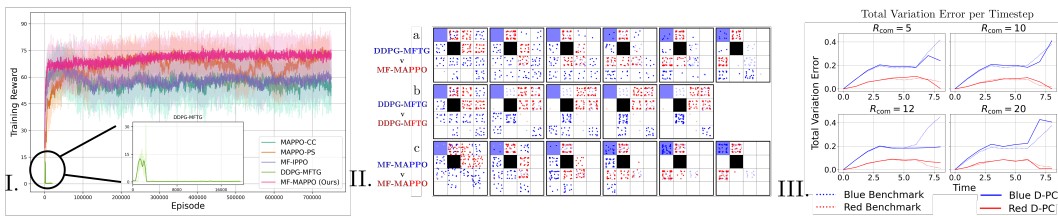

Figure 3: I. Training curve for Battlefield on a 4x4 grid (Blue team); II. Example configuration; III. Comparing $d_{TV}(\cdot)$ for D-PC and Benchmark estimator for different $R_{com}$.

We compare MF-MAPPO with the baselines by evaluating them head-to-head on a $4 \times 4$ grid with full MF information (Table 2). Teams trained with MF-MAPPO demonstrate consistently superior performance. Fixing the Red team policy (row) shows that the MF-MAPPO Blue team achieves the highest rewards, and fixing the Blue team policy (column) shows that the MF-MAPPO Red team performs the strongest as the minimizing team. These results highlight the benefits of using a minimally informed critic together with a shared-team actor: the resulting networks are smaller, more scalable, preserve private information, and alleviate credit-assignment challenges.

Figure 3II shows MF-MAPPO Red agents successfully cover both corridors and deactivate several Blue attackers II(a). Panels II(a) and II(b) highlight that DDPG-MFTG Blue agents do not aggressively pursue the target, illustrating their tendency to passively seek zero-reward outcomes (also reflected in Panel I) rather than take goal-directed actions, unlike II(c) where MF-MAPPO agents exhibit coordinated maneuvering, forming coalitions to reach the target. While cases I and II utilize complete observability, case III evaluates the proposed D-PC estimator against the estimator in Benjamin & Abate (2025a) (Benchmark) under a gradient-penalized MF-MAPPO policy when Red has full information and Blue estimates Red's distribution. Both estimators yield comparable total variation errors relative to the full-information case, with D-PC showing advantages under low communication budgets ($R_{com} < 20$). One can trivially show that the Benchmark satisfies Proposition 3 with $\epsilon = 1 - \mathcal{O}(R_{com}/|\mathcal{X}|)$ and assumes uniform estimates for unobserved states, which degrades estimation accuracy under limited communication. It also relies on accurate information from neighbors to ensure validity of estimates. In contrast, D-PC exchanges inexact information but applies state-dependent corrections (projection), preserving privacy and robustness.

In Figure 4, the Red team faces a dilemma in deciding which target to defend, while the Blue team exploits this ambiguity. Due to DDPG-MFTG's high computational cost and frequent network updates, it is excluded from our analysis. With no other baselines for such large-scale complex MFTGs, we qualitatively assess MF-MAPPO's performance. Figures 4I–II illustrate how identical policies can generate heterogeneous team behaviors, with Blue adapting target selection and Red reallocating defenses, highlighting the flexibility of the mean-field approximation. Furthermore, D-PC again outperforms the Benchmark under limited communication (III) ($R_{com} < 10$) and performs competitively otherwise. Cumulative rewards (IV) show only small deviations, consistent with Proposition 3 and Theorem 4, confirming that agents can rely on local observations with minimal communication, and that performance improves as communication bandwidth increases. We further empirically evaluate the effect of communication quality on opponent-estimate performance (V). Fixing $R_{com} = 7$, injecting zero-mean Gaussian noise with increasing variance produces larger deviations in cumulative rewards; however, these remain within 5% tolerance, due to the projection step after the average-consensus update, ensuring robust performance.

**Epidemiology.** In this scenario, the Red agents emulate a virus and can infect the Blue population; infected Blue individuals can further transmit the disease. Blue agents aim to maximize the size of their healthy population by avoiding infected peers. Because the virus cannot be eliminated,

Table 2: Performance evaluation for Battlefield

|  | MF-IPPO | MAPPO-PS | MAPPO-CC | DDPG-MFTG | MF-MAPPO |
|---|---|---|---|---|---|
| MF-IPPO | 61.54 | 73.85 | 70.60 | 0.00 | **86.26** |
| MAPPO-PS | 55.82 | 72.78 | 71.64 | 0.00 | **86.22** |
| MAPPO-CC | 51.60 | 68.81 | 65.58 | 0.00 | **82.58** |
| DDPG-MFTG | 65.26 | 75.14 | 77.21 | 0.00 | **85.95** |
| MF-MAPPO | **36.10** | **54.43** | **62.00** | 0.00 | **76.26** |

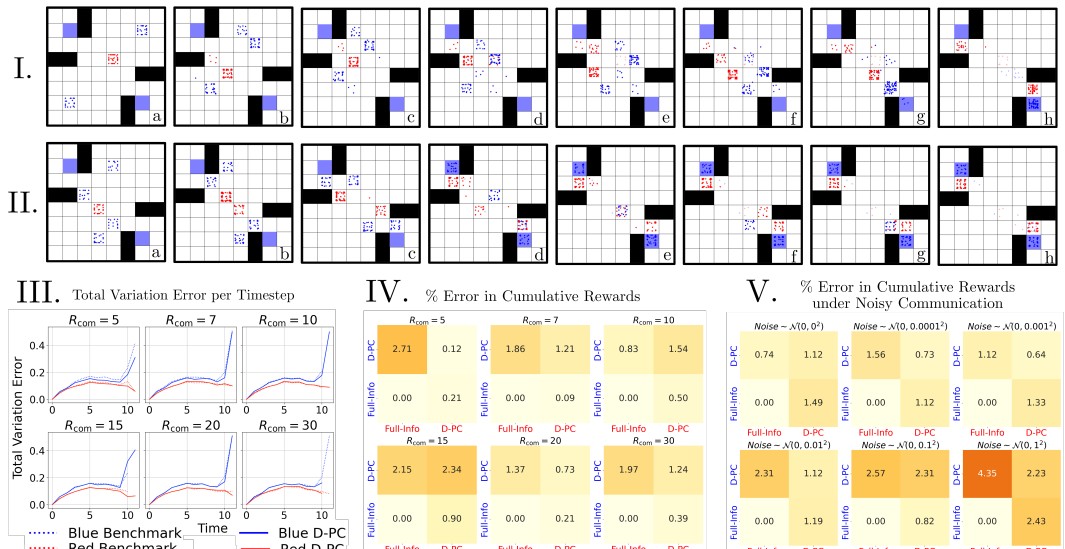

Figure 4: I. Red is concentrated; 30% Blue are at the bottom, rest are at the top II. Blue is evenly split, Red is concentrated III. Comparing $d_{TV}(\cdot)$ for D-PC and the Benchmark estimator for different $R_{com}$ IV. % error in cumulative rewards under varying communication bandwidths V. % error in cumulative rewards under noisy communication $R_{com} = 7$.

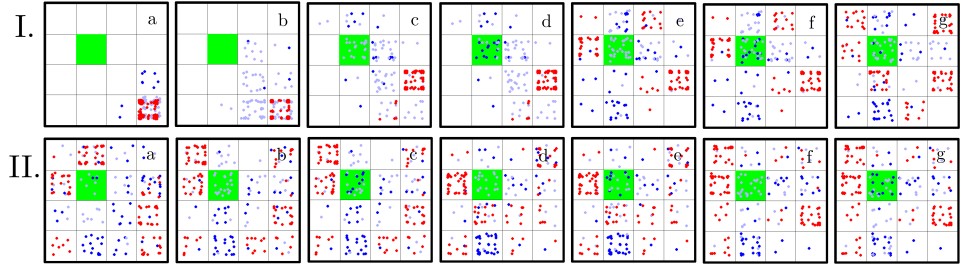

Figure 5: Example configurations of SIS-Epidemiology ZS-MFTG; green cell is the hospital.

the infected Blue agents must visit a hospital state, where healing occurs with some probability. Applying MF-MAPPO to this ZS-MFTG yields intuitive policies: decongestion-like behavior in which agents spread out to reduce contact (Figure 5). Meanwhile, the Red team actively tracks Blue agents to infect them and promptly positions itself around the hospital, while infected Blue agents strategically navigate toward the hospital for treatment.

# 6 CONCLUSION

We introduced MF-MAPPO, a novel MARL algorithm for large-population competitive team games that leverages finite mean-field approximation. With a minimally informed critic and shared team actor, MF-MAPPO scales efficiently while retaining performance, as shown against baselines such as DDPG-MFTG on cRPS and battlefield scenarios using the developed platform `MFEnv`. Despite shared policies, heterogeneous sub-population behaviors emerge, confirming that mean-field approximations do not hinder performance. We showed that MF-MAPPO naturally extends to partial observability via a simple gradient-regularized training scheme, and proposed D-PC, a decentralized mean-field estimator that ensures accuracy and strong performance when integrated with it. Empirically, D-PC outperforms baselines under limited communication. Limitations include MF-MAPPO's scaling with state dimensionality, which motivates future work on dimensionality reduction (e.g., kernel embeddings). Additional directions include extending D-PC to noisy settings and to more general, time-varying network topologies.

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
