---

**Initialize:** NN parameters $\{\theta_{\text{Blue}}, \zeta_{\text{Blue}}\}$ and $\{\theta_{\text{Red}}, \zeta_{\text{Red}}\}$; step size sequences $\{\alpha_m\}$ and $\{\beta_m\}$; entropy decay sequence $\{\omega_m\}$
**for** $i = 1, 2, \ldots$ **do**
    $(\phi_{\theta_{\text{Blue}}^{\text{old}}}, \psi_{\theta_{\text{Red}}^{\text{old}}}) \leftarrow (\phi_{\theta_{\text{Blue}}}, \psi_{\theta_{\text{Red}}})$
    **for** $t = 0, 1, \ldots, T_{\text{rollout}}$ **do**
        Sample joint actions
        $u_{i,t} \sim \phi_{\theta_{\text{Blue}}^{\text{old}}}(x_{i,t}, \mu_t^{N_1}, \nu_t^{N_2})$, $v_{j,t} \sim \psi_{\theta_{\text{Red}}^{\text{old}}}(y_{j,t}, \mu_t^{N_1}, \nu_t^{N_2})$
        Step environment according to kernels $(f_t, g_t)$
        Collect samples $(\mathbf{x}_{t+1}^{N_1}, \mathbf{y}_{t+1}^{N_2}, \mu_{t+1}^{N_1}, \nu_{t+1}^{N_2}, \mathbf{u}_t^{N_1}, \mathbf{v}_t^{N_2}, r_t)$
    **end for**
    **for** $K$ epochs **do**
        Update $\{\theta_{\text{Blue}}, \zeta_{\text{Blue}}\}$ and $\{\theta_{\text{Red}}, \zeta_{\text{Red}}\}$ using (7-8)
    **end for**
**end for**
**Return:** $(\phi_{\theta_{\text{Blue}}}, \psi_{\theta_{\text{Red}}})$

---

## A  DETAILS OF D-PC

**Definition 3** (Visibility Graph). *The bipartite state-based visibility graph at time $t$, denoted by $\mathcal{G}_t^{\text{viz}} = (\mathcal{X}, \mathcal{Y}, \mathcal{E}_t^{\text{viz}})$, is an undirected graph where (i) The vertex sets consist of the states $x \in \mathcal{X}$ and $y \in \mathcal{Y}$; (ii) An undirected edge $(x, y) \in \mathcal{E}_t^{\text{viz}}$ exists if and only if all Blue agents at state $x$ are mutually visible with all Red agents at state $y$ at time $t$; that is, every Blue agent at state $x$ is visible to every Red agent at state $y$ at time $t$.*

Before describing the communication architecture, we define $\mathcal{X}_t^o \subseteq \mathcal{X}$ such that $\hat{\mu}_t(x) > 0$ if and only if $x \in \mathcal{X}_t^o$.

**Definition 4** (Communication Graph). *The state-based communication graph at time $t$, denoted by $\mathcal{G}_{\text{Blue},t}^{\text{com}} = (\mathcal{X}_t^o, \mathcal{E}_t^{\text{com}})$, is an undirected graph where (i) The vertex set $\mathcal{X}_t^o$ consists of the states $x \in \mathcal{X}_t^o$; (ii) An undirected edge $(x_p, x_q) \in \mathcal{E}_t^{\text{com}}$ exists if and only if all agents at state $x_p$ can communicate with all agents at state $x_q$ at time $t$.*

We denote the set of neighbors of vertex $v$ of $\mathcal{G}_t^{\text{viz}}$ and $\mathcal{G}_{\text{Blue},t}^{\text{com}}$ respectively, as $\mathcal{N}_t^{\text{viz}}(v)$ and $\mathcal{N}_{\text{Blue},t}^{\text{com}}(v)$. $\mathcal{Y}_t^o \subseteq \mathcal{Y}$, $\mathcal{G}_{\text{Red},t}^{\text{com}}$ and $\mathcal{N}_{\text{Red},t}^{\text{com}}(v)$ are defined similarly. The following assumption on the topology of the communication graph will simplify the mean-field estimator design problem.

**Assumption 2.** *We assume that $\mathcal{G}_{\text{Blue},t}^{\text{com}}$ and $\mathcal{G}_{\text{Red},t}^{\text{com}}$ are connected for all $t$.*

At time $t$, we assume that each agent $i \in [N]$ observes the population distribution of its opponent's visible states $y \in \mathcal{N}_t^{\text{viz}}(x_{i,t})$. Thus, from the perspective of agent $i$, the ED $\hat{\nu}_{i,t}^{N_2}$ is defined on the constrained simplex $\hat{\nu}_{i,t}^{N_2} \in \mathcal{R}(x_{i,t}) \triangleq \{\eta \in \mathcal{P}(\mathcal{Y}) \mid \eta(z) = \hat{\nu}_t^{N_2}(z), \ \forall \ z \in \mathcal{N}_t^{\text{viz}}(x_{i,t})\}$. given by $\mathcal{R}(x_{i,t})$ combines the Red team's mean-field components known with certainty by Blue agent $i$—i.e., the observable states $\mathcal{N}_t^{\text{viz}}(x_{i,t})$—with those that must be estimated.

**Proposition 4.** *For all $x \in \mathcal{X}$, $\mathcal{R}(x)$ is a closed and convex set.*

It can be easily seen that the true mean-field distribution is an element of this constrained simplex for each agent, i.e., $\hat{\nu}_t^{N_2} \in \mathcal{R}(x_{i,t})$ for all $i \in [N_1]$ and time $t$. The following assumption imposes uniformity of the estimate of the ED across agents in a given state.

**Assumption 3.** *We assume that $\hat{\nu}_{i,t}^{N_2} = \hat{\nu}_{j,t}^{N_2} \triangleq \hat{\nu}_{x,t}^{N_2}$ for all agents satisfying $x_{i,t}^{N_1} = x_{j,t}^{N_1} = x$.*

**Definition 5.** *Let $x \in \mathcal{X}$. The projection operator $\Omega_{\mathcal{R}(x)} : \mathbb{R}^{|\mathcal{Y}|} \to \mathcal{R}(x)$ where $\mathcal{R}(x)$ is a closed and convex set is defined as $\Omega_{\mathcal{R}(x)}[\eta] = \omega^* \triangleq \arg\min_{\omega \in \mathcal{R}(x)} \|\eta - \omega\|_2, \ \eta \in \mathbb{R}^{|\mathcal{Y}|}$. where $\|\cdot\|_2$ is the standard Euclidean norm.*

At each communication round, D-PC performs the following two steps:

1. Weighted average consensus:

$$\xi_{x,t}^{\tau} = \sum_{z \in \mathcal{N}_{\text{Blue},t}^{\text{com}}(x) \cup x} w_{(x,z)}^{\text{Blue}} \hat{\nu}_{z,t}^{\tau-1}, \quad \sum_z w_{(x,z)}^{\text{Blue}} = 1, \quad x \in \mathcal{X}^o. \tag{A.1}$$

2. Projection onto the constraint set:

$$\hat{\nu}_{x,t}^{\tau} = \Omega_{\mathcal{R}(x)} \left[ \xi_{x,t}^{\tau} \right], \ x \in \mathcal{X}^o. \tag{A.2}$$

**Assumption 4.** *The matrix $W^{\text{Blue}} \in \mathbb{R}^{|\mathcal{X}^o| \times |\mathcal{X}^o|}$ formed by the non-negative weights is symmetric and doubly-stochastic. Furthermore, it respects the sparsity structure of the communication graph $\mathcal{G}^{\text{com}}$, that is, $w_{(x,z)}^{\text{Blue}} > 0$ if and only if $z \in \mathcal{N}^{\text{com}}(x) \cup x$. $W^{\text{Red}}$ is constructed similarly.*

To satisfy Assumption 4, one may use the well-known Metropolis Matrix (Xiao et al., 2006). We define the smallest non-zero entry of $\alpha$ as

$$\theta \triangleq \min_{(x,y)} \{ w_{(x,y)} > 0 \}. \tag{A.3}$$

For clarity of the D-PC pseudocode present in Algorithm A, the estimation is presented from the Blue team's perspective which models the distribution of its opponent Red team. One can simultaneously run this estimation algorithm from the Red team's perspective.

---

**Algorithm 2** Dynamic Projected Consensus for Mean-Field Estimation (**D-PC**)

---

**Initialize:** $\hat{\mu}_{t=0}^{N_1} = \mu_{t=0}^{N_1}$ and $\hat{\nu}_{t=0}^{N_2} = \nu_{t=0}^{N_2}$, identical MF-MAPPO trained Lipschitz policies $(\phi^*, \psi^*)$ and graphs $\mathcal{G}_{t=0}^{\text{viz}}$ and $(\mathcal{G}_{\text{Blue},t}^{\text{com}}, \mathcal{G}_{\text{Red},t}^{\text{com}})$
**for** $t = 0, \dots, T$ **do**
    Update the graphs $\mathcal{G}_t^{\text{viz}}$ and $(\mathcal{G}_{\text{Blue},t}^{\text{com}}, \mathcal{G}_{\text{Red},t}^{\text{com}})$.
    Receive reward $r_t(\hat{\mu}_t^{N_1}, \hat{\nu}_t^{N_2})$
    Set $\tau = 0$
    Initialize $\hat{\nu}_{x,t}^{N_2,\tau=0}$ ( $\hat{\mu}_{y,t}^{N_1,\tau=0}$) for all $x \in \mathcal{X}_t^o$ ($y \in \mathcal{Y}_t^o$) using $\mathcal{G}_{t=0}^{\text{viz}}$
    Compute matrix $W^{\text{Blue}}(W^{\text{Red}})$
    **for** $\tau < R_{\text{com}}$ **do**
        Communicate mean-field estimate to neighbors based on $\mathcal{G}_{\text{Blue},t}^{\text{com}}(\mathcal{G}_{\text{Red},t}^{\text{com}})$
        $\tau = \tau + 1$
        **for** all $x \in \mathcal{X}_t^o$ ($y \in \mathcal{Y}_t^o$) **do**
            Compute weighted average consensus (A.1)
            Project result onto constraint set (A.2)
        **end for**
    **end for**
    **Return:** $\hat{\nu}_{x,t}^{N_2,R_{\text{com}}}(\hat{\mu}_{y,t}^{N_1,R_{\text{com}}})$
**end for**

---

**Remark 1.** *One could alternatively cast this problem as a Partially Observable Markov Decision Process (PO-MDP) (Bernstein et al., 2002), where the environment provides arbitrary observations. However, our focus is on policies that explicitly depend on the mean-field distributions of the two teams, for two key reasons (1) Distinct opponent strategies induce different mean-field trajectories, and closed-loop feedback on these distributions enables an appropriate response to the strategies deployed by the opponent (Guan et al., 2024a) and (2) Access to team-level distributions facilitates credit assignment (Pignatelli et al., 2024) in MARL by allowing agents to reason about which collective distributions are optimal. Thus, having access to MF information—particularly that of the opponent—is desirable. Yet, it is unrealistic to assume that the environment directly provides these distributions, which motivates the design of estimation algorithms. Finally, unlike general PO-MDP formulations that often require maintaining long observation histories, our setting involves population sizes for which such history-based tracking is computationally infeasible in terms of memory.*

## B PROOF OF THEORETICAL RESULTS

**Theorem 5** (Guan et al. (2024a))**.** *The optimal identical Blue team policy $\phi_\infty^* \in \Phi$ obtained from the equivalent zero-sum coordinator game is $\epsilon$-optimal Blue team policy. Formally, for all joint*

states $\mathbf{x}^{N_1}$ and $\mathbf{y}^{N_2}$,

$$\min_{\psi^{N_2} \in \Psi^{N_2}} J^{N,\phi_\infty^*,\psi^{N_2}}(\mathbf{x}^{N_1}, \mathbf{y}^{N_2}) \geq \underline{J}^{N*}(\mathbf{x}^{N_1}, \mathbf{y}^{N_2}) - \mathcal{O}\Big(\frac{1}{\sqrt{\underline{N}}}\Big), \text{ where } \underline{N} = \min\{N_1, N_2\}.$$

(B.4)

**Theorem 1.** *The value of the optimal identical Blue team policy $\phi^*$ obtained from the* finite popula-*tion game is within $\epsilon$ of the finite-population lower game value defined in* (4). *Formally, for all joint states $\mathbf{x}^{N_1}$ and $\mathbf{y}^{N_2}$,*

$$\min_{\psi^{N_2}} J^{N,\phi^*,\psi^{N_2}}(\mathbf{x}^{N_1}, \mathbf{y}^{N_2}) \geq \underline{J}^{N*}(\mathbf{x}^{N_1}, \mathbf{y}^{N_2}) - \mathcal{O}\Big(\frac{1}{\sqrt{\underline{N}}}\Big), \text{ where } \underline{N} = \min\{N_1, N_2\}.$$

(6)

*Sketch of Proof.* The proof follows by restricting the optimization domain in 4 to that of identical team policies and then using the definition of the $\max$ operator in tandem with Theorem 5. $\square$

*Proof.* We have the following definition of the lower game value for the finite-population ZS-MFTG:

$$\underline{J}^{N*}(\mathbf{x}^{N_1}, \mathbf{y}^{N_2}) = \max_{\phi^{N_1} \in \Phi^{N_1}} \min_{\psi^{N_2} \in \Psi^{N_2}} J^{N,\phi^{N_1},\psi^{N_2}}(\mathbf{x}^{N_1}, \mathbf{y}^{N_2}).$$

(B.5)

Note that the maximization for the Blue team is being performed over the set of all team policies $\Phi^{N_1}$, including identical as well non-identical team policies. If we restrict ourselves to the set of identical team policies $\Phi \subseteq \Phi^{N_1}$ it follows immediately that

$$\underline{J}^{N*}(\mathbf{x}^{N_1}, \mathbf{y}^{N_2}) \geq \max_{\phi^{N_1} \in \Phi} \min_{\psi^{N_2} \in \Psi^{N_2}} J^{N,\phi^{N_1},\psi^{N_2}}(\mathbf{x}^{N_1}, \mathbf{y}^{N_2}).$$

(B.6)

Suppose that $\phi^*$ is the optimal identical Blue team policy obtained from the finite population game. It follows from (B.5) that

$$\underline{J}^{N*}(\mathbf{x}^{N_1}, \mathbf{y}^{N_2}) \geq \min_{\psi^{N_2} \in \Psi^{N_2}} J^{N,\phi^*,\psi^{N_2}}(\mathbf{x}^{N_1}, \mathbf{y}^{N_2}).$$

(B.7)

Furthermore, let $\phi_\infty^* \in \Phi$ be the optimal identical local Blue team policy obtained from the equivalent zero-sum infinite-population coordinator game (recall, one-to-one correspondence between coordinator policy and local identical team policy). By the definition of the optimality of $\phi^*$ in the space of identical team policies,

$$\min_{\psi^{N_2} \in \Psi^{N_2}} J^{N,\phi^*,\psi^{N_2}}(\mathbf{x}^{N_1}, \mathbf{y}^{N_2}) \geq \min_{\psi^{N_2} \in \Psi^{N_2}} J^{N,\phi_\infty^*,\psi^{N_2}}(\mathbf{x}^{N_1}, \mathbf{y}^{N_2}).$$

(B.8)

Using Theorem 5, and using (B.8), yields the following sequence of inequalities,

$$\underline{J}^{N*}(\mathbf{x}^{N_1}, \mathbf{y}^{N_2}) \geq \min_{\psi^{N_2} \in \Psi^{N_2}} J^{N,\phi^*,\psi^{N_2}}(\mathbf{x}^{N_1}, \mathbf{y}^{N_2})$$

$$\geq \min_{\psi^{N_2} \in \Psi^{N_2}} J^{N,\phi_\infty^*,\psi^{N_2}}(\mathbf{x}^{N_1}, \mathbf{y}^{N_2})$$

$$\geq \underline{J}^{N*}(\mathbf{x}^{N_1}, \mathbf{y}^{N_2}) - \mathcal{O}\Big(\frac{1}{\sqrt{\underline{N}}}\Big),$$

where $\underline{N} = \min\{N_1, N_2\}$, thereby completing the proof.

$\square$

**Theorem 2.** *The approximate policy gradient of the infinite-population Blue (Red) team coordinator policy $\alpha$ ($\beta$) computed using local policies from the finite-population ZS-MFTG via MF-MAPPO ($\hat{J}^{N_1}(\alpha_\theta)$) uniformly tends to the true policy gradient as the population size increases, i.e., $\|\nabla_\theta J_\infty(\alpha_\theta) - \nabla_\theta \hat{J}^{N_1}(\alpha_\theta)\|_2 \to 0$ as $(N_1, N_2) \to \infty$, where $\|\cdot\|_2$ is the 2-norm.*

*Sketch of Proof.* In MF-MAPPO we directly train the finite-population local policies $(\phi_t, \psi_t)$. As a result, the gradient of the coordinator policy $\nabla_\theta J(\alpha_\theta)$ only has action samples $u_{i,t}^{N_1}, i \in [N_1]$ from the finite-population local policies $\phi_t(\cdot|x_{i,t}^{N_1}, \mu_t^{N_1}, \nu_t^{N_2})$ and not $\pi_t^{N_1}(\cdot|x_{i,t}^{N_1}) \sim \alpha(\mu_t^{N_1}, \nu_t^{N_2})$ itself.

Thus, our first step is to construct an approximation of $\pi_t^{N_1}$ using the obtained finite-population action samples $u_{i,t}^{N_1}, i \in [N_1]$, resulting in an approximate finite-population policy gradient expression given by $\nabla_\theta \hat{J}^{N_1}(\alpha_\theta)$. Coupled with existing mean-field approximation results (Cui et al., 2023; Guan et al., 2024a; Shao et al., 2024a) for the infinite-population limit $N_1, N_2 \to \infty$ (LLN, etc.) we show convergence. □

*Proof.* From Guan et al. (2024a) it follows that there exists a one-to-one correspondence between the *deterministic* infinite-population coordinator policies and the local identical team policies followed by the finite-population Blue and Red agents. We extend this formulation to potentially *stochastic* infinite-population coordinator policies and the local finite-population policies as follows: a stochastic Blue coordination policy $\alpha \in \mathcal{A}$ induces an identical team policy $\phi \in \Phi$ according to the rule

$$\phi_t(u_t|x_t, \mu_t^\rho, \nu_t^\rho) = \int_{\pi \in \Pi} \pi_t(u_t|x_t)\alpha_t(\pi_t|\mu_t^\rho, \nu_t^\rho)d\pi \quad \forall \mu_t \in \mathcal{P}(\mathcal{X}), \ \nu_t \in \mathcal{P}(\mathcal{Y}), \ x_t \in \mathcal{X} \text{ and } u_t \in \mathcal{U},$$

(B.9)

where $\pi_t(u_t|x_t)$ corresponds to the identical local policies for all states $x \in \mathcal{X}$ prescribed by the coordinator, i.e., $\pi_t \sim \alpha(\mu_t^\rho, \nu_t^\rho)$.

It is important to note that in MF-MAPPO we directly work with the finite-population local policies $(\phi, \psi)$, unlike prior works that train on the infinite-population coordinator policy, see Section 1. Consequently, the gradient of the coordinator policy $\nabla_\theta J(\alpha_\theta)$ computed using this finite-population approximation (denoted $\nabla_\theta \hat{J}^{N_1}(\alpha_\theta)$) only has action samples $u_{i,t}^{N_1}, i \in [N_1]$ from the finite-population local policies $\phi_t(\cdot|x_{i,t}^{N_1}, \mu_t^{N_1}, \nu_t^{N_2})$ and not direct access to $\pi_t^{N_1} \sim \alpha(\mu_t^{N_1}, \nu_t^{N_2})$. Thus, our first step is to construct an approximation of the policy based on the obtained action samples. We define this approximate policy from the obtained samples as follows,

$$\hat{\pi}_t^{N_1}(u|x) = \begin{cases} \frac{\sum_{i=1}^{N_1} \mathbb{1}_x\left(X_{i,t}^{N_1}\right)\mathbb{1}_u\left(U_{i,t}^{N_1}\right)}{N_1 \mathcal{M}_t^{N_1}(x)} & \text{if } \mathcal{M}_t^{N_1}(x) > 0, \\ 1/|\mathcal{U}| & \text{if } \mathcal{M}_t^{N_1}(x) = 0. \end{cases}$$

(B.10)

We can similarly define $\hat{\sigma}_t^{N_2}$ for the Red team. Using this constructed empirical policy $\hat{\pi}_t^{N_1}$ from the sampled actions $\mathbf{u}_t^{N_1}$, the policy gradient $\nabla_\theta \hat{J}^{N_1}(\alpha_\theta)$ for the finite-population ZS-MFTG is,

$$\nabla_\theta \hat{J}^{N_1}(\alpha_\theta) = \sum_{t=0}^\infty \gamma^t \mathbb{E}_{\mathbf{u}^{N_1} \sim \phi, \mathbf{v}^{N_2} \sim \psi} \left[ Q^{N,\alpha,\beta}(\mu_t^{N_1}, \nu_t^{N_2}, \hat{\pi}_t^{N_1}, \hat{\sigma}_t^{N_2}) \nabla_\theta \log \alpha_\theta(\hat{\pi}_t^{N_1}|\mu_t^{N_1}, \nu_t^{N_2}) \right],$$

(B.11)

where,

$$Q^{N,\alpha,\beta}(\mu_t^{N_1}, \nu_t^{N_2}, \hat{\pi}_t^{N_1}, \hat{\sigma}_t^{N_2})$$
$$= \mathbb{E}_{\mathbf{u}^{N_1} \sim \phi, \mathbf{v}^{N_2} \sim \psi} \left[ \sum_{\tau=0}^\infty \gamma^\tau r(\mu_\tau^{N_1}, \nu_\tau^{N_2}) \big| \mu_{\tau=0}^{N_1} = \mu_t^{N_1}, \nu_{\tau=0}^{N_2} = \nu_t^{N_2}, \pi_{\tau=0}^{N_1} = \hat{\pi}_t^{N_1}, \sigma_{\tau=0}^{N_2} = \hat{\sigma}_t^{N_2} \right].$$

(B.12)

Recall that $\mathbf{u}^{N_1}$ and $\mathbf{v}^{N_2}$ enter (B.11) and (B.12) through the construction of $\hat{\pi}_t^{N_1}$ (B.10) and $\hat{\sigma}_t^{N_2}$ respectively. Furthermore, we have the following expression for the gradient of the infinite-population coordinator game:

$$\nabla_\theta J(\alpha_\theta) = \sum_{t=0}^\infty \gamma^t \mathbb{E}_{\pi \sim \alpha, \sigma \sim \beta} \left[ Q^{\alpha,\beta}(\mu_t, \nu_t, \pi_t, \sigma_t) \nabla_\theta \log \alpha_\theta(\pi_t|\mu_t, \nu_t) \right],$$

(B.13)

where,

$$Q^{\alpha,\beta}(\mu, \nu, \pi, \sigma) = \mathbb{E}_{\pi \sim \alpha, \sigma \sim \beta} \left[ \sum_{\tau=0}^\infty \gamma^\tau r(\mu_\tau, \nu_\tau) \big| \mu_{\tau=0} = \mu, \nu_{\tau=0} = \nu, \pi_{\tau=0} = \pi, \sigma_{\tau=0} = \sigma \right].$$

(B.14)

While (B.12) and (B.14) utilize the joint coordinator policies $(\alpha, \beta)$, the policy gradients (B.11) and (B.13) are taken for each team *individually*. Consequently the policy gradient analysis is done

on a per-team basis. We focus on the policy gradient with respect to the Blue team but the analysis of the Red team's policy gradient is symmetric and can be proved by an identical approach.

The following lemma relates the expectation in (B.11) to $(\alpha, \beta)$ using (B.9).

**Lemma 1.** *Given a function* $g : \mathcal{U} \rightarrow \mathbb{R}$ *of the joint actions* $\mathbf{u}_t^{N_1}$ *such that* $u_{i,t}^{N_1} \sim \phi(\cdot|x_{i,t}^{N_1}, \mu_t^{N_1}, \nu_t^{N_2})$*, or equivalently,* $\mathbf{u}_t^{N_1} \sim \phi(\cdot|\mathbf{x}_t^{N_1}, \mu_t^{N_1}, \nu_t^{N_2})$*,*

$$\mathbb{E}_{\mathbf{u}_t^{N_1} \sim \phi}\left[g(\mathbf{u}_t^{N_1})\right] = \mathbb{E}_{\pi \sim \alpha}\left[\mathbb{E}_{\mathbf{u}_t^{N_1} \sim \pi}\left[g(\mathbf{u}_t^{N_1})\right]\right]$$

*Proof.* By the definition of expectation,

$$\mathbb{E}_{\mathbf{u}_t^{N_1} \sim \phi}\left[g(\mathbf{u}_t^{N_1})\right] = \sum_{\mathbf{u}_t^{N_1}} g(\mathbf{u}_t^{N_1})\phi(\mathbf{u}_t^{N_1}|\mathbf{x}_t^{N_1}, \mu_t^{N_1}, \nu_t^{N_2}),$$

where the sum is taken over all the possible joint actions that can be sampled from $\phi$. By using the definition of $\phi$ from (B.9),

$$\sum_{\mathbf{u}_t^{N_1}} g(\mathbf{u}_t^{N_1})\phi(\mathbf{u}_t^{N_1}|\mathbf{x}_t^{N_1}, \mu_t^{N_1}, \nu_t^{N_2}) = \sum_{\mathbf{u}_t^{N_1}} g(\mathbf{u}_t^{N_1}) \underbrace{\int_{\pi \in \Pi} \pi_t(\mathbf{u}_t^{N_1}|\mathbf{x}_t^{N_1})\alpha_t(\mu_t^{N_1}, \nu_t^{N_2})d\pi}_{=\mathbb{E}_{\pi \sim \alpha}\left[\pi_t(\mathbf{u}_t^{N_1}|\mathbf{x}_t^{N_1})\right]}$$

$$= \sum_{\mathbf{u}_t^{N_1}} g(\mathbf{u}_t^{N_1})\mathbb{E}_{\pi \sim \alpha}\left[\pi_t(\mathbf{u}_t^{N_1}|\mathbf{x}_t^{N_1})\right]$$

$$= \sum_{\mathbf{u}_t^{N_1}} \mathbb{E}_{\pi \sim \alpha}\left[g(\mathbf{u}_t^{N_1})\pi_t(\mathbf{u}_t^{N_1}|\mathbf{x}_t^{N_1})\right]$$

$$= \mathbb{E}_{\pi \sim \alpha}\left[\sum_{\mathbf{u}_t^{N_1}} g(\mathbf{u}_t^{N_1})\pi_t(\mathbf{u}_t^{N_1}|\mathbf{x}_t^{N_1})\right],$$

where the last two expressions follow from linearity (and finiteness) of expectation. It follows then,

$$\mathbb{E}_{\mathbf{u}_t^{N_1} \sim \phi}\left[g(\mathbf{u}_t^{N_1})\right] = \mathbb{E}_{\pi \sim \alpha}\left[\mathbb{E}_{\mathbf{u}_t^{N_1} \sim \pi}\left[g(\mathbf{u}_t^{N_1})\right]\right]$$

$\square$

We also explicitly state results regarding the boundedness of terms appearing in (B.11)-(B.14).

**Remark 2.** *As mentioned in the main text,* $r_t \in [-R_{\max}, R_{\max}]$*. Consequently, the Q-functions* $Q^{\alpha, \beta}$ *and* $Q^{N, \alpha, \beta}$ *from (B.14) and (B.12) for all* $\gamma \in (0, 1)$ *are uniformly bounded with,*

$$\|Q^{\alpha, \beta}\|_\infty \leq \frac{R_{\max}}{1 - \gamma}, \quad \text{and} \quad \|Q^{N, \alpha, \beta}\|_\infty \leq \frac{R_{\max}}{1 - \gamma}.$$

We further assume continuity with respect to the local policies, i.e.,

**Assumption 5.** *The Q-functions* $Q^{\alpha, \beta}$ *and* $Q^{N, \alpha, \beta}$ *from (B.14) and (B.12) are continuous in inputs* $(\pi, \sigma)$ *and* $(\hat{\pi}^{N_1}, \hat{\sigma}^{N_2})$*.*

It is pertinent to note that the action-space $\Pi$ and $\Sigma$ are continuous. Thus, although Assumption 5 seems restrictive, the use of function approximators to train $Q^{\alpha, \beta}$ and $Q^{N, \alpha, \beta}$ like neural networks (endowed with continuous activation functions) in such continuous action-space settings, ensures that the resultant functions are continuous in their inputs Lillicrap et al. (2019).

The following assumption, as done in many works (Cui et al., 2024; 2023), states that the gradient of coordinator policy is continuous and uniformly bounded, i.e.,

**Assumption 6.** *The log-gradient* $\nabla \log \alpha_\theta(\pi_t|\mu_t, \nu_t)$ *is* $L_\nabla$*-Lipschitz continuous and uniformly bounded.*

Furthermore, we have the following result from Guan et al. (2024a):

**Corollary 1.** *Let $\mathbf{X}_t^{N_1}$, $\mathbf{Y}_t^{N_2}$, $\mathcal{M}_t^{N_1}$, and $\mathcal{N}_t^{N_2}$ be the joint states and the corresponding EDs of a finite-population game. Denote the next Blue ED induced by an identical Blue team policy $\phi_t \in \Phi_t$ as $\mathcal{M}_{t+1}^{N_1}$. Then, the following holds:*

$$\mathbb{E}_{\phi_t}\left[\mathrm{d}_{\mathrm{TV}}\big(\mathcal{M}_{t+1}^{N_1}, \mu_{t+1}\big)\big|\, \mathbf{X}_t^{N_1}, \mathbf{Y}_t^{N_2}\right] \leq \frac{|\mathcal{X}|}{2}\sqrt{\frac{1}{N_1}},$$

*where $\mu_{t+1} = \mathcal{M}_t^{N_1} F_t(\mathcal{M}_t^{N_1}, \mathcal{N}_t^{N_2}, \phi_t)$.*

We are interested in deriving an explicit result for $\|\nabla_\theta J(\alpha_\theta) - \nabla_\theta \hat{J}^{N_1}(\alpha_\theta)\|_2$ as the population sizes $(N_1, N_2) \to \infty$. In particular, using shorthand notation $\mathbb{E}_{\alpha,\beta}[\cdot] \triangleq \mathbb{E}_{\pi\sim\alpha,\sigma\sim\beta}[\cdot]$ and $\mathbb{E}_{\phi,\psi}[\cdot] \triangleq \mathbb{E}_{\mathbf{u}^{N_1}\sim\phi,\mathbf{v}^{N_2}\sim\psi}[\cdot]$)

$$\|\nabla_\theta J(\alpha_\theta) - \nabla_\theta \hat{J}^{N_1}(\alpha_\theta)\|_2$$

$$= \Big\| \sum_{t=0}^{\infty} \gamma^t \mathbb{E}_{\alpha,\beta}\left[Q^{\alpha,\beta}(\mu_t,\nu_t,\pi_t,\sigma_t)\nabla_\theta \log \alpha_\theta(\pi_t|\mu_t,\nu_t)\right]$$

$$- \gamma^t \mathbb{E}_{\phi,\psi}\left[Q^{N,\alpha,\beta}(\mu_t^{N_1},\nu_t^{N_2},\hat{\pi}_t^{N_1},\hat{\sigma}_t^{N_2})\nabla_\theta \log \alpha_\theta(\hat{\pi}_t^{N_1}|\mu_t^{N_1},\nu_t^{N_2})\right] \Big\|_2.$$

For the rest of the proof, we implicitly assume $\|\cdot\| \equiv \|\cdot\|_2$ to denote the 2-norm. By Lemma 1, we can rewrite the above as

$$\|\nabla_\theta J(\alpha_\theta) - \nabla_\theta \hat{J}^{N_1}(\alpha_\theta)\|$$

$$= \Big\| \sum_{t=0}^{\infty} \gamma^t \mathbb{E}_{\alpha,\beta}\Big[Q^{\alpha,\beta}(\mu_t,\nu_t,\pi_t,\sigma_t)\nabla_\theta \log \alpha_\theta(\pi_t|\mu_t,\nu_t)$$

$$- \mathbb{E}_{\mathbf{u}\sim\pi,\mathbf{v}\sim\sigma}\left[Q^{N,\alpha,\beta}(\mu_t^{N_1},\nu_t^{N_2},\hat{\pi}_t^{N_1},\hat{\sigma}_t^{N_2})\nabla_\theta \log \alpha_\theta(\hat{\pi}_t^{N_1}|\mu_t^{N_1},\nu_t^{N_2})\right] \Big] \Big\|,$$

where $\hat{\pi}_t^{N_1}, \hat{\sigma}_t^{N_2}$ depend on the random variables $\mathbf{u}_t^{N_1} \sim \pi, \mathbf{v}_t^{N_2} \sim \sigma$ respectively. We add and subtract the term $\mathbb{E}_{\mathbf{u}^{N_1}\sim\pi,\mathbf{v}^{N_2}\sim\sigma}\left[Q^{\alpha,\beta}(\mu_t^{N_1},\nu_t^{N_2},\hat{\pi}_t^{N_1},\hat{\sigma}_t^{N_2})\nabla_\theta \log \alpha_\theta(\hat{\pi}_t^{N_1}|\mu_t^{N_1},\nu_t^{N_2})\right]$ and split the gradient terms as follows.

$$\|\nabla_\theta J(\alpha_\theta) - \nabla_\theta \hat{J}^{N_1}(\alpha_\theta)\|$$

$$\leq \Big\| \sum_{t=0}^{\infty} \gamma^t \mathbb{E}_{\alpha,\beta}\Big[Q^{\alpha,\beta}(\mu_t,\nu_t,\pi_t,\sigma_t)\nabla_\theta \log \alpha_\theta(\pi_t|\mu_t,\nu_t)$$

$$- \mathbb{E}_{\mathbf{u}^{N_1}\sim\pi,\mathbf{v}^{N_2}\sim\sigma}\left[Q^{\alpha,\beta}(\mu_t^{N_1},\nu_t^{N_2},\hat{\pi}_t^{N_1},\hat{\sigma}_t^{N_2})\nabla_\theta \log \alpha_\theta(\hat{\pi}_t^{N_1}|\mu_t^{N_1},\nu_t^{N_2})\right] \Big] \Big\|$$

$$+ \Big\| \sum_{t=0}^{\infty} \gamma^t \mathbb{E}_{\alpha,\beta,\mathbf{u}^{N_1}\sim\pi,\mathbf{v}^{N_2}\sim\sigma}\left[\Big(Q^{\alpha,\beta}(\mu_t^{N_1},\nu_t^{N_2},\hat{\pi}_t^{N_1},\hat{\sigma}_t^{N_2}) - Q^{N,\alpha,\beta}(\mu_t^{N_1},\nu_t^{N_2},\hat{\pi}_t^{N_1},\hat{\sigma}_t^{N_2})\Big)\nabla_\theta \log \alpha_\theta(\hat{\pi}_t^{N_1}|\mu_t^{N_1},\nu_t^{N_2})\right] \Big\|.$$

$$(\text{B.15})$$

We analyze the two terms individually. For the first term,

$$\Big\| \sum_{t=0}^{\infty} \gamma^t \mathbb{E}_{\alpha,\beta}\Big[Q^{\alpha,\beta}(\mu_t,\nu_t,\pi_t,\sigma_t)\nabla_\theta \log \alpha_\theta(\pi_t|\mu_t,\nu_t)$$

$$- \mathbb{E}_{\mathbf{u}^{N_1}\sim\pi,\mathbf{v}^{N_2}\sim\sigma}\left[Q^{\alpha,\beta}(\mu_t^{N_1},\nu_t^{N_2},\hat{\pi}_t^{N_1},\hat{\sigma}_t^{N_2})\nabla_\theta \log \alpha_\theta(\hat{\pi}_t^{N_1}|\mu_t^{N_1},\nu_t^{N_2})\right] \Big] \Big\|, \qquad (\text{B.16})$$

Since the first term in the above expression is a constant with respect to the random variables $\mathbf{u}^{N_1}, \mathbf{v}^{N_2}$ and the expectation of a constant is the value itself, we can equivalently write (B.16) as

$$\Big\| \sum_{t=0}^{\infty} \gamma^t \mathbb{E}_{\alpha,\beta,\mathbf{u}^{N_1}\sim\pi,\mathbf{v}^{N_2}\sim\sigma}\Big[Q^{\alpha,\beta}(\mu_t,\nu_t,\pi_t,\sigma_t)\nabla_\theta \log \alpha_\theta(\pi_t|\mu_t,\nu_t)$$

$$- Q^{\alpha,\beta}(\mu_t^{N_1},\nu_t^{N_2},\hat{\pi}_t^{N_1},\hat{\sigma}_t^{N_2})\nabla_\theta \log \alpha_\theta(\hat{\pi}_t^{N_1}|\mu_t^{N_1},\nu_t^{N_2})\Big] \Big\|, \qquad (\text{B.17})$$

We can split the sum from $t = 0$ to $t = T$ and $t = T + 1$ to $t = \infty$, By uniform bounds on $Q^{\alpha,\beta}$ from Remark 2 and on $\nabla_\theta \log \alpha_\theta(\cdot \mid \mu_t^{N_1}, \nu_t^{N_2})$, $\nabla_\theta \log \alpha_\theta(\cdot \mid \mu_t, \nu_t)$ (Assumption 6), the sum from $t = T + 1$ to $t = \infty$ goes to zero for $T$ large enough (tail sequence). This enables us to focus on the summation from $t = 0$ to $t = T$, namely,

$$\Big\| \sum_{t=0}^{T} \gamma^t \mathbb{E}_{\alpha,\beta,\mathbf{u}^{N_1}\sim\pi,\mathbf{v}^{N_2}\sim\sigma} \Big[ Q^{\alpha,\beta}(\mu_t, \nu_t, \pi_t, \sigma_t) \nabla_\theta \log \alpha_\theta(\pi_t | \mu_t, \nu_t)$$
$$- Q^{\alpha,\beta}(\mu_t^{N_1}, \nu_t^{N_2}, \hat{\pi}_t^{N_1}, \hat{\sigma}_t^{N_2}) \nabla_\theta \log \alpha_\theta(\hat{\pi}_t^{N_1} | \mu_t^{N_1}, \nu_t^{N_2}) \Big] \Big\|. \tag{B.18}$$

Define $h(\mu, \nu, \pi, \sigma) \triangleq Q^{\alpha,\beta}(\mu, \nu, \pi, \sigma) \nabla_\theta \log \alpha_\theta(\pi | \mu, \nu)$. Assumptions 5 and 6 ensure continuity of the function $h(\mu, \nu, \pi, \sigma)$ with respect to all its inputs. Using this definition of $h$, rewrite (B.18) as

$$\Big\| \sum_{t=0}^{T} \gamma^t \mathbb{E}_{\alpha,\beta,\mathbf{u}\sim\pi,\mathbf{v}\sim\sigma} \Big[ h(\mu_t, \nu_t, \pi_t, \sigma_t) - h(\mu_t^{N_1}, \nu_t^{N_2}, \hat{\pi}_t^{N_1}, \hat{\sigma}_t^{N_2}) \Big] \Big\|. \tag{B.19}$$

Using the definition of the expectation operators,

$$\Big\| \sum_{t=0}^{T} \gamma^t \mathbb{E}_{\alpha,\beta,\mathbf{u}^{N_1}\sim\pi,\mathbf{v}^{N_2}\sim\sigma} \Big[ h(\mu_t, \nu_t, \pi_t, \sigma_t) - h(\mu_t^{N_1}, \nu_t^{N_2}, \hat{\pi}_t^{N_1}, \hat{\sigma}_t^{N_2}) \Big] \Big\|$$

$$= \Big\| \sum_{t=0}^{T} \gamma^t \int_\pi \int_\sigma h(\mu_t, \nu_t, \pi_t, \sigma_t) \beta_t(\mu_t, \nu_t) \alpha_t(\mu_t, \nu_t) d\sigma d\pi$$

$$- \int_\pi \int_\sigma \Big[ \sum_{\mathbf{u}^{N_1}} \sum_{\mathbf{v}^{N_2}} h(\mu_t^{N_1}, \nu_t^{N_2}, \hat{\pi}_t^{N_1}, \hat{\sigma}_t^{N_2}) \pi(\mathbf{u}^{N_1}|\mathbf{x}_t^{N_1}) \sigma(\mathbf{v}^{N_2}|\mathbf{y}_t^{N_1}) \Big] \beta_t(\mu_t, \nu_t) \alpha_t(\mu_t, \nu_t) d\sigma d\pi \Big\|$$

We already have, by weak LLN that $\mathrm{d}_{\mathrm{TV}}\big(\mu_t^{N_1}, \mu_t\big) \leq \mathcal{O}(1/\sqrt{N_1})$ and $\mathrm{d}_{\mathrm{TV}}\big(\nu_t^{N_2}, \nu_t\big) \leq \mathcal{O}(1/\sqrt{N_2})$, with the error going to zero as the population size becomes large, i.e., $N_1, N_2 \to \infty$. A similar argument using weak LLN can be made for $(\pi_t^{N_1}, \sigma_t^{N_2}) \to (\pi_t, \sigma_t)$ as $N_1, N_2 \to \infty$. Furthermore, as the population size becomes large, the empirical policies constructed $(\hat{\pi}_t^{N_1}, \hat{\sigma}_t^{N_2}) \to (\pi_t^{N_1}, \sigma_t^{N_2})$ because the number of samples of $\mathbf{u}_t^{N_1}$ and $\mathbf{v}_t^{N_2}$ scale with $(N_1, N_2)$.[1] Thus, it follows that $(\hat{\pi}_t^{N_1}, \hat{\sigma}_t^{N_2}) \to (\pi_t, \sigma_t)$ as the team sizes become large and, coupled with the continuity of $h$ with respect to $\hat{\pi}_t^{N_1}, \hat{\sigma}_t^{N_2}$, the following simplification holds:

$$\sum_{\mathbf{u}} \sum_{\mathbf{v}} h(\mu_t^{N_1}, \nu_t^{N_2}, \hat{\pi}_t^{N_1}, \hat{\sigma}_t^{N_2}) \pi(\mathbf{u}|\mathbf{x}_t^{N_1}) \sigma(\mathbf{v}|\mathbf{y}_t^{N_1}) \to \sum_{\mathbf{u}} \sum_{\mathbf{v}} h(\mu_t, \nu_t, \pi_t, \sigma_t) \pi(\mathbf{u}|\mathbf{x}_t^{N_1}) \sigma(\mathbf{v}|\mathbf{y}_t^{N_1})$$

$$= h(\mu_t, \nu_t, \pi_t, \sigma_t) \sum_{\mathbf{u}} \pi(\mathbf{u}^{N_1}|\mathbf{x}_t^{N_1}) \sum_{\mathbf{v}} \sigma(\mathbf{v}^{N_2}|\mathbf{y}_t^{N_1})$$

$$= h(\mu_t, \nu_t, \pi_t, \sigma_t)$$

For the second term in (B.15), we first focus on the term

$$\Big( Q^{\alpha,\beta}(\mu_t^{N_1}, \nu_t^{N_2}, \hat{\pi}_t^{N_1}, \hat{\sigma}_t^{N_2}) - Q^{N,\alpha,\beta}(\mu_t^{N_1}, \nu_t^{N_2}, \hat{\pi}_t^{N_1}, \hat{\sigma}_t^{N_2}) \Big).$$

The expectation in (B.12) is with respect to $\mathbf{u}^{N_1} \sim \phi, \mathbf{v}^{N_2} \sim \psi$. It follows from Lemma 1, we can rewrite (B.12) as

$$Q^{N,\alpha,\beta}(\mu_t^{N_1}, \nu_t^{N_2}, \hat{\pi}_t^{N_1}, \hat{\sigma}_t^{N_2})$$

$$= \mathbb{E}_{\pi\sim\alpha,\sigma\sim\beta,\mathbf{u}^{N_1}\sim\pi,\mathbf{v}^{N_2}\sim\sigma} \Big[ \sum_{\tau=0}^{\infty} \gamma^\tau r(\mu_\tau^{N_1}, \nu_\tau^{N_2}) \big| \mu_0^{N_1} = \mu_t^{N_1}, \nu_0^{N_2} = \nu_t^{N_2}, \pi_0^{N_1} = \hat{\pi}_t^{N_1}, \sigma_0^{N_2} = \hat{\sigma}_t^{N_2} \Big]$$

---

[1]Note that as $N_1 \to \infty$, $\hat{\pi}_t^{N_1}(\cdot|x) \neq \pi_t^{N_1}(\cdot|x)$ for states $x$ such that $\mu(x) = 0$ by the very definition of the approximation (B.10). However, it is easy to check that the mean-field trajectories remain the same and the unoccupied states have no role to play in the evolution.

On the other hand, the expectation in (B.14) is with respect to $\alpha, \beta$. Using a similar argument as (B.17), we can write (B.14) as

$$Q^{\alpha,\beta}(\mu_t^{N_1}, \nu_t^{N_2}, \hat{\pi}_t^{N_1}, \hat{\sigma}_t^{N_2})$$
$$= \mathbb{E}_{\pi\sim\alpha, \sigma\sim\beta, \mathbf{u}^{N_1}\sim\pi, \mathbf{v}^{N_2}\sim\sigma}\left[\sum_{\tau=0}^{\infty}\gamma^{\tau}r(\mu_{\tau}, \nu_{\tau})|\mu_0 = \mu_t^{N_1}, \nu_0 = \nu_t^{N_2}, \pi_0 = \hat{\pi}_t^{N_1}, \sigma_0 = \hat{\sigma}_t^{N_2}\right].$$

We apply the following lemma along with Assumption 6 to establish convergence of the second term in (B.15).

**Lemma 2.** *Under Assumption 1 and Lipschitz policies B.17, $Q^{N,\alpha,\beta} \to Q^{\alpha,\beta}$ as $N_1, N_2 \to \infty$.*

*Proof.* Assumption 1 allows the $Q^{N,\alpha,\beta}, Q^{\alpha,\beta}$ to have compact support. Thus, it suffices to prove pointwise convergence (Cui et al., 2023) for each $\tau$, i.e.,

$$\mathbb{E}\left[\gamma^{\tau}r(\mu_{\tau}^{N_1}, \nu_{\tau}^{N_2})|\mu_0^{N_1} = \mu_t^{N_1}, \nu_0^{N_2} = \nu_t^{N_2}, \pi_0^{N_1} = \hat{\pi}_t^{N_1}, \sigma_0^{N_2} = \hat{\sigma}_t^{N_2}\right] \to$$
$$\mathbb{E}\left[\gamma^{\tau}r(\mu_{\tau}, \nu_{\tau})|\mu_0 = \mu_t^{N_1}, \nu_0 = \nu_t^{N_2}, \pi_0 = \hat{\pi}_t^{N_1}, \sigma_0 = \hat{\sigma}_t^{N_2}\right] \tag{B.20}$$

as $N_1, N_2 \to \infty$, where the expectation is taken with respect to $\alpha, \beta, \mathbf{u} \sim \pi, \mathbf{v} \sim \sigma$ (dropped for readability). We follow an induction approach similar to the one presented in Cui et al. (2023). It trivially holds at $\tau = 0$. Let us assume that (B.20) holds at time step $\tau$. For subsequent $\tau + 1$,

$$\gamma^{\tau}\left\|\mathbb{E}\left[r(\mu_{\tau+1}, \nu_{\tau+1})|\mu_0 = \mu_t^{N_1}, \nu_0 = \nu_t^{N_2}, \pi_0 = \hat{\pi}_t^{N_1}, \sigma_0 = \hat{\sigma}_t^{N_2}\right]\right.$$
$$\left. - \mathbb{E}\left[r(\mu_{\tau+1}^{N_1}, \nu_{\tau+1}^{N_2})|\mu_0^{N_1} = \mu_t^{N_1}, \nu_0^{N_2} = \nu_t^{N_2}, \pi_0^{N_1} = \hat{\pi}_t^{N_1}, \sigma_0^{N_2} = \hat{\sigma}_t^{N_2}\right]\right\|$$
$$\leq \gamma^{\tau}\left\|\mathbb{E}\left[r\left(\mu_{\tau}F(\mu_{\tau}, \nu_{\tau}, \pi_{\tau}), \nu_{\tau}G(\mu_{\tau}, \nu_{\tau}, \sigma_{\tau})\right)|\mu_0 = \mu_t^{N_1}, \nu_0 = \nu_t^{N_2}, \pi_0 = \hat{\pi}_t^{N_1}, \sigma_0 = \hat{\sigma}_t^{N_2}\right]\right.$$
$$\left. - \mathbb{E}\left[r\left(\mu_{\tau}^{N_1}F(\mu_{\tau}^{N_1}, \nu_{\tau}^{N_2}, \phi_{\tau}), \nu_{\tau}^{N_2}G(\mu_{\tau}^{N_1}, \nu_{\tau}^{N_2}, \psi_{\tau})\right)|\mu_0^{N_1} = \mu_t^{N_1}, \nu_0^{N_2} = \nu_t^{N_2}, \pi_0^{N_1} = \hat{\pi}_t^{N_1}, \sigma_0^{N_2} = \hat{\sigma}_t^{N_2}\right]\right\|$$
$$+ \gamma^{\tau}\left\|\mathbb{E}\left[r\left(\mu_{\tau}^{N_1}F(\mu_{\tau}^{N_1}, \nu_{\tau}^{N_2}, \phi_{\tau}), \nu_{\tau}^{N_2}G(\mu_{\tau}^{N_1}, \nu_{\tau}^{N_2}, \psi_{\tau})\right)|\mu_0^{N_1} = \mu_t^{N_1}, \nu_0^{N_2} = \nu_t^{N_2}, \pi_0^{N_1} = \hat{\pi}_t^{N_1}, \sigma_0^{N_2} = \hat{\sigma}_t^{N_2}\right]\right.$$
$$\left. - \mathbb{E}\left[r(\mu_{\tau+1}^{N_1}, \nu_{\tau+1}^{N_2})|\mu_0^{N_1} = \mu_t^{N_1}, \nu_0^{N_2} = \nu_t^{N_2}, \pi_0^{N_1} = \hat{\pi}_t^{N_1}, \sigma_0^{N_2} = \hat{\sigma}_t^{N_2}\right]\right\|$$

The convergence of the first term follows from the weak LLN Cui et al. (2023) and a similar argument as used for (B.19). By using the Lipshcitzness of the reward function (Assumption 1) and Corollary 1, the second term in the expression is bounded by $\mathcal{O}\left(\frac{1}{\sqrt{N}}\right)$, where $\underline{N} = \min(N_1, N_2)$ and goes to zero as $(N_1, N_2) \to \infty$. $\qquad\square$

It follows from the convergence of the two individual terms in B.15 that $\|\nabla_{\theta}J(\alpha_{\theta}) - \nabla_{\theta}\hat{J}^{N_1}(\alpha_{\theta})\|$ as $(N_1, N_2) \to \infty$ $\qquad\square$

**Theorem 3.** *Let $\mathcal{G}_1$ denote the finite-population game where the agents utilize the identical team policies $\phi_t^*$ and $\psi_t^*$ derived from MF-MAPPO trained on $\mathcal{G}_1$ and let the finite-population game $\mathcal{G}_2$ with the same state-action space, dynamics, and rewards, but with population sizes $\bar{N}_1$ and $\bar{N}_2$ such that $\bar{N}_1/\bar{N}_2 = N_1/N_2$ and $\min(\bar{N}_1, \bar{N}_2) \geq \min(N_1, N_2)$. Then, $(\phi_t^*, \psi_t^*)$ remain $\epsilon$-optimal for $\mathcal{G}_2$.*

*Sketch of Proof.* The key idea of the proof is to exploit the fact that the equivalent infinite-population games for $\mathcal{G}_1$ and $\mathcal{G}_2$ are the same as $N_1/N_2 = \bar{N}_1/\bar{N}_2$ (Guan et al., 2024a). We first compute the performance of $\phi^*$, the optimal identical team policy derived for the $\mathcal{G}_1$ using MF-MAPPO (finite-population training), when applied to the *infinite-population* coordinator setting. We then compare the performance of the aforementioned $\phi^*$ for $\mathcal{G}_2$ with the *infinite-population* coordinator. Then, equating the common equivalent coordinators for $\mathcal{G}_1$ and $\mathcal{G}_2$, the result follows. $\qquad\square$

*Proof.* Consider game $\mathcal{G}_1$. Let us restrict ourselves to the set of identical team policies $\Phi \subseteq \Phi^{N_1}$ and suppose that $\phi^*$ is the optimal identical Blue team policy obtained from the $(N_1, N_2)$ finite population game via MF-MAPPO. From Theorem 1, we already know that

$$\min_{\psi^{N_2} \in \Psi^{N_2}} J^{N,\phi^*,\psi^{N_2}}(\mathbf{x}^{N_1}, \mathbf{y}^{N_2}) \geq \underline{J}^{N*}(\mathbf{x}^{N_1}, \mathbf{y}^{N_2}) - \mathcal{O}\Big(\frac{1}{\sqrt{\underline{N}}}\Big).$$

Denote $J^{N,\phi^*,\psi^{N_2}*}(\mathbf{x}^{N_1}, \mathbf{y}^{N_2}) \triangleq \min_{\psi^{N_2} \in \Psi^{N_2}} J^{N,\phi^*,\psi^{N_2}}(\mathbf{x}^{N_1}, \mathbf{y}^{N_2})$, where $\psi^{N_2*}$ is the potentially non-identical policy that solves the optimization problem.

Now, for $\mathcal{G}_2$ we want to show that

$$\min_{\psi^{\bar{N}_2} \in \Psi^{\bar{N}_2}} J^{\bar{N},\phi^*,\psi^{\bar{N}_2}}(\mathbf{x}^{\bar{N}_1}, \mathbf{y}^{\bar{N}_2}) \geq \underline{J}^{\bar{N}*}(\mathbf{x}^{\bar{N}_1}, \mathbf{y}^{\bar{N}_2}) - \mathcal{O}\Big(\frac{1}{\sqrt{\underline{N}}}\Big), \tag{B.21}$$

where $\phi^*$ corresponds to the identical policy derived from $\mathcal{G}_1$.

We know that the $\frac{N_1}{N_2} = \frac{\bar{N}_1}{\bar{N}_2}$. Thus, the infinite-population coordinator games remain identical for both $\mathcal{G}_1$ and $\mathcal{G}_2$. We prove the theorem in the following two steps.

**Step 1.** We first show that for $\phi^*$ from $\mathcal{G}_1$ we have the following inequality for all joint states $\mathbf{x}^{N_1} \in \mathcal{X}^{N_1}$ and $\mathbf{y}^{N_2} \in \mathcal{Y}^{N_2}$,

$$J_\infty^{\phi^*,\tilde{\psi}_\infty^*}(\mu^{N_1}, \nu^{N_2}) \geq J^{N,\phi^*,\psi^{N_2}*}(\mathbf{x}^{N_1}, \mathbf{y}^{N_2}) - \mathcal{O}\Big(\frac{1}{\sqrt{\underline{N}}}\Big), \tag{B.22}$$

where $\mu^{N_1} = \mathrm{Emp}_\mu(\mathbf{x}^{N_1})$ and $\nu^{N_2} = \mathrm{Emp}_\nu(\mathbf{y}^{N_2})$ and $\tilde{\psi}_\infty^* \in \Psi$ is given by,

$$J_\infty^{\phi^*,\tilde{\psi}_\infty^*}(\mu_0, \nu_0) = \min_{\psi \in \Psi} J_\infty^{\phi^*,\psi}(\mu_0, \nu_0). \tag{B.23}$$

The above equation (B.23) determines the optimal Red team response in the *infinite-population* domain, when the Blue team plays $\phi^*$ derived from $\mathcal{G}_1$. We prove this through an inductive argument.

*Base case:* At the terminal timestep $T$, the two value functions are the same. Thus, we have, for all joint states $\mathbf{x}_T^{N_1} \in \mathbf{X}^{N_1}$ and $\mathbf{y}_T^{N_2} \in \mathbf{Y}^{N_2}$, that

$$J_{\infty,T}^{\phi^*,\tilde{\psi}_\infty^*}(\mu_T^{N_1}, \nu_T^{N_1}) = J_T^{N,\phi^*,\psi^{N_2}*}(\mathbf{x}_T^{N_1}, \mathbf{y}_T^{N_2}) = r_t(\mu_T^{N_1}, \nu_T^{N_1}).$$

*Inductive hypothesis:* Assume that, at time step $t + 1$, the following holds for all $\mathbf{x}_{t+1}^{N_1} \in \mathbf{X}^{N_1}$ and $\mathbf{y}_{t+1}^{N_2} \in \mathbf{Y}^{N_2}$,

$$J_{\infty,t+1}^{\phi^*,\tilde{\psi}_\infty^*}(\mu_{t+1}^{N_1}, \nu_{t+1}^{N_1}) \geq J_{t+1}^{N,\phi^*,\psi^{N_2}*}(\mathbf{x}_{t+1}^{N_1}, \mathbf{y}_{t+1}^{N_2}) - \mathcal{O}\Big(\frac{1}{\sqrt{\underline{N}}}\Big). \tag{B.24}$$

*Induction:* Consider arbitrary $\mathbf{x}_t^{N_1} \in \mathbf{X}^{N_1}$ and $\mathbf{y}_t^{N_2} \in \mathbf{Y}^{N_2}$.

For simplicity, we do not emphasize the correspondence between the joint states and the EDs for the rest of the proof, as it is clear from the context. For the identical team policies $(\phi_t^*, \psi_t) \in \Phi \times \Psi$, at time step $t$, denote

$$\mu_{t+1}^{\phi^*} \triangleq \mu_t^{N_1} F(\mu_t^{N_1}, \nu_t^{N_2}, \phi_t^*)$$
$$\nu_{t+1}^{\psi} \triangleq \nu_t^{N_2} G(\mu_t^{N_1}, \nu_t^{N_2}, \psi_t). \tag{B.25}$$

For notational simplicity, we drop the conditions $\mathbf{X}_t^{N_1} = \mathbf{x}_t^{N_1}$ and $\mathbf{Y}_t^{N_2} = \mathbf{y}_t^{N_2}$ in the following derivations. Then, we have

$$J_t^{N,\phi^*,\psi^{N_2*}}(\mathbf{x}_t^{N_1}, \mathbf{y}_t^{N_2})$$

$$= \min_{\psi_t^{N_2} \in \Psi_t^{N_2}} r_t(\mu_t^{N_1}, \nu_t^{N_2}) + \mathbb{E}_{\phi_t^*,\psi_t^{N_2}}\left[ J_{t+1}^{N,\phi^*,\psi^{N_2*}}(\mathbf{X}_{t+1}^{N_1}, \mathbf{Y}_{t+1}^{N_2}) \right]$$

$$\overset{(i)}{\leq} r_t(\mu_t^{N_1}, \nu_t^{N_2}) + \min_{\psi_t^{N_2} \in \Psi_t^{N_2}} \mathbb{E}_{\phi_t^*,\psi_t^{N_2}}\left[ J_{\infty,t+1}^{\phi^*,\tilde{\psi}_\infty^*}(\mathcal{M}_{t+1}^{N_1}, \mathcal{N}_{t+1}^{N_2}) \right] + \mathcal{O}\left(\frac{1}{\sqrt{\underline{N}}}\right)$$

$$\overset{(ii)}{\leq} r_t(\mu_t^{N_1}, \nu_t^{N_2}) + \min_{\psi_t \in \Psi_t} \mathbb{E}_{\phi_t^*,\psi_t}\left[ J_{\infty,t+1}^{\phi^*,\tilde{\psi}_\infty^*}(\mathcal{M}_{t+1}^{N_1}, \mathcal{N}_{t+1}^{N_2}) \right] + \mathcal{O}\left(\frac{1}{\sqrt{\underline{N}}}\right)$$

$$\overset{(iii)}{=} r_t(\mu_t^{N_1}, \nu_t^{N_2}) + \mathcal{O}\left(\frac{1}{\sqrt{\underline{N}}}\right) + \min_{\psi_t \in \Psi_t} \mathbb{E}_{\phi_t^*,\psi_t}\left[ J_{\infty,t+1}^{\phi^*,\tilde{\psi}_\infty^*}(\mathcal{M}_{t+1}^{N_1}, \mathcal{N}_{t+1}^{N_2}) - J_{\infty,t+1}^{\phi^*,\tilde{\psi}_\infty^*}(\mu_{t+1}^{\phi^*}, \nu_{t+1}^{\psi_t}) \right.$$

$$\left. + J_{\infty,t+1}^{\phi^*,\tilde{\psi}_\infty^*}(\mu_{t+1}^{\phi^*}, \nu_{t+1}^{\psi_t}) \right]$$

$$\overset{(iv)}{\leq} r_t(\mu_t^{N_1}, \nu_t^{N_2}) + \mathcal{O}\left(\frac{1}{\sqrt{\underline{N}}}\right) + \min_{\psi_t \in \Psi_t} \mathbb{E}_{\phi_t^*,\psi_t}\left[ L_{J,t+1} \mathrm{d}_{\mathrm{TV}}\left(\mathcal{M}_{t+1}^{N_1}, \mu_{t+1}^{\phi^*}\right) \right.$$

$$\left. + L_{J,t+1} \mathrm{d}_{\mathrm{TV}}\left(\mathcal{N}_{t+1}^{N_2}, \nu_{t+1}^{\psi_t}\right) + J_{\infty,t+1}^{\phi^*,\tilde{\psi}_\infty^*}(\mu_{t+1}^{\phi^*}, \nu_{t+1}^{\psi_t}) \right]$$

$$\overset{(v)}{=} r_t(\mu_t^{N_1}, \nu_t^{N_2}) + \mathcal{O}\left(\frac{1}{\sqrt{\underline{N}}}\right) + \min_{\psi_t \in \Psi_t} J_{\infty,t+1}^{\phi^*,\tilde{\psi}_\infty^*}(\mu_{t+1}^{\phi^*}, \nu_{t+1}^{\psi_t}) + L_{J,t+1}\mathbb{E}_{\psi_t}\left[ \mathrm{d}_{\mathrm{TV}}\left(\mathcal{N}_{t+1}^{N_2}, \nu_{t+1}^{\psi_t}\right) \right]$$

$$+ L_{J,t+1}\mathbb{E}_{\phi_t^*}\left[ \mathrm{d}_{\mathrm{TV}}\left(\mathcal{M}_{t+1}^{N_1}, \mu_{t+1}^{\phi^*}\right) \right]$$

$$\overset{(vi)}{\leq} r_t(\mu_t^{N_1}, \nu_t^{N_2}) + \min_{\psi_t \in \Psi_t} J_{\infty,t+1}^{\phi^*,\tilde{\psi}_\infty^*}(\mu_{t+1}^{\phi^*}, \nu_{t+1}^{\psi_t}) + \mathcal{O}\left(\frac{1}{\sqrt{\underline{N}}}\right)$$

$$\overset{(vii)}{=} r_t(\mu_t^{N_1}, \nu_t^{N_2}) + \min_{\nu_{t+1} \in \mathbb{R}_{\nu,t}(\mu_t^{N_1}, \nu_t^{N_2})} J_{\infty,t+1}^{\phi^*,\tilde{\psi}_\infty^*}(\mu_{t+1}^{\phi^*}, \nu_{t+1}^{\psi_t}) + \mathcal{O}\left(\frac{1}{\sqrt{\underline{N}}}\right)$$

$$\overset{(viii)}{=} J_{\infty,t}^{\phi^*,\tilde{\psi}_\infty^*}(\mu_t^{N_1}, \nu_t^{N_2}) + \mathcal{O}\left(\frac{1}{\sqrt{\underline{N}}}\right)$$

For inequality (i), we used the inductive hypothesis; for inequality (ii), we reduced the optimization domain of the Red team to the set of *identical* team policies; inequality (iv) is a result of the Lipschitz continuity of the value function (Guan et al., 2024a); for equality (v), we use the fact that the mean-fields $(\mu_{t+1}^{\phi^*}, \nu_{t+1}^{\psi_t})$ are induced deterministically from the distributions at time $t$; inequality (vi) holds as a consequence of Corollary 1; (vii) converts the optimization domain from the policy space $\psi_t \in \Psi_t$ to the corresponding reachable set $\mathbb{R}_{\nu,t}(\mu_t^{N_1}, \nu_t^{N_2}) \triangleq \{\nu_{t+1} | \exists \psi_t \in \Psi_t \text{ s.t. } \nu_{t+1} = \nu_t G_t(\mu_t, \nu_t, \psi_t)\}$ following (Guan et al., 2024a); (vii) follows from the definition of $J_{\infty,t}^{\phi^*,\tilde{\psi}_\infty^*}(\mu_t^{N_1}, \nu_t^{N_2})$. Thus, $J^{N,\phi^*,\psi^{N_2*}}(\mathbf{x}^{N_1}, \mathbf{y}^{N_2}) = J_0^{N,\phi^*,\psi^{N_2*}}(\mathbf{x}^{N_1}, \mathbf{y}^{N_2})$ and (B.22) follows.

**Step 2.** We show that for $\phi^*$ from $\mathcal{G}_1$ we have the following inequality for all joint states $\mathbf{x}^{\bar{N}_1} \in \mathcal{X}^{\bar{N}_1}$ and $\mathbf{y}^{\bar{N}_2} \in \mathcal{Y}^{\bar{N}_2}$ in $\mathcal{G}_2$,

$$\min_{\psi^{\bar{N}_2} \in \Psi^{\bar{N}_2}} J^{\bar{N},\phi^*,\psi^{\bar{N}_2}}(\mathbf{x}^{\bar{N}_1}, \mathbf{y}^{\bar{N}_2}) \geq J_\infty^{\phi^*,\tilde{\psi}_\infty^*}(\mu^{\bar{N}_1}, \nu^{\bar{N}_2}) - \mathcal{O}\left(\frac{1}{\sqrt{\underline{N}}}\right), \tag{B.26}$$

where $\mu^{\bar{N}_1} = \mathrm{Emp}_\mu(\mathbf{x}^{\bar{N}_1})$, $\nu^{\bar{N}_2} = \mathrm{Emp}_\nu(\mathbf{y}^{\bar{N}_2})$ and $\underline{N} = \min(\bar{N}_1, \bar{N}_2)$. The proof is constructed based on induction. Fix an arbitrary Red team policy $\psi^{\bar{N}_2} \in \Psi^{\bar{N}_2}$.

*Base case:* At the terminal timestep $T$, since there is no decision to be made, both value functions are equal to the terminal reward and are thus the same. Formally, for all $\mathbf{x}_T^{\bar{N}_1} \in \mathbf{X}^{\bar{N}_1}$ and $\mathbf{y}_T^{\bar{N}_2} \in \mathbf{Y}^{\bar{N}_2}$,

$$J_T^{\bar{N},\phi^*,\psi^{\bar{N}_2}}(\mathbf{x}_T^{\bar{N}_1}, \mathbf{y}_T^{\bar{N}_2}) = J_{\infty,T}^{\phi^*,\tilde{\psi}_\infty^*}(\mu_T^{N_1}, \nu_T^{N_1}) = r_t(\mu_T^{\bar{N}_1}, \nu_T^{\bar{N}_1}),$$

where $\mu_T^{\bar{N}_1} = \mathrm{Emp}_\mu(\mathbf{x}_T^{\bar{N}_1})$ and $\nu_T^{\bar{N}_2} = \mathrm{Emp}_\nu(\mathbf{y}_T^{\bar{N}_2})$. For simplicity, we do not emphasize the correspondence between the joint states and the EDs for the rest of the proof, as it is clear from the context.

*Inductive hypothesis:* Assume that at $t+1$, the following holds for all joint states $\mathbf{x}_{t+1}^{\bar{N}_1} \in \mathbf{X}^{\bar{N}_1}$ and $\mathbf{y}_{t+1}^{\bar{N}_2} \in \mathbf{Y}^{\bar{N}_2}$:

$$J_{t+1}^{\bar{N},\phi^*,\psi^{\bar{N}_2}}(\mathbf{x}_{t+1}^{\bar{N}_1}, \mathbf{y}_{t+1}^{N_2}) \geq J_{\infty,t+1}^{\phi^*,\tilde{\psi}_\infty^*}(\mu_{t+1}^{\bar{N}_1}, \nu_{t+1}^{\bar{N}_1}) - \mathcal{O}\left(\frac{1}{\sqrt{\underline{\bar{N}}}}\right). \tag{B.27}$$

*Induction:* At timestep $t$, consider an arbitrary pair of joint states $\mathbf{x}_t^{\bar{N}_1} \in \mathbf{X}^{\bar{N}_1}$ and $\mathbf{y}_t^{\bar{N}_2} \in \mathbf{Y}^{\bar{N}_2}$, and their corresponding EDs $\mu_t^{\bar{N}_1}$ and $\nu_t^{\bar{N}_2}$.

For the identical team policy $\phi^* \in \Phi$, denote $\mu_{t+1}^{\phi^*}$) from (B.25). Furthermore, from Theorem 1 in Guan et al. (2024a) there exists a $\nu_{\text{apprx},t+1}^{\psi_t^{\bar{N}_2}}$ for the Red team policy $\psi_t^{N_2}$ such that

$$\mathbb{E}_{\psi_t^{\bar{N}_2}}\left[d_{\text{TV}}\big(\mathcal{N}_{t+1}^{\bar{N}_2}, \nu_{\text{apprx},t+1}^{\psi_t^{\bar{N}_2}}\big)\big|\mathbf{X}_t^{\bar{N}_1} = \mathbf{x}_t^{\bar{N}_1}, \mathbf{Y}_t^{\bar{N}_2} = \mathbf{y}_t^{\bar{N}_2}\right] \leq \frac{|\mathcal{Y}|}{2}\sqrt{\frac{1}{\bar{N}_2}}. \tag{B.28}$$

Then, for all joint states $\mathbf{x}_t^{\bar{N}_1} \in \mathbf{X}^{\bar{N}_1}$ and $\mathbf{y}_t^{\bar{N}_2} \in \mathbf{Y}^{\bar{N}_2}$, we have

$$J_t^{\bar{N},\phi^*,\psi^{\bar{N}_2}}(\mathbf{x}_t^{\bar{N}_1}, \mathbf{y}_t^{\bar{N}_2})$$

$$= r_t(\mu_t^{\bar{N}_1}, \nu_t^{\bar{N}_2}) + \mathbb{E}_{\phi^*,\psi^{\bar{N}_2}}\left[J_{t+1}^{\bar{N},\phi^*,\psi^{\bar{N}_2}}(\mathbf{X}_{t+1}^{\bar{N}_1}, \mathbf{Y}_{t+1}^{\bar{N}_2})\right]$$

$$\overset{(i)}{\geq} r_t(\mu_t^{\bar{N}_1}, \nu_t^{\bar{N}_2}) + \mathbb{E}_{\phi^*,\psi^{\bar{N}_2}}\left[J_{\infty,t+1}^{\phi^*,\tilde{\psi}_\infty^*}(\mathcal{M}_{t+1}^{\bar{N}_1}, \mathcal{N}_{t+1}^{\bar{N}_2})\right] - \mathcal{O}\left(\frac{1}{\sqrt{\underline{\bar{N}}}}\right)$$

$$= r_t(\mu_t^{\bar{N}_1}, \nu_t^{\bar{N}_2}) - \mathcal{O}\left(\frac{1}{\sqrt{\underline{\bar{N}}}}\right) + \mathbb{E}_{\phi^*,\psi^{\bar{N}_2}}\Big[J_{\infty,t+1}^{\phi^*,\tilde{\psi}_\infty^*}(\mathcal{M}_{t+1}^{\bar{N}_1}, \mathcal{N}_{t+1}^{\bar{N}_2})$$

$$- J_{\infty,t+1}^{\phi^*,\tilde{\psi}_\infty^*}(\mu_{t+1}^{\phi^*}, \nu_{\text{apprx},t+1}^{\psi_t^{\bar{N}_2}}) + J_{\infty,t+1}^{\phi^*,\tilde{\psi}_\infty^*}(\mu_{t+1}^{\phi^*}, \nu_{\text{apprx},t+1}^{\psi_t^{\bar{N}_2}})\Big]$$

$$\overset{(ii)}{\geq} r_t(\mu_t^{\bar{N}_1}, \nu_t^{\bar{N}_2}) + J_{\infty,t+1}^{\phi^*,\tilde{\psi}_\infty^*}(\mu_{t+1}^{\phi^*}, \nu_{\text{apprx},t+1}^{\psi_t^{\bar{N}_2}}) - \mathcal{O}\left(\frac{1}{\sqrt{\underline{\bar{N}}}}\right) - L_{J,t+1}\underbrace{\mathbb{E}_{\phi^*}\left[d_{\text{TV}}\big(\mathcal{M}_{t+1}^{\bar{N}_1}, \mu_{t+1}^{\phi^*}\big)\right]}_{=\mathcal{O}\left(\frac{1}{\sqrt{\bar{N}_1}}\right) \text{ due to Corollary 1}}$$

$$- \underbrace{L_{J,t+1}\mathbb{E}_{\psi^{N_2}}\left[d_{\text{TV}}\big(\mathcal{N}_{t+1}^{N_2}, \nu_{\text{apprx},t+1}^{\psi_t^{\bar{N}_2}}\big)\right]}_{=\mathcal{O}\left(\frac{1}{\sqrt{\bar{N}_2}}\right) \text{ due to(B.28)}}$$

$$\overset{(iii)}{\geq} r_t(\mu_t^{\bar{N}_1}, \nu_t^{\bar{N}_2}) + J_{\infty,t+1}^{\phi^*,\tilde{\psi}_\infty^*}(\mu_{t+1}^{\phi^*}, \nu_{\text{apprx},t+1}^{\psi_t^{\bar{N}_2}}) - \mathcal{O}\left(\frac{1}{\sqrt{\underline{\bar{N}}}}\right)$$

$$\overset{(iv)}{\geq} r_t(\mu_t^{\bar{N}_1}, \nu_t^{\bar{N}_2}) + \min_{\nu_{t+1} \in \mathbb{R}_{\nu,t}(\mu_t^{\bar{N}_1}, \nu_t^{\bar{N}_2})} J_{\infty,t+1}^{\phi^*,\tilde{\psi}_\infty^*}(\mu_{t+1}^{\phi^*}, \nu_{t+1}) - \mathcal{O}\left(\frac{1}{\sqrt{\underline{\bar{N}}}}\right)$$

$$\overset{(v)}{=} J_{\infty,t}^{\phi^*,\tilde{\psi}_\infty^*}(\mu_t^{\bar{N}_1}, \nu_t^{\bar{N}_2}) - \mathcal{O}\left(\frac{1}{\sqrt{\underline{\bar{N}}}}\right).$$

For inequality (i), we used the inductive hypothesis; for inequality (ii), we utilized the Lipschitz continuity of the coordinator value function (Guan et al., 2024a) using mean-field distributions $(\mu_{t+1}^{\phi^*}, \nu_{\text{apprx},t+1}^{\psi_t^{\bar{N}_2}})$ from (B.25) and the the deterministic transitions from $t$ to $t+1$; inequality (iv) follows from the definition of the min operator; and equality (v) follows from the definition of $J_{\infty,t}^{\phi^*,\tilde{\psi}_\infty^*}(\mu_t^{N_1}, \nu_t^{N_2})$ which completes the induction.

Since the Red team policy $\psi^{\bar{N}_2} \in \Psi^{\bar{N}_2}$ is arbitrary, we have that, for all joint states $\mathbf{x}^{\bar{N}_1} \in \mathbf{X}^{\bar{N}_1}$ and $\mathbf{y}^{\bar{N}_2} \in \mathbf{Y}^{\bar{N}_2}$,

$$\min_{\psi^{\bar{N}_2} \in \Psi^{\bar{N}_2}} J^{\bar{N},\phi^*,\psi^{\bar{N}_2}}(\mathbf{x}^{\bar{N}_1}, \mathbf{y}^{\bar{N}_2}) = \min_{\psi^{\bar{N}_2} \in \Psi^{\bar{N}_2}} J_0^{\bar{N},\phi^*,\psi^{\bar{N}_2}}(\mathbf{x}^{\bar{N}_1}, \mathbf{y}^{\bar{N}_2})$$

$$\geq J_\infty^{\phi^*,\tilde{\psi}_\infty^*}(\mu^{\bar{N}_1}, \nu^{\bar{N}_2}) - \mathcal{O}\left(\frac{1}{\sqrt{\underline{\bar{N}}}}\right).$$

**Step 3.** Combining (B.22) from Step 1 with Theorem 1, we have for all joint states $\mathbf{x}^{N_1} \in \mathcal{X}^{N_1}$ and $\mathbf{y}^{N_2} \in \mathcal{Y}^{N_2}$,

$$J_\infty^{\phi^*, \tilde{\psi}_\infty^*}(\mu^{N_1}, \nu^{N_1}) \geq J^{N, \phi^*, \psi^{N_2*}}(\mathbf{x}^{N_1}, \mathbf{y}^{N_2}) - \mathcal{O}\left(\frac{1}{\sqrt{\underline{N}}}\right)$$

$$\geq \underline{J}^{N*}(\mathbf{x}^{N_1}, \mathbf{y}^{N_2}) - \mathcal{O}\left(\frac{1}{\sqrt{\underline{N}}}\right), \tag{B.29}$$

where $\mu^{N_1} = \mathrm{Emp}_\mu(\mathbf{x}^{N_1})$ and $\nu^{N_2} = \mathrm{Emp}_\nu(\mathbf{y}^{N_2})$.

Assuming that the initial distributions are the same for $\mathcal{G}_1$ and $\mathcal{G}_2$, i.e., $\mu^{N_1} = \mu^{\bar{N}_1}$ and $\nu^{N_2} = \nu^{\bar{N}_2}$ we have the following sequence of inequalities

$$\min_{\psi^{\bar{N}_2} \in \Psi^{\bar{N}_2}} J^{\bar{N}, \phi^*, \psi^{\bar{N}_2}}(\mathbf{x}^{\bar{N}_1}, \mathbf{y}^{\bar{N}_2}) \overset{(i)}{\geq} J_\infty^{\phi^*, \tilde{\psi}_\infty^*}(\mu^{\bar{N}_1}, \nu^{\bar{N}_2}) - \mathcal{O}\left(\frac{1}{\sqrt{\bar{\underline{N}}}}\right)$$

$$\overset{(ii)}{\geq} \underline{J}^{N*}(\mathbf{x}^{N_1}, \mathbf{y}^{N_2}) - \mathcal{O}\left(\frac{1}{\sqrt{\underline{N}}}\right) - \mathcal{O}\left(\frac{1}{\sqrt{\bar{\underline{N}}}}\right)$$

$$\overset{(iii)}{\geq} \underline{J}_\infty^{\phi_\infty^*, \psi_\infty^*}(\mu^{N_1}, \nu^{N_1}) - \mathcal{O}\left(\frac{1}{\sqrt{\underline{N}}}\right) - \mathcal{O}\left(\frac{1}{\sqrt{\bar{\underline{N}}}}\right)$$

$$\overset{(iv)}{\geq} \underline{J}^{\bar{N}*}(\mathbf{x}^{\bar{N}_1}, \mathbf{y}^{\bar{N}_2}) - \mathcal{O}\left(\frac{1}{\sqrt{\underline{N}}}\right),$$

where $J_\infty^{\phi_\infty^*, \psi_\infty^*}(\mu_0, \nu_0) = \max_{\phi \in \Phi} \min_{\psi \in \Psi} J_\infty^{\phi, \psi}(\mu_0, \nu_0)$. Note that (i) follows from Step 2 (B.26); (ii) is a consequence of (B.29) and inequalities (iii) and (iv) follow from Lemmas 5 (applied on game $\mathcal{G}_1$) and 6 (applied on game $\mathcal{G}_2$) in Guan et al. (2024b) respectively, combined with the fact that $\min(\bar{N}_1, \bar{N}_2) > \min(N_1, N_2)$, completing the proof. $\qquad\square$

**Proposition 2.** *If the log-probability of the Blue team policy is bounded such that for all $x \in \mathcal{X}, u \in \mathcal{U}, \mu \in \mathcal{P}(\mathcal{X}), \nu \in \mathcal{P}(\mathcal{Y})$, $\|\nabla_\eta \log \phi(u \mid x, \mu, \nu)\|_2 \leq L_\phi/2|\mathcal{U}|$, where the gradient is taken with respect to $\eta \in \mathcal{P}(\mathcal{X}) \times \mathcal{P}(\mathcal{Y}) \triangleq [\mu^\mathsf{T}, \nu^\mathsf{T}]^\mathsf{T}$ and $L_\phi > 0$, then $\phi(u \mid x, \mu, \nu)$ is Lipschitz continuous with Lipschitz constant $L_\phi$, i.e.,*

$$\sum_u |\phi_t(u|x, \hat{\mu}, \hat{\nu}) - \phi_t(u|x, \hat{\mu}', \hat{\nu}')| \leq L_\phi\left(\mathrm{d_{TV}}(\hat{\mu}, \hat{\mu}') + \mathrm{d_{TV}}(\hat{\nu}, \hat{\nu}')\right) \ \forall \ x \in \mathcal{X}. \tag{9}$$

*Proof.* We prove the proposition in two parts.

**Step 1.** We first show that $\|\nabla_\eta \log \phi(u \mid x, \mu, \nu)\|_2 \leq \frac{L_\phi}{2|\mathcal{U}|}$ implies

$$\sum_u |\log \phi_t(u|x, \hat{\mu}, \hat{\nu}) - \log \phi_t(u|x, \hat{\mu}', \hat{\nu}')| \leq L_\phi\left(\mathrm{d_{TV}}(\hat{\mu}, \hat{\mu}') + \mathrm{d_{TV}}(\hat{\nu}, \hat{\nu}')\right) \ \forall \ x \in \mathcal{X}.$$
$$\tag{B.30}$$

By the Definition of $\eta$, we can denote $\log \phi_t(u|x, \eta) \triangleq \log \phi_t(u|x, \hat{\mu}, \hat{\nu})$. By the Mean-Value Theorem and the bounded gradient from Step 1 we have for all $x \in \mathcal{X}, u \in \mathcal{U}, \eta \in \mathcal{P}(\mathcal{X}) \times \mathcal{P}(\mathcal{Y})$,

$$|\log \phi_t(u|x, \eta) - \log \phi_t(u|x, \eta')| \leq \sup_{\eta \in \mathcal{P}(\mathcal{X}) \times \mathcal{P}(\mathcal{Y})} \|\nabla_\eta \log \phi(u \mid x, \eta)\|_2 \|\eta - \eta'\|_2$$

$$\leq \sup_{\eta \in \mathcal{P}(\mathcal{X}) \times \mathcal{P}(\mathcal{Y})} \|\nabla_\eta \log \phi(u \mid x, \eta)\|_2 \|\eta - \eta'\|_1$$

$$\leq \frac{L_\phi}{2|\mathcal{U}|} \cdot 2\left(\mathrm{d_{TV}}(\hat{\mu}, \hat{\mu}') + \mathrm{d_{TV}}(\hat{\nu}, \hat{\nu}')\right)$$

$$= \frac{L_\phi}{|\mathcal{U}|}\left(\mathrm{d_{TV}}(\hat{\mu}, \hat{\mu}') + \mathrm{d_{TV}}(\hat{\nu}, \hat{\nu}')\right).$$

Summing over all actions for any given $x \in \mathcal{X}$,

$$\sum_u |\log \phi_t(u|x, \eta) - \log \phi_t(u|x, \eta')| \leq \sum_u \frac{L_\phi}{|\mathcal{U}|} \left( \mathrm{d}_{\mathrm{TV}}(\hat{\mu}, \hat{\mu}') + \mathrm{d}_{\mathrm{TV}}(\hat{\nu}, \hat{\nu}') \right)$$

$$\leq L_\phi \left( \mathrm{d}_{\mathrm{TV}}(\hat{\mu}, \hat{\mu}') + \mathrm{d}_{\mathrm{TV}}(\hat{\nu}, \hat{\nu}') \right) \sum_u \frac{1}{|\mathcal{U}|}$$

$$\leq L_\phi \left( \mathrm{d}_{\mathrm{TV}}(\hat{\mu}, \hat{\mu}') + \mathrm{d}_{\mathrm{TV}}(\hat{\nu}, \hat{\nu}') \right),$$

completing the proof of Part 1.

**Step 2.** Using Step 1, we show that (B.30) implies (9). Define $h(z) \triangleq e^z$ and values $a \triangleq \log \phi_t(u|x, \hat{\mu}', \hat{\nu}')$ and $b \triangleq \log \phi_t(u|x, \hat{\mu}, \hat{\nu})$. It follows from the Mean-Value Theorem that for all $x \in \mathcal{X}$ and $u \in \mathcal{U}$,

$$|h(b) - h(a)| = |h'(c)||b - a|, \ \min(a, b) \leq c \leq \max(a, b)$$

$$|\phi_t(u|x, \hat{\mu}, \hat{\nu}) - \phi_t(u|x, \hat{\mu}', \hat{\nu}')| = e^c |\log \phi_t(u|x, \hat{\mu}, \hat{\nu}) - \log \phi_t(u|x, \hat{\mu}', \hat{\nu}')|$$

$$\leq e^{\max(a, b)} |\log \phi_t(u|x, \hat{\mu}, \hat{\nu}) - \log \phi_t(u|x, \hat{\mu}', \hat{\nu}')|.$$

Note that $e^a = \phi_t(u|x, \hat{\mu}', \hat{\nu}') \leq 1$ and $e^b = \phi_t(u|x, \hat{\mu}, \hat{\nu}) \leq 1$. Consequently $e^{\max(a, b)} \leq 1$ and we have for all $x \in \mathcal{X}$,

$$|\phi_t(u|x, \hat{\mu}, \hat{\nu}) - \phi_t(u|x, \hat{\mu}', \hat{\nu}')| \leq |\log \phi_t(u|x, \hat{\mu}, \hat{\nu}) - \log \phi_t(u|x, \hat{\mu}', \hat{\nu}')|$$

$$\sum_u |\phi_t(u|x, \hat{\mu}, \hat{\nu}) - \phi_t(u|x, \hat{\mu}', \hat{\nu}')| \leq \sum_u |\log \phi_t(u|x, \hat{\mu}, \hat{\nu}) - \log \phi_t(u|x, \hat{\mu}', \hat{\nu}')|$$

$$\sum_u |\phi_t(u|x, \hat{\mu}, \hat{\nu}) - \phi_t(u|x, \hat{\mu}', \hat{\nu}')| \leq L_\phi \left( \mathrm{d}_{\mathrm{TV}}(\hat{\mu}, \hat{\mu}') + \mathrm{d}_{\mathrm{TV}}(\hat{\nu}, \hat{\nu}') \right),$$

where the last inequality follows from (B.30) $\qquad \square$

**Proposition 3.** *Consider a given $\epsilon$-accurate estimator, i.e., $\mathrm{d}_{\mathrm{TV}}(\hat{\nu}_{i,t}^{N_2}, \hat{\nu}_t^{N_2}) < \epsilon$, for all $i, t$, where $\hat{\nu}_t^{N_2}$ is the true opponent MF at time $t$ and $\epsilon > 0$. For the identical team-policy pair $(\phi_t^*, \psi_t^*)$ obtained via gradient-regularized MF-MAPPO and deployed using this estimator, the cumulative regret satisfies $\Delta J(\phi_t^*, \psi_t^*) \leq K\epsilon + \mathcal{O}(1/\sqrt{N})$ for some constant $K > 0$.*

*Sketch of Proof.* MFs induced by identical team policies in an infinite-population game closely approximate the EDs induced in the corresponding finite-population game. Combining this consequence with Lipschitz continuity in the dynamics, rewards (Assumption 1) and policies (Proposition 2), we prove the statement. $\qquad \square$

*Proof.* For the performance guarantee, from (10), we have,

$$\Delta J(\phi_t^*, \psi_t^*) = \mathbb{E}_{\phi^*, \psi^*} \left[ \left| \sum_{t=0}^T r_t(\mathcal{M}_t^{N_1}, \mathcal{N}_t^{N_2}) - \sum_{t=0}^T r_t(\hat{\mathcal{M}}_t^{N_1}, \hat{\mathcal{N}}_t^{N_2}) \right| \ \middle| \ \mathbf{x}_0^{N_1} = \hat{\mathbf{x}}_0^{N_1}, \mathbf{y}_0^{N_2} = \hat{\mathbf{y}}_0^{N_2} \right].$$

From Lipschitz rewards (Assumption 1), we have,

$$\Delta J(\phi_t^*, \psi_t^*; \Gamma)$$

$$= \mathbb{E}_{\phi^*, \psi^*} \left[ \left| \sum_{t=0}^T r_t(\mathcal{M}_t^{N_1}, \mathcal{N}_t^{N_2}) - \sum_{t=0}^T r_t(\hat{\mathcal{M}}_t^{N_1}, \hat{\mathcal{N}}_t^{N_2}) \right| \ \middle| \ \mathbf{x}_0^{N_1} = \hat{\mathbf{x}}_0^{N_1}, \mathbf{y}_0^{N_2} = \hat{\mathbf{y}}_0^{N_2} \right]$$

$$\leq \mathbb{E}_{\phi^*, \psi^*} \left[ \sum_{t=0}^T L_r \left( \mathrm{d}_{\mathrm{TV}}(\mathcal{M}_t^{N_1}, \hat{\mathcal{M}}_t^{N_1}) + \mathrm{d}_{\mathrm{TV}}(\mathcal{N}_t^{N_2}, \hat{\mathcal{N}}_t^{N_2}) \right) \ \middle| \ \mathbf{x}_0^{N_1} = \hat{\mathbf{x}}_0^{N_1}, \mathbf{y}_0^{N_2} = \hat{\mathbf{y}}_0^{N_2} \right]$$

$$\leq L_r \sum_{t=0}^T \mathbb{E}_{\phi^*, \psi^*} \left[ \mathrm{d}_{\mathrm{TV}}(\mathcal{M}_t^{N_1}, \hat{\mathcal{M}}_t^{N_1}) + \mathrm{d}_{\mathrm{TV}}(\mathcal{N}_t^{N_2}, \hat{\mathcal{N}}_t^{N_2}) \ \middle| \ \mathbf{x}_0^{N_1} = \hat{\mathbf{x}}_0^{N_1}, \mathbf{y}_0^{N_2} = \hat{\mathbf{y}}_0^{N_2} \right], \quad \text{(B.31)}$$

where the last inequality follows from the linearity of expectations.

**Step 1.** We first show the following for all time steps $t$,

$$\mathbb{E}_{\phi^*,\psi^*}\Big[\mathrm{d}_{\mathrm{TV}}\big(\mathcal{M}_{t+1}^{N_1},\hat{\mathcal{M}}_{t+1}^{N_1}\big) + \mathrm{d}_{\mathrm{TV}}\big(\mathcal{N}_{t+1}^{N_2},\hat{\mathcal{N}}_{t+1}^{N_2}\big)\Big|\mathbf{x}_t^{N_1},\hat{\mathbf{x}}_t^{N_1},\mathbf{y}_t^{N_2},\hat{\mathbf{y}}_t^{N_2}\Big]$$

$$\leq \kappa_1 + \kappa_2\Big(\mathrm{d}_{\mathrm{TV}}\big(\mathcal{M}_t^{N_1},\hat{\mathcal{M}}_t^{N_1}\big) + \mathrm{d}_{\mathrm{TV}}\big(\mathcal{N}_t^{N_2},\hat{\mathcal{N}}_t^{N_2}\big)\Big),$$

where $\kappa_1 \triangleq \frac{1}{2}(L_\phi + L_\psi)\epsilon + \mathcal{O}\big(\frac{1}{\sqrt{\underline{N}}}\big)$ and $\kappa_2 \triangleq (1 + L_f + \frac{1}{2}L_\phi + \frac{1}{2}L_\psi)$.

By the linearity property of expectations,

$$\mathbb{E}_{\phi^*,\psi^*}\Big[\mathrm{d}_{\mathrm{TV}}\big(\mathcal{M}_{t+1}^{N_1},\hat{\mathcal{M}}_{t+1}^{N_1}\big) + \mathrm{d}_{\mathrm{TV}}\big(\mathcal{N}_{t+1}^{N_2},\hat{\mathcal{N}}_{t+1}^{N_2}\big)\Big|\mathbf{x}_t^{N_1},\hat{\mathbf{x}}_t^{N_1},\mathbf{y}_t^{N_2},\hat{\mathbf{y}}_t^{N_2}\Big]$$

$$= \mathbb{E}_{\phi^*}\Big[\mathrm{d}_{\mathrm{TV}}\big(\mathcal{M}_{t+1}^{N_1},\hat{\mathcal{M}}_{t+1}^{N_1}\big)\Big|\mathbf{x}_t^{N_1},\hat{\mathbf{x}}_t^{N_1},\mathbf{y}_t^{N_2},\hat{\mathbf{y}}_t^{N_2}\Big] + \mathbb{E}_{\psi^*}\Big[\mathrm{d}_{\mathrm{TV}}\big(\mathcal{N}_{t+1}^{N_2},\hat{\mathcal{N}}_{t+1}^{N_2}\big)\Big|\mathbf{x}_t^{N_1},\hat{\mathbf{x}}_t^{N_1},\mathbf{y}_t^{N_2},\hat{\mathbf{y}}_t^{N_2}\Big]$$

We now bound the Blue team's mean-field, i.e., the first term on the RHS. The Red team can be bounded similarly (second term).

$$\mathbb{E}_{\phi^*}\Big[\mathrm{d}_{\mathrm{TV}}\big(\mathcal{M}_{t+1}^{N_1},\hat{\mathcal{M}}_{t+1}^{N_1}\big)\Big|\mathbf{x}_t^{N_1},\hat{\mathbf{x}}_t^{N_1},\mathbf{y}_t^{N_2},\hat{\mathbf{y}}_t^{N_2}\Big]$$

$$\leq \mathbb{E}_{\phi^*}\Big[\mathrm{d}_{\mathrm{TV}}\big(\mathcal{M}_{t+1}^{N_1},\mathcal{M}_{t+1}\big)\Big|\mathbf{x}_t^{N_1},\hat{\mathbf{x}}_t^{N_1},\mathbf{y}_t^{N_2},\hat{\mathbf{y}}_t^{N_2}\Big]$$

$$+ \mathbb{E}_{\phi^*}\Big[\mathrm{d}_{\mathrm{TV}}\big(\mathcal{M}_{t+1},\hat{\mathcal{M}}_{t+1}\big)\Big|\mathbf{x}_t^{N_1},\hat{\mathbf{x}}_t^{N_1},\mathbf{y}_t^{N_2},\hat{\mathbf{y}}_t^{N_2}\Big]$$

$$+ \mathbb{E}_{\phi^*}\Big[\mathrm{d}_{\mathrm{TV}}\big(\hat{\mathcal{M}}_{t+1}^{N_1},\hat{\mathcal{M}}_{t+1}\big)\Big|\mathbf{x}_t^{N_1},\hat{\mathbf{x}}_t^{N_1},\mathbf{y}_t^{N_2},\hat{\mathbf{y}}_t^{N_2}\Big], \tag{B.32}$$

where $\mathcal{M}_{t+1} = \mathcal{M}_t^{N_1}F(\mathcal{M}_t^{N_1},\mathcal{N}_t^{N_2},\phi^*)$ and $\hat{\mathcal{M}}_{t+1} = \hat{\mathcal{M}}_t^{N_1}F(\hat{\mathcal{M}}_t^{N_1},\hat{\mathcal{N}}_t^{N_2},\phi^*,\Gamma_{\text{D-PC}})$ are short-hand notations for the next induced mean-field from the *infinite*-population *deterministic* dynamics (Guan et al., 2024a) with and without the estimator $\Gamma_{\text{D-PC}}$. From Corollary 1 in Guan et al. (2024a), we have

$$\mathbb{E}_{\phi^*}\Big[\mathrm{d}_{\mathrm{TV}}\big(\mathcal{M}_{t+1}^{N_1},\mathcal{M}_{t+1}\big)\Big|\mathbf{x}_t^{N_1},\hat{\mathbf{x}}_t^{N_1},\mathbf{y}_t^{N_2},\hat{\mathbf{y}}_t^{N_2}\Big] \leq \mathcal{O}\big(\frac{1}{\sqrt{\underline{N}}}\big),$$

$$\mathbb{E}_{\phi^*}\Big[\mathrm{d}_{\mathrm{TV}}\big(\hat{\mathcal{M}}_{t+1}^{N_1},\hat{\mathcal{M}}_{t+1}\big)\Big|\mathbf{x}_t^{N_1},\hat{\mathbf{x}}_t^{N_1},\mathbf{y}_t^{N_2},\hat{\mathbf{y}}_t^{N_2}\Big] \leq \mathcal{O}\big(\frac{1}{\sqrt{\underline{N}}}\big).$$

Thus, we are left to simplify

$$\mathbb{E}_{\phi^*}\Big[\mathrm{d}_{\mathrm{TV}}\big(\mathcal{M}_{t+1},\hat{\mathcal{M}}_{t+1}\big)\Big|\mathbf{x}_t^{N_1},\hat{\mathbf{x}}_t^{N_1},\mathbf{y}_t^{N_2},\hat{\mathbf{y}}_t^{N_2}\Big]$$

$$= \mathrm{d}_{\mathrm{TV}}\big(\mathcal{M}_t^{N_1}F(\mathcal{M}_t^{N_1},\mathcal{N}_t^{N_2},\phi^*),\hat{\mathcal{M}}_t^{N_1}F(\hat{\mathcal{M}}_t^{N_1},\hat{\mathcal{N}}_t^{N_2},\phi^*,\Gamma_{\text{D-PC}})\big),$$

where the expectation vanishes due to *deterministic* transitions under identical policies $(\phi^*,\psi^*)$. Now,

$$2\mathrm{d}_{\mathrm{TV}}\big(\mathcal{M}_t^{N_1}F(\mathcal{M}_t^{N_1},\mathcal{N}_t^{N_2},\phi^*),\hat{\mathcal{M}}_t^{N_1}F(\hat{\mathcal{M}}_t^{N_1},\hat{\mathcal{N}}_t^{N_2},\phi^*,\Gamma_{\text{D-PC}})\big) \tag{B.33}$$

$$= \sum_{x'\in\mathcal{X}}\Big|\sum_{x\in\mathcal{X}}\sum_{u\in\mathcal{U}} f_t(x'|x,u,\mathcal{M}_t^{N_1},\mathcal{N}_t^{N_2})\phi_t^*(u|x,\mathcal{M}_t^{N_1},\mathcal{N}_t^{N_2})\mathcal{M}_t^{N_1}(x)$$

$$- \sum_{x\in\mathcal{X}}\sum_{u\in\mathcal{U}} f_t(x'|x,u,\hat{\mathcal{M}}_t^{N_1},\hat{\mathcal{N}}_t^{N_2})\phi_t^*(u|x,\hat{\mathcal{M}}_t^{N_1},\hat{\mathcal{N}}_{x,t}^{N_2})\hat{\mathcal{M}}_t^{N_1}(x)\Big|$$

$$\leq \sum_{x'\in\mathcal{X}}\sum_{x\in\mathcal{X}}\sum_{u\in\mathcal{U}}\Big|f_t(x'|x,u,\mathcal{M}_t^{N_1},\mathcal{N}_t^{N_2})\phi_t^*(u|x,\mathcal{M}_t^{N_1},\mathcal{N}_t^{N_2})\mathcal{M}_t^{N_1}(x)$$

$$- f_t(x'|x,u,\hat{\mathcal{M}}_t^{N_1},\hat{\mathcal{N}}_t^{N_2})\phi_t^*(u|x,\hat{\mathcal{M}}_t^{N_1},\hat{\mathcal{N}}_{x,t}^{N_2})\hat{\mathcal{M}}_t^{N_1}(x)\Big|.$$

Firstly, we add and subtract $f_t(x'|x,u,\hat{\mathcal{M}}_t^{N_1},\hat{\mathcal{N}}_t^{N_2})\phi_t^*(u|x,\hat{\mathcal{M}}_t^{N_1},\hat{\mathcal{N}}_t^{N_2})\hat{\mathcal{M}}_t^{N_1}(x)$ to split absolute value as

$$2\mathrm{d}_{\mathrm{TV}}\big(\mathcal{M}_t^{N_1}F(\mathcal{M}_t^{N_1},\mathcal{N}_t^{N_2},\phi^*),\hat{\mathcal{M}}_t^{N_1}F(\hat{\mathcal{M}}_t^{N_1},\hat{\mathcal{N}}_t^{N_2},\phi^*,\Gamma_{\text{D-PC}})\big) \leq \sum_{x'\in\mathcal{X}}\sum_{x\in\mathcal{X}}\sum_{u\in\mathcal{U}}(I_1 + I_2),$$

where,

$$I_1 \triangleq \left| f_t(x'|x,u,\mathcal{M}_t^{N_1},\mathcal{N}_t^{N_2})\phi_t^*(u|x,\mathcal{M}_t^{N_1},\mathcal{N}_t^{N_2})\mathcal{M}_t^{N_1}(x) - f_t(x'|x,u,\hat{\mathcal{M}}_t^{N_1},\hat{\mathcal{N}}_t^{N_2})\phi_t^*(u|x,\hat{\mathcal{M}}_t^{N_1},\hat{\mathcal{N}}_t^{N_2})\hat{\mathcal{M}}_t^{N_1}(x) \right|$$

$$I_2 \triangleq \left| f_t(x'|x,u,\hat{\mathcal{M}}_t^{N_1},\hat{\mathcal{N}}_t^{N_2})\phi_t^*(u|x,\hat{\mathcal{M}}_t^{N_1},\hat{\mathcal{N}}_t^{N_2})\hat{\mathcal{M}}_t^{N_1}(x) - f_t(x'|x,u,\hat{\mathcal{M}}_t^{N_1},\hat{\mathcal{N}}_t^{N_2})\phi_t^*(u|x,\hat{\mathcal{M}}_t^{N_1},\hat{\mathcal{N}}_{x,t}^{N_2})\hat{\mathcal{M}}_t^{N_1}(x) \right|.$$

Similar to (9), we can simplify $I_2$ using Proposition 2 and obtain

$$\sum_{x'\in\mathcal{X}}\sum_{x\in\mathcal{X}}\sum_{u\in\mathcal{U}} I_2 < L_\phi\epsilon.$$

We now individually bound the terms in $I_1$ using the definition of total variation distance, the identity

$$|abc - a'b'c'| \leq |a-a'|bc + a'|b-b'|c + a'b'|c-c'|, \ a,b,c \geq 0,$$

Assumption 1 and Proposition 2 as follows:

$$\sum_{x'\in\mathcal{X}}\sum_{x\in\mathcal{X}}\sum_{u\in\mathcal{U}} I_1 \leq 2\mathrm{d}_{\mathrm{TV}}\big(\mathcal{M}_t^{N_1},\hat{\mathcal{M}}_t^{N_1}\big) + (L_\phi + L_f)\left(\mathrm{d}_{\mathrm{TV}}\big(\mathcal{M}_t^{N_1},\hat{\mathcal{M}}_t^{N_1}\big) + \mathrm{d}_{\mathrm{TV}}\big(\mathcal{N}_t^{N_2},\hat{\mathcal{N}}_t^{N_2}\big)\right)$$

Thus we can substitute the expressions for $I_1$ and $I_2$ back in (B.33) and rewrite (B.32) to get

$$\mathbb{E}_{\phi^*}\left[\mathrm{d}_{\mathrm{TV}}\big(\mathcal{M}_{t+1}^{N_1},\hat{\mathcal{M}}_{t+1}^{N_1}\big)\Big|\mathbf{x}_t^{N_1},\hat{\mathbf{x}}_t^{N_1},\mathbf{y}_t^{N_2},\hat{\mathbf{y}}_t^{N_2}\right]$$

$$< \mathrm{d}_{\mathrm{TV}}\big(\mathcal{M}_t^{N_1},\hat{\mathcal{M}}_t^{N_1}\big)$$

$$+ \frac{1}{2}(L_\phi + L_f)\left(\mathrm{d}_{\mathrm{TV}}\big(\mathcal{M}_t^{N_1},\hat{\mathcal{M}}_t^{N_1}\big) + \mathrm{d}_{\mathrm{TV}}\big(\mathcal{N}_t^{N_2},\hat{\mathcal{N}}_t^{N_2}\big)\right) + \frac{1}{2}L_\phi\epsilon + \mathcal{O}\Big(\frac{1}{\sqrt{\underline{N}}}\Big) \qquad \text{(B.34)}$$

By symmetry,

$$\mathbb{E}_{\psi^*}\left[\mathrm{d}_{\mathrm{TV}}\big(\mathcal{N}_{t+1}^{N_2},\hat{\mathcal{N}}_{t+1}^{N_2}\big)\Big|\mathbf{x}_t^{N_1},\hat{\mathbf{x}}_t^{N_1},\mathbf{y}_t^{N_2},\hat{\mathbf{y}}_t^{N_2}\right]$$

$$< \mathrm{d}_{\mathrm{TV}}\big(\mathcal{N}_t^{N_2},\hat{\mathcal{N}}_t^{N_2}\big)$$

$$+ \frac{1}{2}(L_\psi + L_f)\left(\mathrm{d}_{\mathrm{TV}}\big(\mathcal{M}_t^{N_1},\hat{\mathcal{M}}_t^{N_1}\big) + \mathrm{d}_{\mathrm{TV}}\big(\mathcal{N}_t^{N_2},\hat{\mathcal{N}}_t^{N_2}\big)\right) + \frac{1}{2}L_\psi\epsilon + \mathcal{O}\Big(\frac{1}{\sqrt{\underline{N}}}\Big) \qquad \text{(B.35)}$$

Adding (B.34) and (B.35) we obtain,

$$\mathbb{E}_{\phi^*,\psi^*}\left[\mathrm{d}_{\mathrm{TV}}\big(\mathcal{M}_{t+1}^{N_1},\hat{\mathcal{M}}_{t+1}^{N_1}\big) + \mathrm{d}_{\mathrm{TV}}\big(\mathcal{N}_{t+1}^{N_2},\hat{\mathcal{N}}_{t+1}^{N_2}\big)\Big|\mathbf{x}_t^{N_1},\hat{\mathbf{x}}_t^{N_1},\mathbf{y}_t^{N_2},\hat{\mathbf{y}}_t^{N_2}\right]$$

$$< (1 + L_f + \frac{1}{2}L_\phi + \frac{1}{2}L_\psi)\left(\mathrm{d}_{\mathrm{TV}}\big(\mathcal{M}_t^{N_1},\hat{\mathcal{M}}_t^{N_1}\big) + \mathrm{d}_{\mathrm{TV}}\big(\mathcal{N}_t^{N_2},\hat{\mathcal{N}}_t^{N_2}\big)\right) + \frac{1}{2}(L_\phi + L_\psi)\epsilon + \mathcal{O}\Big(\frac{1}{\sqrt{\underline{N}}}\Big)$$

$$= \kappa_1 + \kappa_2\left(\mathrm{d}_{\mathrm{TV}}\big(\mathcal{M}_t^{N_1}(x),\hat{\mathcal{M}}_t^{N_1}(x)\big) + \mathrm{d}_{\mathrm{TV}}\big(\mathcal{N}_t^{N_2},\hat{\mathcal{N}}_t^{N_2}\big)\right).$$

**Step 2.** Define

$$a_t \triangleq \mathbb{E}_{\phi^*,\psi^*}\left[\mathrm{d}_{\mathrm{TV}}\big(\mathcal{M}_t^{N_1},\hat{\mathcal{M}}_t^{N_1}\big) + \mathrm{d}_{\mathrm{TV}}\big(\mathcal{N}_t^{N_2},\hat{\mathcal{N}}_t^{N_2}\big)\Big|\mathbf{x}_0^{N_1} = \hat{\mathbf{x}}_0^{N_1}, \mathbf{y}_0^{N_2} = \hat{\mathbf{y}}_0^{N_2}\right]$$

At $t = 0$, $\mathcal{M}_0^{N_1} = \hat{\mathcal{M}}_0^{N_1}$ and $\mathcal{N}_0^{N_2} = \hat{\mathcal{N}}_0^{N_2}$ as we begin the fully observable and estimated scenarios under the same initial joint states , i.e., $\mathbf{x}_0^{N_1} = \hat{\mathbf{x}}_0^{N_1}, \mathbf{y}_0^{N_2} = \hat{\mathbf{y}}_0^{N_2}$. Thus, $a_0 = 0$. We proceed to show that $a_{t+1} \leq \kappa_1 + \kappa_2 a_t$ where $\kappa_1$ and $\kappa_2$ are the same as Step 1. By the law of iterated expectations,

$$\mathbb{E}_{\phi^*,\psi^*}\left[\mathrm{d}_{\mathrm{TV}}\big(\mathcal{M}_{t+1}^{N_1},\hat{\mathcal{M}}_{t+1}^{N_1}\big) + \mathrm{d}_{\mathrm{TV}}\big(\mathcal{N}_{t+1}^{N_2},\hat{\mathcal{N}}_{t+1}^{N_2}\big)\Big|\mathbf{x}_0^{N_1} = \hat{\mathbf{x}}_0^{N_1}, \mathbf{y}_0^{N_2} = \hat{\mathbf{y}}_0^{N_2}\right]$$

$$= \mathbb{E}_{\phi^*,\psi^*}\left[\left[\mathbb{E}_{\phi^*,\psi^*}\mathrm{d}_{\mathrm{TV}}\big(\mathcal{M}_{t+1}^{N_1},\hat{\mathcal{M}}_{t+1}^{N_1}\big) + \mathrm{d}_{\mathrm{TV}}\big(\mathcal{N}_{t+1}^{N_2},\hat{\mathcal{N}}_{t+1}^{N_2}\big)\Big|\mathbf{x}_t^{N_1},\hat{\mathbf{x}}_t^{N_1},\mathbf{y}_t^{N_2},\hat{\mathbf{y}}_t^{N_2}\right]\Big|\mathbf{x}_0^{N_1} = \hat{\mathbf{x}}_0^{N_1}, \mathbf{y}_0^{N_2} = \hat{\mathbf{y}}_0^{N_2}\right]$$

$$< \mathbb{E}_{\phi^*,\psi^*}\left[\kappa_1 + \kappa_2\left(\mathrm{d}_{\mathrm{TV}}\big(\mathcal{M}_t^{N_1},\hat{\mathcal{M}}_t^{N_1}\big) + \mathrm{d}_{\mathrm{TV}}\big(\mathcal{N}_t^{N_2},\hat{\mathcal{N}}_t^{N_2}\big)\right)|\mathbf{x}_0^{N_1} = \hat{\mathbf{x}}_0^{N_1}, \mathbf{y}_0^{N_2} = \hat{\mathbf{y}}_0^{N_2}\right]$$

$$= \kappa_1 + \kappa_2\mathbb{E}_{\phi^*,\psi^*}\left[\mathrm{d}_{\mathrm{TV}}\big(\mathcal{M}_t^{N_1},\hat{\mathcal{M}}_t^{N_1}\big) + \mathrm{d}_{\mathrm{TV}}\big(\mathcal{N}_t^{N_2},\hat{\mathcal{N}}_t^{N_2}\big)|\mathbf{x}_0^{N_1} = \hat{\mathbf{x}}_0^{N_1}, \mathbf{y}_0^{N_2} = \hat{\mathbf{y}}_0^{N_2}\right]$$

$$= \kappa_1 + \kappa_2 a_t,$$

where we use the result obtained from Step 1 along with the linearity property of expectations. Using this new notation, (B.31) simplifies to

$$\Delta J(\phi_t^*, \psi_t^*) \leq L_r \sum_{t=0}^{T} a_t$$

We have $a_0 = 0$, $a_1 < \kappa_1$, $a_2 < \kappa_1 + \kappa_2 a_1$ or $a_2 < \kappa_1(\kappa_2 + 1)$, $a_3 < \kappa_2 a_2 + \kappa_1$ or $a_3 < \kappa_1(\kappa_2^2 + \kappa_2 + 1)$ and so on. This can be written compactly for all $t > 0$ as

$$a_t < \kappa_1 \sum_{\tau=0}^{t-1} \kappa_2^\tau = \kappa_1 \frac{\kappa_2^t - 1}{\kappa_2 - 1},$$

as $\kappa_2 = (1 + L_f + \frac{1}{2}L_\phi + \frac{1}{2}L_\psi) \neq 1$. This is a geometric sum and thus we have

$$\Delta J(\phi_t^*, \psi_t^*) \leq L_r \sum_{t=0}^{T} a_t$$

$$= L_r \sum_{t=1}^{T} a_t$$

$$< L_r \sum_{t=1}^{T} \kappa_1 \frac{\kappa_2^t - 1}{\kappa_2 - 1}$$

$$= \frac{L_r \kappa_1}{\kappa_2 - 1} \sum_{t=1}^{T} \kappa_2^t - 1$$

$$= \frac{L_r \kappa_1}{\kappa_2 - 1} \left( \kappa_2 \frac{\kappa_2^T - 1}{\kappa_2 - 1} - T \right)$$

$$= \underbrace{\frac{1}{2} \frac{L_r(L_\phi + L_\psi)}{\kappa_2 - 1} \left( \kappa_2 \frac{\kappa_2^T - 1}{\kappa_2 - 1} - T \right)}_{K} \epsilon + \mathcal{O}\Big(\frac{1}{\sqrt{\underline{N}}}\Big) \frac{L_r}{\kappa_2 - 1} \left( \kappa_2 \frac{\kappa_2^T - 1}{\kappa_2 - 1} - T \right)$$

$$= K\epsilon + \mathcal{O}\Big(\frac{1}{\sqrt{\underline{N}}}\Big).$$

$\square$

**Theorem 4.** *D-PC satisfies Proposition 3 with $\epsilon = \mathcal{O}\big(e^{-cR_{\text{com}}}\big)$ with $c > 0$.*

*Sketch of Proof.* The exponential convergence of D-PC at each time-step $t$ follows directly from Theorems 6-8 in Nedić & Liu (2016) as the number of communication rounds $R_{\text{com}} \to \infty$. Therefore, for finite communication rounds, there exist constants $c_x, c_y > 0$ and $\rho_x, \rho_y \in (0, 1)$ such that for any number of communication rounds $R_{\text{com}}$ and any underlying distributions $(\hat{\mu}_t^{N_1}, \hat{\nu}_t^{N_2})$ D-PC satisfies

$$\text{d}_{\text{TV}}\big(\hat{\nu}_{x,t}^{N_2}, \hat{\nu}_t^{N_2}\big) < c_x \rho_x^{R_{\text{com}}} \triangleq \epsilon_x, \quad \text{for all } x \in \mathcal{X}^o \text{ and } t$$

$$\text{d}_{\text{TV}}\big(\hat{\mu}_{y,t}^{N_1}, \hat{\mu}_t^{N_1}\big) < c_y \rho_y^{R_{\text{com}}} \triangleq \epsilon_y, \quad \text{for all } y \in \mathcal{Y}^o \text{ and } t. \qquad (\text{B.36})$$

We define $\epsilon = \max(\epsilon_x, \epsilon_y)$. $\square$

*Proof.* We begin by defining set regularity.

**Definition 6** (Set Regularity (Nedić & Liu, 2016)). *Let $Z \subseteq \mathbb{R}^n$ be a nonempty set. A (finite) collection of closed convex sets $\{Y_j\}_{j=1}^J \subseteq \mathbb{R}^n$ is regular (in Euclidian norm) with respect to the set $Z$, if there exists a constant $r \geq 1$ such that*

$$\inf_{y \in Y} \|z - y\|_2 \leq r \max_j \inf_{y \in Y_j} \|z - y\|_2, z \in Z,$$

*where $Y \triangleq \bigcap_{j=1,\ldots,J} Y_j$ is non-empty. We refer to the scalar $r$ as the regularity constant. When the preceding relation holds with $Z = \mathbb{R}^n$, we say that the sets $\{Y_j\}_{j=1}^J$ are uniformly regular.*

**Proposition 5.** *The constraint sets $\mathcal{R}(x)$ for all $x \in \mathcal{X}^o$ are uniformly regular with regularity constant $\kappa \geq 1$, i.e.,*

$$\inf_{\nu \in \mathcal{F}} \|\omega - \nu\|_2 \leq \kappa \max_{x \in \mathcal{X}} \inf_{\nu_x \in \mathcal{R}(x)} \|\omega - \nu_x\|_2, \omega \in \mathbb{R}^{|\mathcal{X}|}.$$

*Proof.* Refer to Proposition 1 from Nedić & Liu (2016). □

Following Theorem 8 from Nedić & Liu (2016), we have

$$\rho_x = \left(1 - \frac{\theta^2}{4|\mathcal{X}^o|\mathrm{diam}(\mathcal{G}_{\mathrm{Blue}}^{\mathrm{com}})(\kappa+1)^2}\right), \quad c_x = \frac{1}{2}\sqrt{|\mathcal{Y}|}\sum_{x \in \mathcal{X}^o}\|\hat{\nu}_x^{\tau=0} - \xi_x^{\tau=1}\|_2^2$$

$$\rho_y = \left(1 - \frac{\theta^2}{4|\mathcal{Y}^o|\mathrm{diam}(\mathcal{G}_{\mathrm{Red}}^{\mathrm{com}})(\kappa+1)^2}\right), \quad c_y = \frac{1}{2}\sqrt{|\mathcal{X}|}\sum_{x \in \mathcal{Y}^o}\|\hat{\mu}_y^{\tau=0} - \xi_y^{\tau=1}\|_2^2,$$

where $\kappa$ is the regularity constant as in Proposition 5 and $\theta$ follows from (A.3). □

## C   RPS AND cRPS SETUP

### C.1   STATE SPACE

We have three states in this representation of the game: rock, paper and scissors. We denote this state space as $\mathcal{S} = \{\mathrm{R}, \mathrm{P}, \mathrm{S}\}$. The empirical distribution of the Blue team is denoted by $\mu \in \mathcal{P}(\mathcal{S})$ and that of the Red team is denoted by $\nu \in \mathcal{P}(\mathcal{S})$. Since we have three states for each team, both EDs lie in a three-dimensional simplex denoted by $\mathcal{P}(\mathcal{S})$.

### C.2   ACTION SPACE

#### C.2.1   RPS

At each state, we define three actions denoted by $\mathcal{A} = \{\mathrm{CW}, \mathrm{CCW}, \mathrm{Stay}\}$. These actions represent the ability of the agents to move from one state to the other in the following fashion:

1. CW denotes a clockwise cyclic action from one state to the other, i.e., from $\mathrm{R} \to \mathrm{P}$, $\mathrm{P} \to \mathrm{S}$, $\mathrm{S} \to \mathrm{R}$.

2. CCW denotes a counterclockwise cyclic movement, i.e., from $\mathrm{R} \to \mathrm{S}$, $\mathrm{S} \to \mathrm{P}$, $\mathrm{P} \to \mathrm{R}$.

3. Stay denotes the idle action (remain in the same state as before).

#### C.2.2   cRPS

At each state we have two actions denoted by $\mathcal{A} = \{\mathrm{CW}, \mathrm{Stay}\}$. These actions represent the ability of the agents to move from one state to the other in the following fashion:

1. CW denotes a clockwise cyclic action from one state to the other, i.e., from $\mathrm{R} \to \mathrm{P}$, $\mathrm{P} \to \mathrm{S}$, $\mathrm{S} \to \mathrm{R}$.

2. Stay denotes the idle action (remain in the same state as before).

Thus, we cannot directly jump from R to S within a single step, but must go via P. Mathematically, S does not lie in the reachable set of R. The reachable set $\mathcal{R}(s)$ for each state $s$ at a given time step under this modified action space is as follows

$$\mathcal{R}(\mathrm{R}) = \{\mathrm{R}, \mathrm{P}\}, \quad \mathcal{R}(\mathrm{P}) = \{\mathrm{P}, \mathrm{S}\}, \quad \mathcal{R}(\mathrm{S}) = \{\mathrm{S}, \mathrm{R}\}.$$

The state-action space of both RPS and cRPS are presented in Figure C.1.

## C.3 Dynamics and Transition Probabilities

For both RPS and cRPS, we consider deterministic transitions $\mathcal{T}(s, a, s')$, which implies that given a state-action pair $(s, a)$, the agent reaches a unique next state $s'$ with certainty (no distribution over the reachable states). Thus, for state R and action Stay, the transition function $\mathcal{T} : \mathcal{S} \times \mathcal{A} \times \mathcal{S} \to \{0, 1\}$ is given as:

$$\mathcal{T}(\text{R}, \text{Stay}, \text{R}) = 1, \quad \mathcal{T}(\text{R}, \text{Stay}, \text{P}) = 0, \quad \mathcal{T}(\text{R}, \text{Stay}, \text{S}) = 0.$$

This implies that an agent in the state R upon taking the Stay action remains in state R. Similarly,

$$\mathcal{T}(\text{R}, \text{CW}, \text{R}) = 0, \quad \mathcal{T}(\text{R}, \text{CW}, \text{P}) = 1, \quad \mathcal{T}(\text{R}, \text{CW}, \text{S}) = 0.$$

represent the method to transition from R to P .

## C.4 Reward Structure

In a two player RPS game, the reward matrix for Player 1 is defined as:

$$A = \begin{bmatrix} 0 & -1 & 1 \\ 1 & 0 & -1 \\ -1 & 1 & 0 \end{bmatrix}.$$

We extend this two-player framework to the multi-agent team game formulation. Define the pairwise reward for an agent at state $x \in \mathcal{S}$ within the Blue team and at state $y \in \mathcal{S}$ from the Red team as

$$r(x, y) \triangleq A_{xy},$$

where $A_{xy}$ represents the element from the reward matrix $A$ corresponding to the states $x$ (row player) and $y$ (column player). In lieu of the zero-sum structure, the reward for the agent at $y$ with respect to $x$ becomes $-r(x, y)$. Thus, for each player $x_i \in \mathcal{S}$ and $i = 1, 2, \ldots, N_1$ in the Blue team and $y_j \in \mathcal{S}$ and $j = 1, 2, \ldots, N_2$ in the Red team, the reward for the Blue team can be defined as

$$R_{\text{Blue}}(\mathbf{x}, \mathbf{y}) = \frac{1}{N_1} \sum_{i=1}^{N_1} \underbrace{\left[ \frac{1}{N_2} \sum_{j=1}^{N_2} r(x_i, y_j) \right]}_{\text{reward for agent } i}.$$

Rewriting the term inside the square brackets as

$$\frac{1}{N_2} \sum_{j=1}^{N_2} r(x_i, y_j) = \frac{1}{N_2} \sum_{y \in \mathcal{S}} \sum_{j=1}^{N_2} r(x_i, y) \mathbf{1}_{y_j=y}$$

$$= A_{x_i s_0} \sum_{j=1}^{N_2} \frac{1}{N_2} \mathbf{1}_{y_j=s_0} + A_{x_i s_1} \sum_{j=1}^{N_2} \frac{1}{N_2} \mathbf{1}_{y_j=s_1}$$

$$+ A_{x_i s_2} \sum_{j=1}^{N_2} \frac{1}{N_2} \mathbf{1}_{y_j=s_2}$$

$$= A_{x_i s_0} \nu(s_0) + A_{x_i s_1} \nu(s_1) + A_{x_i s_2} \nu(s_2)$$

$$= A(x_i)\nu, \tag{C.37}$$

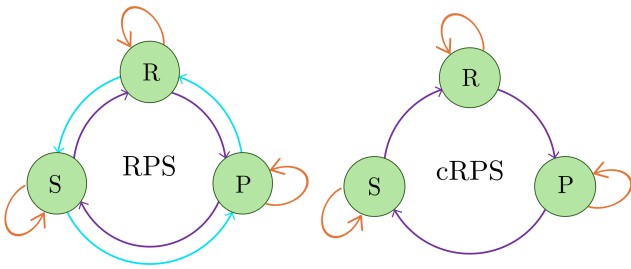

Figure C.1: State-action spaces for RPS and cRPS.

where $A(x_i)$ is the row of the reward matrix corresponding to state $x_i$. Using (C.37), the total Blue reward can be expressed as

$$R_{\text{Blue}}(\mathbf{x}, \mathbf{y}) = \frac{1}{N_1} \sum_{i=1}^{N_1} A(x_i) \nu$$

$$= \Big( \frac{1}{N_1} \sum_{x \in \mathcal{S}} \sum_{i=1}^{N_1} A(x_i) \mathbf{1}_{x_i = x} \Big) \nu$$

$$= \Big( A(s_0) \mu(s_0) + A(s_1) \mu(s_1) + A(s_2) \mu(s_2) \Big) \nu$$

$$= \mu^\mathsf{T} A \nu.$$

**Proposition 6.** *With initial conditions $\mu_{t=0} = [1, 0, 0]^\mathsf{T}$ and $\nu_{t=0} = [0, 1, 0]^\mathsf{T}$, all mean-field optimal trajectories satisfy $\mu_t^* = \nu_t^* = [\frac{1}{3}, \frac{1}{3}, \frac{1}{3}]^\mathsf{T}$ for all $t \geq 2$, and $\mu_1^* = [0, 1 - \eta, \eta]^\mathsf{T}$ where $\eta \in [\frac{1}{3}, \frac{2}{3}]$ and $\nu_1^* = [0, \frac{2}{3}, \frac{1}{3}]^\mathsf{T}$. Furthermore, the unique game value is given by $-\frac{1}{3}$.*

*Proof.* For the constrained RPS game under the stated initial condition, we cannot obtain the target distribution $[\frac{1}{3} \quad \frac{1}{3} \quad \frac{1}{3}]^\mathsf{T}$ after a single time step but this may be possible for $t \geq 2$. To this end, consider the following candidate trajectory respecting the transition dynamics:

$$\mu_0^\mathsf{T} = [1 \quad 0 \quad 0],$$
$$\mu_1^\mathsf{T} = [1 - x_1 \quad x_1 \quad 0],$$
$$\mu_2^\mathsf{T} = [1 - x_1 - x_2 \quad x_1 + x_2 - x_3 \quad x_3],$$
$$\mu_t^\mathsf{T} = [\tfrac{1}{3} \quad \tfrac{1}{3} \quad \tfrac{1}{3}], \quad \forall t > 2.$$
$$\nu_0^\mathsf{T} = [0 \quad 1 \quad 0],$$
$$\nu_1^\mathsf{T} = [0 \quad 1 - y_1 \quad y_1],$$
$$\nu_2^\mathsf{T} = [y_3 \quad 1 - y_1 - y_2 \quad y_1 + y_2 - y_3],$$
$$\nu_t^\mathsf{T} = [\tfrac{1}{3} \quad \tfrac{1}{3} \quad \tfrac{1}{3}], \quad \forall t > 2.$$

In order to respect the simplex structure for $\mu_t$ and $\nu_t$, we have the following constraints at all times:

$$0 \leq x_1, x_2, x_3 \leq 1, \quad x_2 \leq 1 - x_1, \quad x_3 \leq x_1.$$

Similarly,

$$0 \leq y_1, y_2, y_3 \leq 1, \quad y_2 \leq 1 - y_1, \quad y_3 \leq y_1.$$

For the distribution at $t = 2$ to be $[\frac{1}{3} \quad \frac{1}{3} \quad \frac{1}{3}]^\mathsf{T}$ for both teams, we get the additional constraints

$$x_3 = y_3 = \frac{1}{3},$$

$$x_1 + x_2 - x_3 = y_1 + y_2 - y_3 = \frac{1}{3},$$

$$\Rightarrow x_1 + x_2 = y_1 + y_2 = \frac{2}{3},$$

which implies that

$$x_1, y_1 \leq \frac{2}{3},$$

since $x_2, y_2 \geq 0$. The constraints now take the form

$$\frac{1}{3} \leq x_1 \leq \frac{2}{3}, \tag{C.38}$$

and similarly,

$$\frac{1}{3} \leq y_1 \leq \frac{2}{3}. \tag{C.39}$$

The objective function for cRPS is given by

$$J^{N,\phi,\psi}(\mu_0, \nu_0) = \mathbb{E}_{\phi,\psi}\Big[\sum_{t=1}^{T}\mu_t^\mathsf{T} A \nu_t \Big| \mu_0, \nu_0\Big], \tag{C.40}$$

which leads to the optimization problem

$$\max_{\phi^t}\min_{\psi^t}\quad J^{N_1,N_2,\phi^t,\psi^t}(\mu_0, \nu_0) = \mu_1^\mathsf{T} A \nu_1 \ . \tag{C.41}$$

Substituting $\mu_1^\mathsf{T} = [1 - x_1, x_1, 0]$ and $\nu_1^\mathsf{T} = [0, 1 - y_1, y_1]$ results in the following expression for the maximizing Blue team:

$$\max_{\phi^t}\quad J^{N_1,N_2,\phi^t,\psi^t}(\mu_0, \nu_0) = x_1 + 2y_1 - 3x_1 y_1 - 1$$
$$= x_1(1 - 3y_1) + (2y_1 - 1).$$

Since this equation is linear in $x_1$, the solution to the maximization problem subject to the constraint (C.38) is

$$x_1 = \frac{1}{3}, \quad y_1 > \frac{1}{3}, \tag{C.42}$$

$$x_1 = \frac{2}{3}, \quad y_1 < \frac{1}{3}, \tag{C.43}$$

$$x_1 \in [\frac{1}{3}, \frac{2}{3}], \quad y_1 = \frac{1}{3}. \tag{C.44}$$

Following the same approach for the minimizing Red team, we get the following objective,

$$\min_{\psi^t}\quad J^{N_1,N_2,\phi^t,\psi^t}(\mu_0, \nu_0) = x_1 + 2y_1 - 3x_1 y_1 - 1$$
$$= y_1(2 - 3x_1) + (x_1 - 1),$$

subject to the constraint (C.39), with the solution being:

$$y_1 = \frac{1}{3}, \quad x_1 < \frac{2}{3}, \tag{C.45}$$

$$y_1 = \frac{2}{3}, \quad x_1 > \frac{2}{3}, \tag{C.46}$$

$$y_1 \in [\frac{1}{3}, \frac{2}{3}], \quad x_1 = \frac{2}{3}. \tag{C.47}$$

Constraint (C.38) ensures that (C.46) cannot hold, while constraint (C.39) similarly prevents (C.43) from holding.

Consider now the case when $y_1 > \frac{1}{3}$. From (C.42) it follows that $x_1 = \frac{1}{3}$. Conversely, if the Blue team commits to a distribution with $x_1 = \frac{1}{3}$, the Red team's best response given by (C.45) gives $y_1 = \frac{1}{3}$, resulting in an incentive for the Red team to deviate from $y_1 > \frac{1}{3}$. Thus, (C.42) does not constitute an optimal solution. Following a similar argument, it can be shown that (C.47) is not an optimal solution either, as illustrated below.

Assume that $x_1 = \frac{2}{3}$. From (C.47), $y_1 \in [\frac{1}{3}, \frac{2}{3}]$. Now, if the Red team announces that it will deploy the distribution $y_1 \in [\frac{1}{3}, \frac{2}{3}]$, the Blue team's response for $x_1$ follows from (C.42) and (C.44). We have already established that (C.42) is not an optimal solution. This implies that $x_1 \in [\frac{1}{3}, \frac{2}{3}]$ can be a possible response to the Red team. However it violates (C.47), where $x_1 = \frac{2}{3}$ follows from strict equality. Thus, (C.47) does not constitute an optimal solution as the Blue team has an incentive to deviate.

Now, suppose the Blue team announces a distribution where $x_1 \in [\frac{1}{3}, \frac{2}{3}]$. In this case, the Red team's optimal response, derived from (C.45) and (C.44), is $y_1 = \frac{1}{3}$. Conversely, if the Red team announces that its distribution will be $y_1 = \frac{1}{3}$, the Blue team will still follow $x_1 \in [\frac{1}{3}, \frac{2}{3}]$. Since neither team has

an incentive to deviate from these distributions, they form an optimal trajectory. Thus, the solution to the bilinear optimization problem for two-time step convergence takes the form:

$$\mu_1^* = \begin{bmatrix} 1 - x_1 \\ x_1 \\ 0 \end{bmatrix} \quad \text{and} \quad \nu_1^* = \begin{bmatrix} 0 \\ \frac{2}{3} \\ \frac{1}{3} \end{bmatrix}, \tag{C.48}$$

such that $x_1 \in [\frac{1}{3}, \frac{2}{3}]$, leading to a game value of $-\frac{1}{3}$. This establishes the distribution at $t = 1$ and confirms the existence of a two-time step optimal trajectory, thereby proving the first part of the proposition.

Now note the following:

1. The original objective function (C.40) can be expressed in a bilinear form (similar to the expressions for $\mu_0, \mu_1, \mu_2$ using $x_1, x_2, x_3$). This makes it concave in the first argument and convex in the second argument.

2. The mean-fields $\mu$ and $\nu$ lie on a simplex and are hence, compact and convex.

Thus, by the generalized version of von Neumann's minimax theorem Sorin (2002), we conclude that the game value is unique, proving the second part of the proposition [2]. □

### C.5 Implementation Details and Hyperparameters

The state distributions are represented as arrays that are concatenated together to form the global observation. This becomes the input to the critic network which consists of a single hidden layer of 64 neurons and two tanh activation functions. The output is a single value that is equal to the estimated value function. On the other hand, the actor-network consists of a single MLP layer of 64 neurons that is concatenated with the local agent observation. Additionally, the logits are converted to a probability distribution through a softmax layer. The dimension scales with $|\mathcal{A}|$. Both the actor and critic networks are initialized using orthogonal initialization (Huang et al., 2022).

The single-stage RPS game is trained for 5,000 time steps with the actor and critic learning rates set to 0.0005 and 0.001, respectively, which remain constant throughout training. The networks are updated using the ADAM optimizer Kingma & Ba (2017) every 50 time steps for 10 epochs and a PPO clip value of 0.1. The entropy is decayed from 0.01 to 0.001 geometrically. We use an episode length of 1 after which the rewards are bootstrapped.

Moreover, since we have a single "team" buffer and the input/output dimensions are small, we do not use a mini-batch based update. For cRPS we use an episode length of 10 after which the rewards are bootstrapped. cRPS is trained using 200,000 time steps (=20,000 episodes) and is updated every 100 time steps. The algorithm was trained on a single NVIDIA GeForce RTX 3070 GPU and the training times are given in Tables E.1 and 1.

## D  Battlefield Setup

### D.1 State and Action Space

We consider a large-scale two-team (Blue and Red) ZS-MFTG on an $n \times n$ grid world. The state of the $i^{th}$ Blue agent is defined as the pair $x_i = (p_i^x, s_i^x)$ where $p_i^x \in \mathcal{S}_{\text{position}}$ denotes the position of the agent in the grid world and $s_i^x \in \mathcal{S}_{\text{status}} = \{0, 1\}$ defines the status of the agent: 0 being inactive and 1 being active. Similarly, we define the state of the Red agent as $y_i = (p_i^y, s_i^y)$. The state spaces for the Blue and Red teams are denoted by $\mathcal{X} = \mathcal{Y} = \mathcal{S}_{\text{position}} \times \mathcal{S}_{\text{status}}$, respectively. The mean-fields of the Blue ($\mu$) and Red ($\nu$) teams are distributions over the joint position and status space, i.e., $\mu, \nu \in \mathcal{P}(\mathcal{S}_{\text{position}} \times \mathcal{S}_{\text{status}})$. The action spaces are given by $\mathcal{U} = \mathcal{V} = \{\text{Up}, \text{Down}, \text{Left}, \text{Right}, \text{Stay}\}$ for both teams, representing discrete movements in the grid world. The learned identical team

---

[2]Note: The optimal infinite horizon trajectory itself need not be unique (we have shown that $x_1$ can take a range of values).

policy assigns actions based on an agent's local position and status, as well as the observed mean-fields of both teams. In the following subsections, we elaborate on the weakly coupled transition dynamics and reward structure introduced in the game, followed by a detailed discussion of the training procedure and network architecture for MF-MAPPO in this example.

## D.2 Interaction Between Agents

The transitions between states for agents belonging to both teams are characterized by their dynamics. These dynamics are probabilistic and depend on interactions among agents and are weakly coupled through their mean-field distributions. The weak coupling dynamics is keeping in line with the assumption in Guan et al. (2024a).

An agent at a given grid cell can be deactivated by the opponent team with a nonzero probability if the empirical mean-field of the opponent team at the grid cell supersedes that of the agent's own team. Similarly, a deactivated agent can be revived if the empirical mean-field of the agent's team is greater than the opponent's. This is referred to as numerical advantage. The total transition probability from state $(p, s)$ to state $(p', s')$ by taking an action $a$ is given by

$$\mathbb{P}\big((p', s') \mid (p, s), a, \mu, \nu\big) = \mathbb{P}\big(p' \mid (p', s'), a\big)\, \mathbb{P}\big(s' \mid (p, s), \mu, \nu\big),$$

where the first term on the right-hand side corresponds to the deterministic position transition when the agent is active. The second term corresponding to the status transition is given by

$$\mathbb{P}\big(0 \mid (p, 1), \mu, \nu\big) = \text{clip}_{[0,1]}\big(\alpha_x(\nu(p) - \mu(p))\big),$$
$$\mathbb{P}\big(1 \mid (p, 1), \mu, \nu\big) = 1 - \mathbb{P}\big(0 \mid (p, 1), \mu, \nu\big),$$

and

$$\mathbb{P}\big(1 \mid (p, 0), \mu, \nu\big) = \text{clip}_{[0,1]}\big(\beta_x(\mu(p) - \nu(p))\big),$$
$$\mathbb{P}\big(0 \mid (p, 0), \mu, \nu\big) = 1 - \mathbb{P}\big(1 \mid (p, 0), \mu, \nu\big),$$

where $\nu(p) - \mu(p)$ is the Red team's numerical advantage over the Blue team at $p$ Similarly, the Blue team's numerical advantage over Red is given by $\mu(p) - \nu(p)$. $\alpha_x$ and $\beta_x$ are tuning parameters to control the Red team's deactivation power and Blue team's reactivation power respectively. The Red team, being the defending team, is given a slight advantage in terms of higher deactivation power. This enables the possibility of capturing Blue team agents. However, to avoid degeneracy, the Red team agents are not allowed to enter the target. For our experiments, we assume $\alpha_x = 15$, $\alpha_y = 5$, and $\beta_x = \beta_y = 0$.

## D.3 Reward Structure

The team rewards only depend on the mean-fields of the two teams. For the battlefield scenario, the Blue team agents receive a positive reward corresponding to the fraction of agents that reach the target alive. This is a one-time reward that depends on the change in the fraction of the population of the agents at the target, i.e., if $\mu_t|_{\text{target}} = \mu_{t+1}|_{\text{target}}$, then the team does not receive any positive reward. Each agent in the team receives an identical "team reward." The reward function is mathematically formulated as

$$R_{\text{Blue}, t+1}(\mu, \nu) = \kappa\, \Delta\mu_{t+1}|_{\text{target}},$$

where,

$$\Delta\mu_{t+1}|_{\text{target}} = \mu_{t+1}(p^x = \text{Target}, s^x = 1) - \mu_t(p^x = \text{Target}, s^x = 1).$$

We have chosen $\kappa = 100$ in our simulations (heavier emphasis on reaching the target). The Red team's reward is the negative of the Blue team since we have a zero-sum game. Each team aims to maximize its own expected reward.

## D.4 Implementation and Hyperparameters

The state distribution for a grid world of size $n \times n$ is represented as a three-dimensional array of size $(2, n, n)$ for each team. The first layer depicts the mean-field of the agents over an $n \times n$ grid that are

alive and active, while the second layer gives information about the team's deactivated population. Each team's distribution is then concatenated together to form the global observation. This becomes the common information that is the input to the critic network which in our case is of size $(4, n, n)$ as we have two teams. Both neural networks consist of two main parts: a convolutional block and a fully connected block.

For the critic, the first CNN layer is the input layer that takes the 4 channels and outputs 32 channels, with a kernel size of 3x3, stride of 1, and padding of 1. Followed by ReLU activation, we have a hidden layer that takes 32 channels and outputs 64 channels, with the same kernel size, stride, and padding. Lastly, after another ReLU activation, we have the output layer that takes 64 channels and outputs 64 channels, again with the same kernel size, stride, and padding. After another ReLU layer, the output of the CNN is passed through an MLP. Namely, a fully connected (dense) layer takes the flattened output of the convolutional block and reduces it to 128 units. Between the input and the output layers, we have a single tanh activation function.

On the other hand, the input to the actor-network is split into two CNN blocks: one to process the common information and one to process the local information. The local information channel, is an array of size $(1, n, n)$ that locates the position of the agent with value +1 if it is active and -1 if it has been deactivated. This local information is passed through a single CNN layer that outputs 16 channels with a kernel size of 3x3, stride of 1, and padding of 1 while the common information is passed through two such layers with the output of 32 channels. Both outputs are then followed by a ReLU activation function and the latent representation of the common information combined with the local agent observation is then passed through an MLP architecture.

A fully connected (dense) layer takes the flattened output of the convolutional block and reduces it to 512 units. We have a single hidden layer that reduces the dimension further to 128 and then the output logits. The layers are separated by the tanh activation functions. Finally, the logits are converted to a probability distribution through a softmax layer. Both the actor and critic networks are initialized using orthogonal initialization (Huang et al., 2022). The architectures of the shared-team actor and minimally-informed critic networks for this example are shown in Figures D.2 and D.3 respectively.

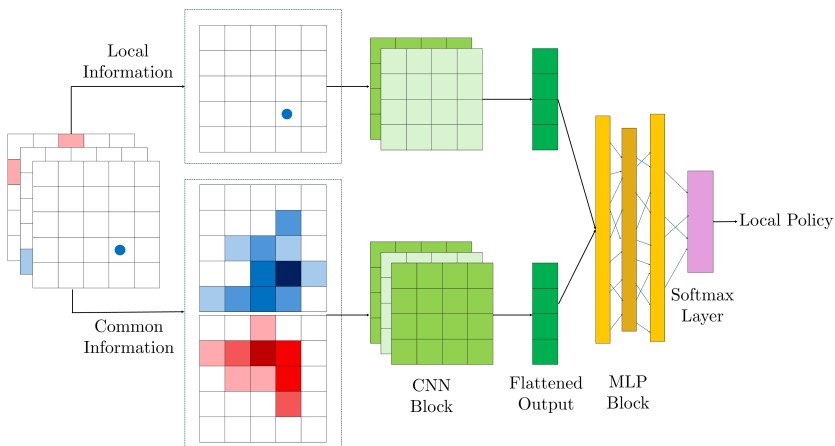

Figure D.2: MF-MAPPO: Shared-team actor for battlefield

All maps are trained using a single NVIDIA GeForce RTX 3070 GPU. The actor and critic learning rates are set to 0.0005 and 0.001 and both decay geometrically by a factor of 0.999. The networks are updated using the ADAM optimizer (Kingma & Ba, 2017) with two mini-batches for 10 epochs and a PPO clip value of 0.1. The entropy coefficient is initialized to 0.01 and decays with a factor of 0.995.

Maps 1 and 2 which are $4 \times 4$ grid worlds are trained for $5 \times 10^6$ and $4.5 \times 10^6$ time steps, respectively, and in both cases, the episode length is 20 time steps and the update frequency is every 500 time steps. The total training period is about one day. On the other hand, Map 3 being $8 \times 8$ in dimension,

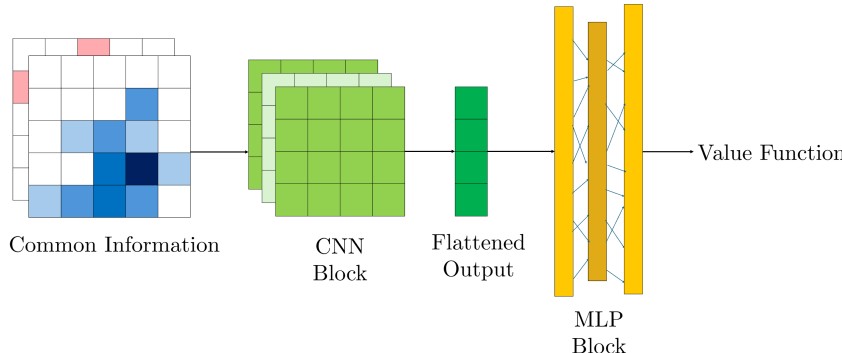

Figure D.3: MF-MAPPO: Minimally-informed critic for battlefield

has an episode length of 64, is trained for $9 \times 10^6$ time steps and its network is updated every 1,000 time steps. The total training period is approximately three days.

For evaluating D-PC, we employ MF-MAPPO to learn Lipschitz policies under a fully observable mean-field regime, and use these policies to evaluate the performance of our dynamic estimation algorithm, D-PC. We consider a time-invariant, edgeless visibility graph, wherein agents have access only to the mean-field distribution of their opponent corresponding to their current cell. For the communication graph, we assume a subgrid based topology that we detail in Section E.3. The communication graph is naturally time-varying since the number of states occupied by the agents need not be fixed. In accordance with Assumption 2 and the theoretical requirements of both D-PC and benchmark algorithms, we explicitly impose the constraint that the communication graph is connected at all time steps.

At each time step $t$, the initial Red team's estimate for the D-PC algorithm by the Blue team (i.e., at $\tau = 0$) for every state $x$ is taken as the projection of the estimate from the previous time step $t - 1$ onto the constraint set $\mathcal{R}(x)$, that is,

$$\hat{\nu}_{x,t}^{N_2,\tau=0} = \Omega_{\mathcal{R}(x)} \left[ \hat{\nu}_{x,t-1}^{N_2,R_{\mathrm{com}}} \right], \quad x \in \mathcal{X}_t^o.$$

At $t = 0$ and $\tau = 0$, we assume $\hat{\nu}_{x,t=0}^{N_2,\tau=0} \sim \mathrm{unif}\left(\mathcal{R}(x)\right)$ for all $x \in \mathcal{X}_t^o$.

## E  ADDITIONAL RESULTS

In this section, we present additional simulation results from various environments present on `MFEnv`. In particular, we look at the standard version of Rock-Paper-Scissors and more experiments on the zero-sum battlefield game - both trained using MF-MAPPO. We also present a single team grid world navigation task wherein agents learn a Lipschitz constrained MF-MAPPO policy and use D-PC to estimate their empirical distributions. This grid world navigation task in particular highlights the applicability of the presented algorithms to general mean-field team settings, not restricted to competitive zero-sum team games.

### E.1  ROCK-PAPER-SCISSORS (RPS)

We first extend the two-player Rock-Paper-Scissors (RPS) game to a game played between two populations as described in Appendix C. The Nash equilibrium for this population-based RPS game is the uniform population distribution $[1/3, 1/3, 1/3]$ over the 3 states (Raghavan, 1994).

We compare MF-MAPPO with DDPG-MFTG (Shao et al., 2024b) based on the training time, average test rewards and attainment of the computed Nash distributions for $N_1 = N_2 = 1,000$ agents. We exclude MADDPG (Lowe et al., 2017) from our comparison, as it scales poorly to hundreds or thousands of agents due to its reliance on all agents' local and global observations and actions as inputs to its critic networks. We include the training curve for cRPS in Figure E.5 for reference.

From the learning curves in Figures E.4 and E.5 one can see that the DDPG-MFTG algorithm failed to converge to the analytical game value of zero, while MF-MAPPO almost immediately attained the Nash game value. This corroborates with the results presented for cRPS (Table 1) in the main text. However, as shown in Table E.1, MF-MAPPO does take slightly longer to train since, unlike DDPG-MFTG, since MF-MAPPO avoids mini-batch training, following Yu et al. (2022).

We tested the learned policy with a fixed initial distribution $\mu_{t=0} = [1, 0, 0]^\mathsf{T}$ and $\nu_{t=0} = [0, 1, 0]^\mathsf{T}$, and the resulting trajectories are visualized in Figure E.6. All simulations were run for 150 instances. The trajectories of the Blue and Red team ED are depicted in cyan and pink, respectively, alongside the mean trajectory. The randomness in these trajectories arises from the finite-population approximation under a stochastic optimal policy, resulting in stochastic EDs. As shown in Figures E.6, DDPG-MFTG diverges from the equilibrium whereas MF-MAPPO converges immediately.

### E.2 BATTLEFIELD

#### E.2.1 VALIDATION CASES FOR MF-MAPPO

The following subsection qualitatively discusses the battlefield game for different map layouts. For these results, both teams are deploying policies trained using MF-MAPPO.

**Map A.** The first map is a simple $4 \times 4$ grid world with a single target that we use to validate our algorithm. The target is partially blocked by an obstacle, see Figure E.7. For the initial condition in Figure E.7, the Blue team is initially split into two equal groups. The Blue team decides to merge the two sub-groups of agents into a single group. With this formation, the Red team has zero numerical advantage over the Blue team when they encounter in (g), resulting in all Blue agents safely arriving at the target. In comparison, if the two Blue subgroups do not merge but move toward the target one at a time, it will lead to 50% of the Blue team population being deactivated (first subgroup), followed by the remaining 50% (second subgroup). This demonstrates how the observation of mean-field distributions guides rational decision-making.

**Map B.** This map is identical to the one presented in Section 5. In the first scenario (Figure E.8), 70% of the Blue agents start at cell [2, 2], and 30% at cell [1, 1], while the Red team is evenly split between cells [0, 1] and [3, 3]. Half of the Red team at [0, 1] successfully blocks the 30% Blue agent group from entering the left corridor due to its numerical advantage, which forces the Blue agents to opt for the right corridor. At the same time, the larger Blue group with 70% of the population utilized their numerical advantage over the half Red team at the top right and deactivated all the Red agents as shown in (c) and reached the target at time step (d). This allowed the smaller 30% group to follow through the same corridor without losing agents. In the second scenario (Figure E.9), with

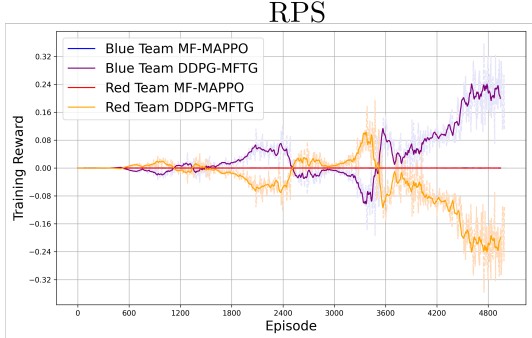

Figure E.4: Training curve for RPS.

Table E.1: Performance comparison for RPS

| Approach | Training Time | Average Reward | NE Attained? |
|----------|---------------|----------------|--------------|
| MF-MAPPO | 5min 17s | 0.0 | ✓ |
| DDPG-MFTG | 1min 34s | 0.334 | ✗ |

the Red team evenly distributed at the corners, 30% of the Blue agents start at cell [2, 2] and 70% at cell [3, 1]. The Red team's numerical advantage at [3, 3] forces the Blue agents to move around and regroup (Figures E.9(b)-E.9(f)). Once united, the Blue team's numerical advantage forces the Red subgroup at [0,1] to disperse to avoid deactivation, allowing Blue to reach the target.

**Map C.** In Figure E.10, we deploy the MF-MAPPO policies trained with $N_1 = N_2 = 100$ (from the main text) to larger populations of $\bar{N}_1 = \bar{N}_2 = 1500$ and $\bar{N}_1 = \bar{N}_2 = 1000$ using the same initial condition as Figure 4I and II respectively. The teams continue to achieve their objectives as established in Theorem 3, exhibiting behaviors similar to $N_1 = N_2 = 100$ setting. Section 5 presented scenarios featuring structured initial configurations for both teams over an $8 \times 8$ grid. It is important to emphasize, however, that the algorithm is trained on a diverse set of initial conditions for a given map, ranging from agents concentrated within a few selected cells to agents distributed randomly across the grid world. The following examples demonstrate that the teams are able to accomplish their objectives even in scenarios where agents are dispersed across the environment rather than clustered into just 1-2 subgroups using the same policy from Section 5. In Figure E.11, the Blue team is initialized randomly, and local subgroups of agents emerge and coordinate to reach the target. This behavior is particularly pronounced near the upper target, where a greater numerical advantage facilitates successful coalition formation (Figures E.11(c)–(e)).

Turning to the randomly distributed Red team agents in Figure E.12, it is observed that they concentrate near the two target entrances and successfully neutralize most Blue team subgroups.

### E.2.2   COMPARISON OF MF-MAPPO WITH BASELINE

While Table 2 reports the performance of the learned policies when evaluated against each other, Table E.2 provides a complementary analysis of the sample efficiency of the baselines in the Battlefield environment. We define sample efficiency as the reward obtained per interaction with the environment:

$$\text{Sample Efficiency} = \frac{\text{Total rewards}}{\text{Total environment steps}}.$$

In addition, we report the average episode length, which serves as an indicator of how quickly episodes terminate. Shorter episodes suggest that the Blue agents increasingly learn to reach the target, while the Red agents simultaneously learn to defend the target and deactivate Blue agents—either outcome leading to episode termination.

**Initial Condition 1.** Figure E.13I. compares the two algorithms against the baseline defending team. MF-MAPPO Blue agents exhibit coordinated maneuvering, forming coalitions to reach the target (a), whereas DDPG-MFTG Blue agents (b) show limited coordination, with only nearby agents reaching the target and distant agents failing to engage. In (c) we pit the Blue team against the MF-MAPPO defenders instead of the DDPG-MFTG defenders. The results align with those in Figure 3, where MF-MAPPO Blue agents effectively leverage their numerical advantage, enabling a larger

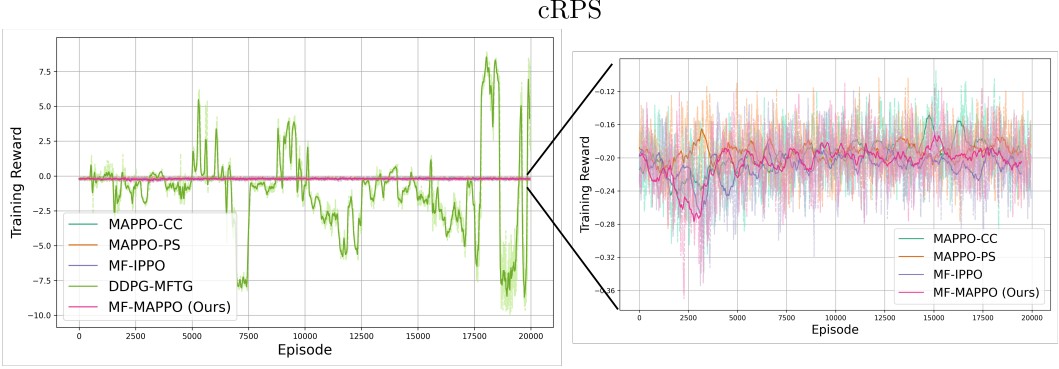

Figure E.5: Training curve for cRPS.

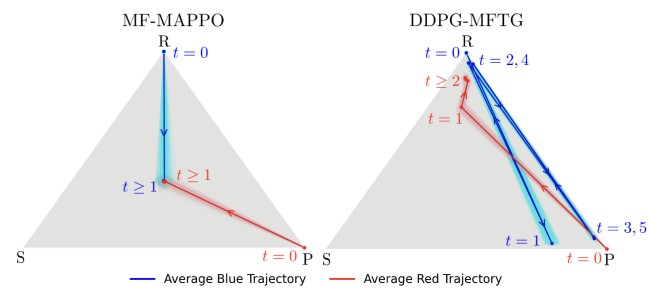

Figure E.6: ED trajectories induced by learned team policies on the state distribution simplex. Mean trajectories are averaged based on the 150 runs from fixed initialization $\mu_{t=0} = [1,0,0]^\mathsf{T}$ and $\nu_{t=0} = [0,1,0]^\mathsf{T}$; $N_1 = N_2 = 1,000$.

Figure E.7: Red is concentrated; Blue is evenly split.

Table E.2: Performance metrics for Battlefield

| Algorithm | Critic Input | Sample Efficiency | Avg. Episode Length |
|-----------|--------------|-------------------|---------------------|
| MF-IPPO | $\{(x_{i,t}^{N_1}, \mu_t)\}_{i=1}^{N_1}$ | 6.3392 | 9.258 |
| MAPPO-PS | $\{(x_{i,t}^{N_1}, \mu_t, \nu_t)\}_{i=1}^{N_1}$ | 5.4581 | 12.084 |
| MAPPO-CC | $(\mathbf{x}_t^{N_1}, \mathbf{y}_t^{N_2}, \mu_t, \nu_t)$ | 5.7878 | 9.8808 |
| DDPG-MFTG | $(\phi_t, \mu_t, \nu_t)$ | 0.0548 | 20.0 |
| MF-MAPPO | $(\mu_t, \nu_t)$ | **14.6238** | **4.8448** |

number of agents to reach the target. (d) represents MF-MAPPO vs. MF-MAPPO to illustrate the goal strategies and expected behavior.

**Initial Condition 2.** We present a different initial condition for the $4 \times 4$ Battlefield game (Figure E.13II.), where, using similar arguments as in the previous case and the main text, we can establish the superiority of MF-MAPPO agents over the baseline DDPG-MFTG, whether MF-MAPPO serves as the attacker or the defender.

### E.3 Mean-Field Estimation for Grid World Navigation Using D-PC

We consider a $9 \times 9$ grid with a target region at the center that is surrounded by penetrable obstacles (see Figure E.14). At a given cell, the population can penetrate an adjacent obstacle-state $x_{\text{obstacle}}$ with a probability proportional to its distribution at that cell, according to

$$f(x_{\text{obstacle}}|x, u, \mu_t) \propto \exp^{b(\mu_t(x)-c)}, \tag{E.49}$$

where $b > 0$ and $c \in (0,1)$. We run the algorithms for $T = 1,000$ time steps for varying values of $R_{\text{com}}$ under a subgrid-based communication topology. For each value of $R_{\text{com}}$, we evaluate our algorithm as well as the benchmark over multiple seeds of different initial mean-field distributions $\mu_{t=0}^N = \hat{\mu}_{t=0}^N$ and report the average estimation error over these seeds.

We use an edgeless visibility graph and a subgrid-based communication graph. In the subgrid communication topology, an agent located at state $x$ can exchange information with its neighboring states arranged in a $3 \times 3$ grid centered around $x$, as depicted in Figure E.14. Naturally, agents at the boundary have reduced connectivity: edge cells communicate within a $2 \times 3$ neighborhood, and corner cells within a $2 \times 2$ configuration, also shown in Figure E.14. We evaluate the estimation performance for communication rounds $R_{\text{com}} \in \{1,2,3,4,5,6,7,8\}$ and consider population

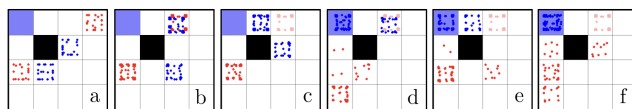

Figure E.8: Red is evenly split; 30% Blue are at $[1, 1]$ and 70% are at $[2, 2]$.

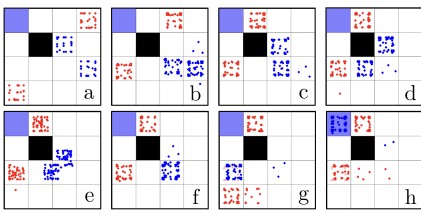

Figure E.9: Red is evenly split; 70% Blue are at $[3, 1]$ and 30% are at $[2, 2]$.

sizes of $N = 100$ and $N = 1,000$ (Recall, Theorem 3). From Figures E.15 and E.17 we see that up to approximately $R_{\mathrm{com}} = 4$, the proposed algorithm consistently outperforms the benchmark, particularly in the early phases where communication is limited.

The benchmark algorithm (Benjamin & Abate, 2025b) assumes that each state directly acquires the exact distributions of its communication neighbors $\mathcal{N}^{\mathrm{com}}(x)$. Consequently, it achieves better accuracy when $R_{\mathrm{com}}$ approaches $\mathrm{diam}(\mathcal{G}_t^{\mathrm{com}})$ communication rounds. While our method converges slightly more gradually, it offers notable practical advantages: it avoids relying on direct access to neighbor distributions—thereby preserving privacy—and performs competitively even under tighter communication constraints. Furthermore, for $R_{\mathrm{com}} < \mathrm{diam}(\mathcal{G}_t^{\mathrm{com}})$, the benchmark estimates unobserved states using a uniform distribution assumption—potentially leading to inaccurate representations of the actual mean-field. This inaccuracy is especially visible for small communication budgets ($R_{\mathrm{com}} = 1, 2, 3$). Although increasing rounds mitigates this error, in real-world scenarios where fast decision-making is critical such as in real-time opponent modeling, large-scale communication may be infeasible. In contrast, our algorithm remains tractable, communication-efficient, and robust across a range of practical deployment settings, offering a compelling trade-off between estimation quality and operational cost. The reward plots in Figures E.16 and E.18, corresponding to both population sizes, demonstrate that our method exhibits notably lower regret compared to the benchmark when evaluated against the fully observable policy. This improvement is particularly evident when $R_{\mathrm{com}} < \frac{1}{2}\mathrm{diam}(\mathcal{G}_t^{\mathrm{com}})$. For higher number of communication rounds our performance remains competitive with the benchmark. The plots also show that the estimation error incurred reduces with an increase in population size, predominantly noticeable at the peak around $t = 100$. Furthermore, the increase in population size from $N = 100$ to $N = 1,000$ results in reduced noise in the total variation errors and smoother error plots, thereby resulting in smaller variance. Both these results corroborate with the finite population mean-field approximation guarantees presented from Theorem 5. Empirically, we assert that our algorithm is particularly well-suited for scenarios with limited communication budgets, especially when $R_{\mathrm{com}} < \frac{1}{2}\mathrm{diam}(\mathcal{G}_t^{\mathrm{com}})$, where it consistently delivers competitive accuracy with substantially reduced overhead.

A direct consequence of Theorem 4 is that increasing the number of communication rounds $R_{\mathrm{com}}$ enables the estimation error $\epsilon$ to be made arbitrarily small. In finite populations, however, stochasticity in the MF approximation causes the full-information and estimated trajectories, $\mathcal{M}^{N_1}, \mathcal{N}^{N_2}$ and $\hat{\mathcal{M}}^{N_1}, \hat{\mathcal{N}}^{N_2}$, to vary across runs of GR-MF-MAPPO even under identical initial conditions. We confirm that D-PC achieves exponential convergence, which becomes most apparent in the infinite-population limit where state evolution is deterministic. Figure E.19 illustrates this behavior by plotting the estimation error at selected time steps against $R_{\mathrm{com}}$, consistent with Theorem 4.

### E.4 NON-LIPSCHITZ POLICIES AND ESTIMATION

While introducing partially observable ZS-MFTGs, we emphasized the importance of enforcing Lipschitz continuity of policies with respect to the mean-field distribution. In this section, we examine how varying the Lipschitz gradient penalty coefficient $\lambda$ influences the accuracy of the mean-field

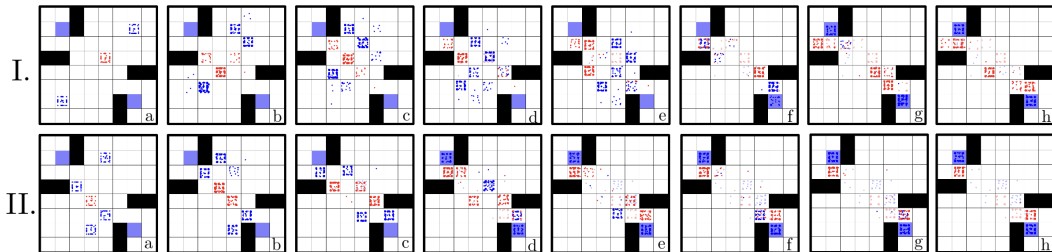

Figure E.10: I. 1500 agents in each team where Red is concentrated; 30% Blue are at the bottom, rest are at the top II. 1000 agents in each team with Blue evenly split; Red is concentrated.

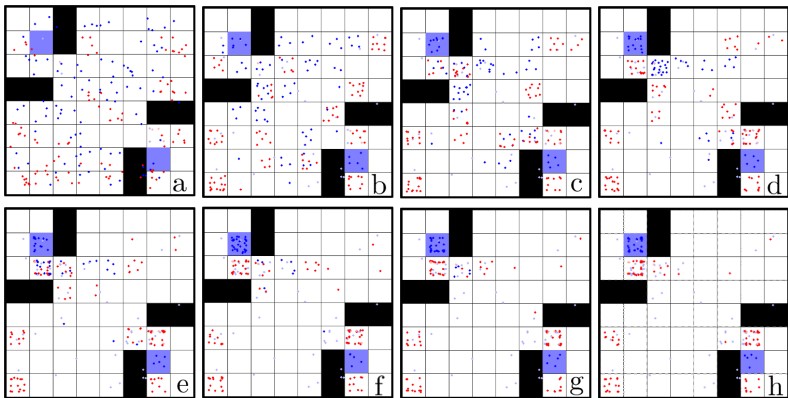

Figure E.11: Blue team is randomly spread around the map.

estimation procedure. Specifically, we introduce the following term in the objective (8):

$$\lambda \, \mathbb{E} \left[ \nabla_\eta \| \log \phi(u \mid x, \mu, \nu) \|^2 \right], \tag{E.50}$$

where $\lambda$ denotes the Lipschitz gradient penalty coefficient. Larger values of $\lambda$ enforce stronger smoothness constraints on the policy, promoting more regular behavior across variations in the mean-field input. In contrast, setting $\lambda = 0$ removes this constraint entirely. We conduct two sets of experiments in the finite-population grid world setting.

**Battlefield.** We utilize the same $4 \times 4$ battlefield introduced in Section 5 and consider the case wherein the Red team is estimating the Blue team's distribution and the Blue team has full information. We consider three distinct policies, each trained under a different value of $\lambda$, and allow the teams to estimate mean-field distributions that evolve accordingly. From Table E.3 we observe that for most cases, $\Delta J(\phi_t^*, \psi_t^*)$ (represented as % for normalized values) is higher for the policy trained without a Lipschitz constraint and the error reduces as $\lambda$ is increased, especially for larger values of $R_{\text{com}}$. This is because even small inaccuracies in the estimated distributions can lead to large discrepancies in the resulting mean-field, causing substantial deviations from the ideal fully observable behavior. Thus, enforcing Lipschitz continuity in policies offers tangible practical benefits by promoting smoother and more robust behaviors. Moreover, such policies tend to yield behaviors that are better aligned with the demands of real-world deployment, making them a compelling design choice when evaluating estimation performance.

**Navigation.** We consider the navigation problem defined above and now focus on the cumulative total variation error, i.e., $\sum_t d_{\text{TV}}(\mu_t, \hat{\mu}_t)$. Figure E.20 show the performance of our proposed D-PC algorithm and the benchmark method from Benjamin & Abate (2025b). In both estimation settings, once again, gradient-regularization yields trajectories that closely resemble the fully-observable scenario (lower errors).

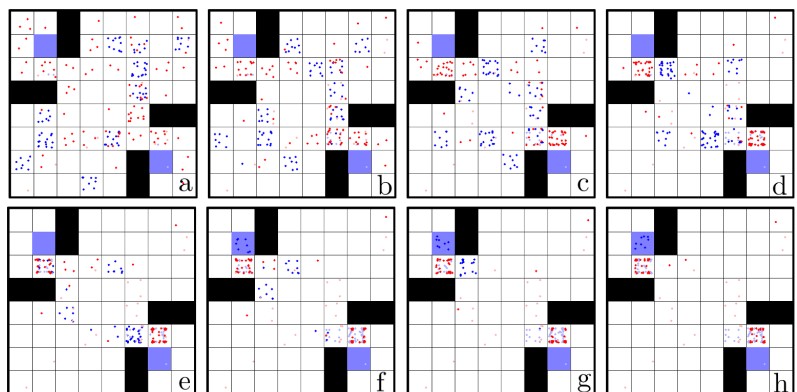

Figure E.12: Red team is randomly spread around the map.

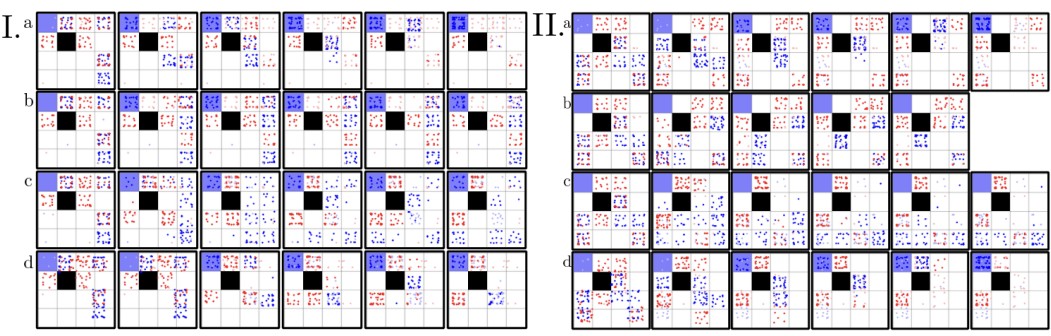

Figure E.13: a. MF-MAPPO Blue vs. DDPG-MFTG Red; b. DDPG-MFTG Blue vs. DDPG-MFTG Red; c. MF-MAPPO Red vs. DDPG-MFTG Blue; d. MF-MAPPO Red vs. MF-MAPPO Blue.

# F    ROLE OF LLMs

The authors acknowledge the use of GPT-4 and GPT-5 for polishing the main text and for quick access to existing results in probability theory, in particular, for Theorem 2 and Proposition 3.

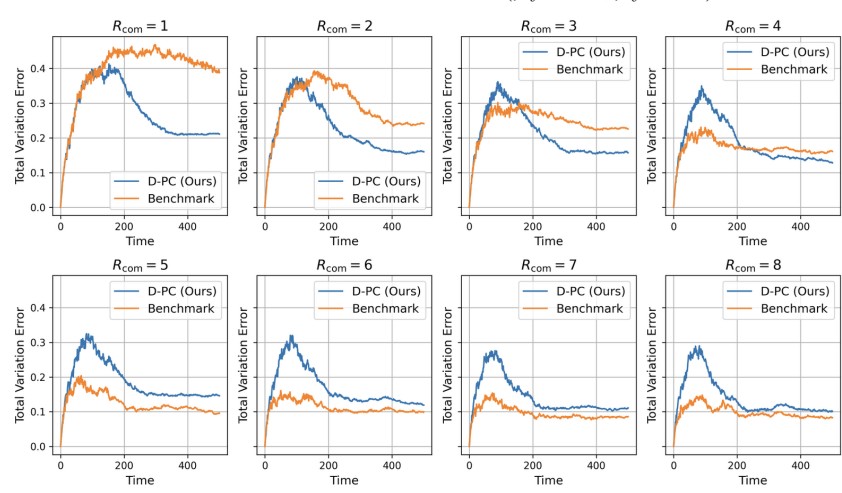

Figure E.14: Subgrid Communication Graph.

Total Variation Error vs. Time ($\mu_t^{N_1=100}, \hat{\mu}_t^{N_1=100}$)

Figure E.15: Total variation error at each time step for different values of $R_{\mathrm{com}}$ with $N = 100$.

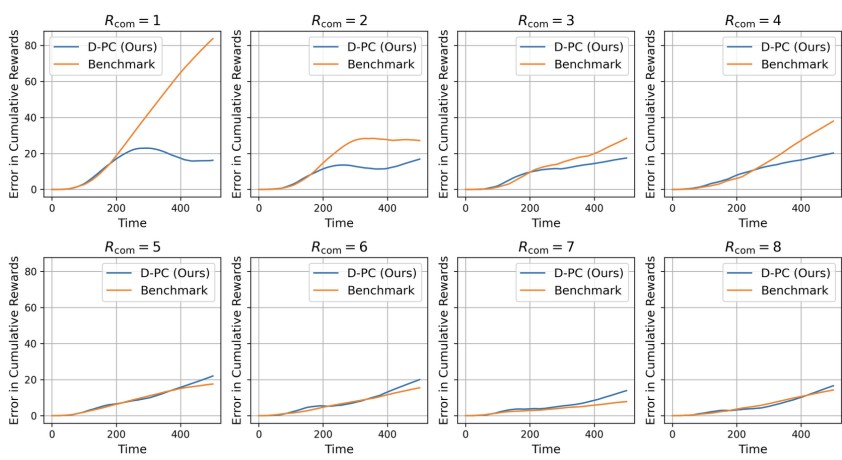

Figure E.16: Regret at each time step for different values of $R_{\mathrm{com}}$ with $N = 100$.

Table E.3: Effect of gradient-regularization on $\Delta J(\phi_t^*, \psi_t^*)$

| $\lambda$ | $R_{\mathrm{com}} = 5$ | $R_{\mathrm{com}} = 10$ | $R_{\mathrm{com}} = 15$ | $R_{\mathrm{com}} = 20$ |
|---|---|---|---|---|
| 0.0 | 0.39 | **0.20** | 1.15 | 2.10 |
| 0.005 | **0.09** | 0.65 | 0.47 | 0.27 |
| 0.01 | 0.49 | 0.26 | **0.38** | **0.03** |

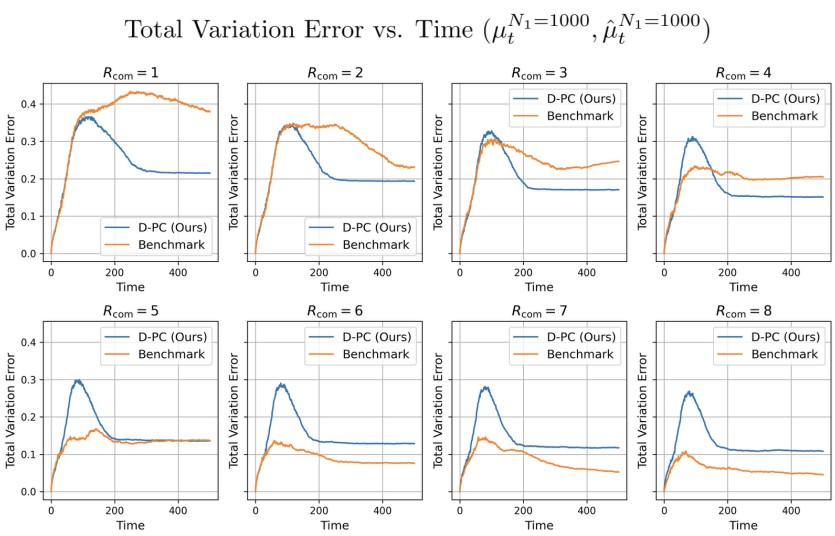

Figure E.17: Total variation error at each time step for different values of $R_{\text{com}}$ with $N = 1000$.

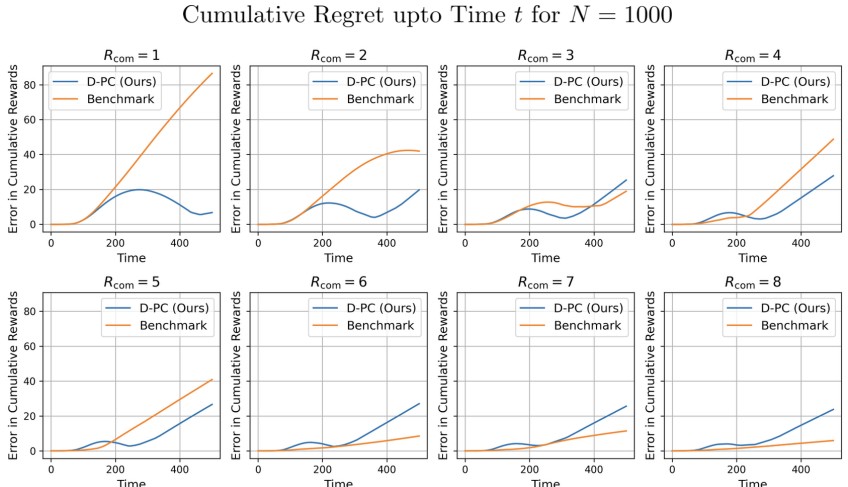

Figure E.18: Regret at each time step for different values of $R_{\text{com}}$ with $N = 1000$.

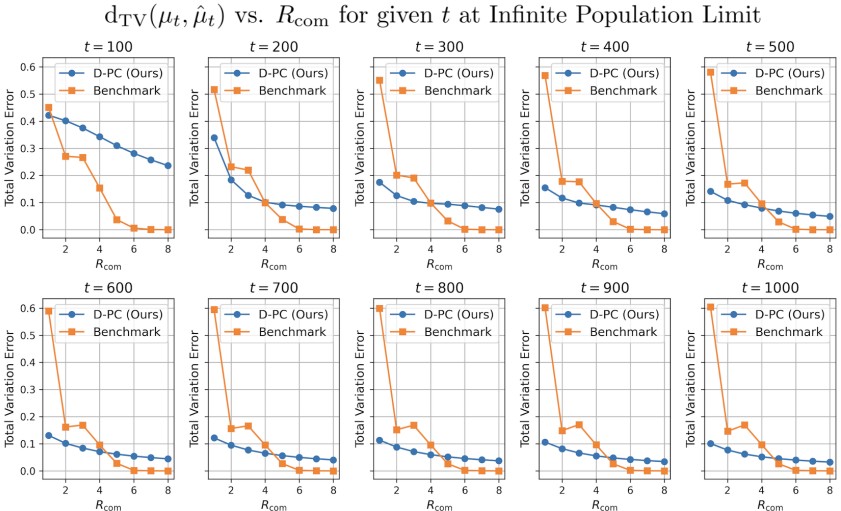

Figure E.19: Exponential convergence of D-PC at the infinite population limit, in the absence of stochasticity.

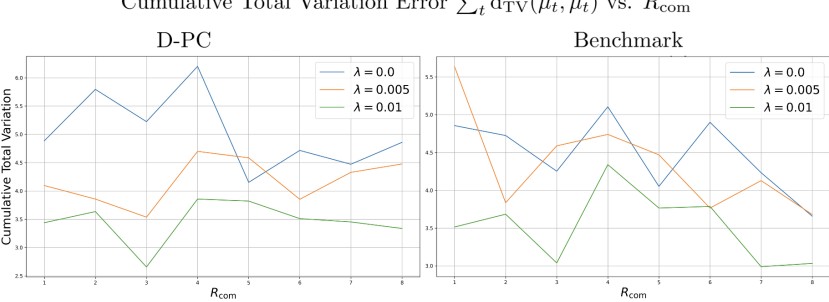

Figure E.20: Cumulative total variation error by various estimation algorithms in a $9 \times 9$ grid world for different values of $\lambda$ under a subgrid communication graph with $N = 1,000$.