# OpenReview forum: "Learning Large-Scale Competitive Team Behaviors with Mean-Field Interactions and Online Opponent Modeling"
_ICLR.cc/2026/Conference — Submitted to ICLR 2026_

### Official Review · Reviewer_B2tA · 2025-10-31

**Soundness:** 3
**Presentation:** 2
**Contribution:** 2
**Rating:** 4
**Confidence:** 4

**Summary:**

This paper proposes MF-MAPPO, a mean-field extension of PPO tailored to zero-sum mean-field team games that mix intra-team cooperation with inter-team competition, and trains it directly in finite-population simulators. The core ideas are a shared actor with a minimally informed, mean-field-only critic per team (so value estimation depends only on the two population distributions), theoretical guarantees linking finite-population training to the infinite-population limit (policy-gradient consistency and ε-optimality that improves with team size), and a gradient-regularized variant that stabilizes policies under partial observability. The authors also introduce D-PC, a decentralized opponent mean-field estimator with convergence and regret bounds, and release MFEnv, benchmark environments (constrained RPS and a grid-based battlefield) where MF-MAPPO consistently outperforms DDPG-MFTG and exhibits heterogeneous, coordinated behaviors at scale. The most significant novelty is the combination of (i) a PPO-style shared actor/minimal-critic architecture specialized to competitive mean-field team settings with (ii) theory that justifies training on finite populations and (iii) a provably correct opponent MF estimator that works under limited communication.

**Strengths:**

The paper studies a reasonable and timely setting: large-population competitive team games, approached via finite mean-field approximations.

The proposed algorithm feels natural given the theory (e.g., Theorems 2 and 3) and is a tidy fit to the mean-field structure.

The method combines an actor–critic algorithm tailored for mean-field games with a decentralized mean-field estimation framework (D-PC), which is an interesting and coherent design.

**Weaknesses:**

The paper would benefit from a stronger high-level overview that clearly explains the problem setup, why it matters, and how the pieces (finite-population training, minimally informed critic, D-PC) fit together; this would better convey the value of the work.

Experiments are confined to numerical/custom environments, and the set of baselines is limited; broader comparisons and ablations (including against adapted mainstream MARL baselines) are needed to support generality and impact claims.

The introduction of the partially observable setting needs clearer motivation and justification. While it is intended to model the lack of access to opponents’ mean-field statistics, the paper should provide a more explicit comparison to the fully observable case and a discussion of performance with and without accurate opponent mean-field estimation (e.g., sensitivity to estimation error and communication constraints).

**Questions:**

Could you provide a clear description of the problem setup and how the three components of finite-population training, minimally informed critic, and D-PC interlock?

What specific failure modes in standard MARL (e.g., leakage, non-stationarity, variance in value estimates) does each component address, and how did these considerations drive your architectural choices?

Beyond DDPG-MFTG, how do you perform against adapted mainstream MARL baselines (e.g., MAPPO, QMIX/VDN variants, IPPO, PPO-centralized-critic, FACMAC) when extended to your mean-field team setting? Please include learning curves, sample efficiency, and final returns.

How does performance degrade as the opponent mean-field estimate is corrupted by controlled noise, delay, or bandwidth limits?

---

> ### Author Response · Authors · 2025-11-20
> **Response to Reviewer B2tA**
>
> We would like to thank the reviewer for their overall supportive review, viewing MF-MAPPO as a “tidy fit” and D-PC as “interesting and coherent” to the overall problem. It seems that while you state our theoretical results are strong and our experiments effectively address the challenge of scalability in MARL, you would also like to see follow-up experiments.
>
> **Weakness 1/Questions 1 and 2 (clear description of the setup):**
> The ZS-MFTG problem aims at modeling large-population teams that compete while cooperating internally under weakly-coupled mean-field dynamics. Although the theoretical foundations are well established, computing policies at scale remain challenging. Below, we outline the primary failure modes and how our approach addresses them:
>
> (i) **Finite-population training**: Generally, in the mean-field setting (MFGs/MFCs/MFTGs), previous work [1-3] relies on infinite-population solutions that assume access to an oracle providing exact mean-field quantities. This is unrealistic as “sampling in an infinite population” is a purely theoretical construct. In practice, **only finite-population samples are available/observable**: local states $(\mathbf{x}^{N_1}_t, \mathbf{y}^{N_2}_t)$ with actions $(\mathbf{u}^{N_1}_t, \mathbf{v}^{N_2}_t)$. Our framework is explicitly designed to operate under this realistic constraint. Theorem 2 establishes that policy gradients computed via finite-population rollouts converge to their infinite-population counterparts as population size grows. This provides the theoretical justification for MF-MAPPO and ensures the practical relevance of our approach.
>
> (ii) **Minimally informed critic**: In an effort to make the computation of solutions **efficient** and achieve faster training, we exploit the mean-field information structure and utilize a minimally informed critic. This feature ensures that the critic is not overwhelmed by irrelevant information; this is particularly important in situations where agents want to protect their privacy by not sharing their private observations. As demonstrated in Tables 1 and E.2, this approach significantly enhances performance and distinguishes MF-MAPPO from standard parameter-sharing and MAPPO baselines.
>
> (iii) **D-PC**: Classical MFTG methods [4, 5] assume complete knowledge of team-level state distributions, which is unrealistic in **decentralized systems**. The D-PC mechanism relaxes this requirement, enabling agents to obtain consistent mean-field estimates without centralized information.
>
> Overall, our contributions relax several restrictive assumptions commonly seen in the mean-field literature and address the computational challenges, moving the theoretical framework closer to practical, real-world applicability.
>
> **Weakness 2/Question 3 (Baselines):**  [see common response #1]
> We have updated the paper to include the training curves for Battlefield in Figure 3 I. and for cRPS in the supplementary material (Figure E.5). As mentioned in the main text of the paper (lines 407-410) centralized Q-learning methods such as QMix, MADDPG, and FACMAC are not suitable as baselines in our setting because their centralized action-value functions depend on the joint action and full system state, making them poorly scalable in large-population regimes (even for the simple cRPS scenario, these algorithms run out of RTX 3070 GPU memory with merely 50-100 agents). Thus, we use MF-adaptations of MAPPO, IPPO, and PPO-centralized critics along with DDPG-MFTG. We see that MF-MAPPO is superior to these algorithms, thereby establishing its impact for solving ZS-MFTGs.
>
> **Weakness 3/Question 4 (D-PC under bandwidth limitations and noise):**
> The explicit comparison to the fully observable case under communication constraints requested by the reviewer has been conducted in Figure 4 IV. We have updated the main text to make the comparison more explicit. We have also further expanded the paper to include a comparison with the fully observable case under noisy estimates [see common response #3] based on the reviewer’s comment. We appreciate the feedback, which helped us further clarify the concept and highlight the efficacy of D-PC.

---

> > ### Author Response · Authors · 2025-11-20
> > **Response to Reviewer B2tA (contd)**
> >
> > **References**
> >
> > [1] Kai Shao, Jiacheng Shen, Chijie An, and Mathieu Laurière. Reinforcement learning for finite space mean-field type games. arXiv preprint arXiv:2409.18152, 2024a.
> >
> > [2] Julien Perolat, Sarah Perrin, Romuald Elie, Mathieu Laurière, Georgios Piliouras, Matthieu Geist, Karl Tuyls, and Olivier Pietquin. Scaling up mean field games with online mirror descent, 2021. URL https://arxiv.org/abs/2103.00623.
> >
> > [3] Ren´e Carmona, Mathieu Laurière, and Zongjun Tan. Model-free mean-field reinforcement learning: Mean-field MDP and mean-field Q-learning, 2021. URL https://arxiv.org/abs/1910.12802.
> >
> > [4] Yue Guan, Mohammad Afshari, and Panagiotis Tsiotras. Zero-sum games between mean-field teams: Reachability-based analysis under mean-field sharing. Proceedings of the AAAI Conference on Artificial Intelligence, 38(9):9731–9739, Mar. 2024a. doi: 10.1609/aaai.v38i9.28831.  URL https://ojs.aaai.org/index.php/AAAI/article/view/28831.
> >
> > [5] Kai Shao, Jiacheng Shen, Chijie An, and Mathieu Lauri`ere. Reinforcement learning for finite space mean-field type games. arXiv preprint arXiv:2409.18152, 2024a.

---

### Official Review · Reviewer_vvaB · 2025-11-03

**Soundness:** 3
**Presentation:** 3
**Contribution:** 1
**Rating:** 2
**Confidence:** 5

**Summary:**

This paper introduces MF-MAPPO, a mean-field extension of PPO tailored for large-scale competitive team games (ZS-MFTGs). The authors propose a shared actor-critic architecture with minimally informed critics and demonstrate scalability and robustness through finite-population training. They also present a gradient-regularized variant for partially observable settings, supported by a decentralized estimation algorithm (D-PC). The work is evaluated on custom benchmark environments designed to reflect collaborative-competitive dynamics.

**Strengths:**

* The paper is well-written and clearly structured, making complex ideas accessible.
* The theoretical framework is sound, with appropriate use of mean-field approximations and convergence guarantees.
* The integration of PPO with mean-field theory is executed cleanly, and the proposed architecture is computationally efficient.

**Weaknesses:**

* The paper lacks significantly novel theoretical contributions. Most theorems are either direct consequences of existing results or minor extensions.
* The proposed MF-MAPPO algorithm is a relatively straightforward adaptation of PPO to a mean-field setting, and the use of shared actor-critic networks is not conceptually new.
* The benchmark environments, while tailored, do not convincingly demonstrate capabilities beyond prior work in terms of emergent behavior or strategic complexity.
* The D-PC estimation method is a modest variation on existing consensus algorithms and does not introduce fundamentally new ideas.

**Questions:**

I appreciate the idea of integrating the idea of PPO into Mean Field RL. However, it appears that the algorithm presented in the paper merely applies a PPO - style (gradient - clipping) policy update mechanism to Mean Field Games (MFG), without delving into the underlying mechanism. In other words, similar to the theorem discovered by Trust Region Policy Optimization (TRPO) in the classical Reinforcement Learning (RL) domain, is it possible to identify the fundamental principle that reveals Mean Field Games (MFG) also possess a "basis" for justifying the application of a PPO - style algorithm?

---

> ### Author Response · Authors · 2025-11-20
> **Response to Reviewer vvaB**
>
> We thank the reviewer for their valuable feedback and their appreciation of our attempt to integrate PPO into the MF setting. It seems that there is some confusion about what our paper is trying to accomplish. We hope that our answers below will convince you that our paper deserves a better score.
>
> Firstly, we would like to mention that mean-field _**team**_ games (MFTGs) differ significantly from both non-cooperative mean-field games (MFGs) and cooperative mean-field control/team problems (MFCs), as MFTGs deal with mixed competitive-collaborative scenarios - forming a distinct class of problems on their own. Algorithms used for MFCs and MFGs cannot be used directly  for solving MFTGs with closed-loop (MF-dependent) policies.
>
> Existing work typically treats MFGs and MFCs as _single_ infinite-population systems, whereas MFTGs consist of **two opposing infinite-population systems**, generating competitive dynamics even in the infinite-population limit. Similarly, multi-population MFGs (MP-MFGs) have been studied in the past, but often restrict agent dynamics and policies to be independent of both other agents and other population distributions, a rather stringent assumption with limited applicability.
>
> Moreover, prior works in MFGs/MFCs/MFTGs [1–3] rely on infinite-population oracles to provide exact mean-fields, reward functions, or state distributions. This is inherently unrealistic: “sampling from an infinite population” is a theoretical construct. In practice, **only finite-population observations are available**, and—to our knowledge—there are no existing algorithms in this field of research that train directly in the finite-population domain that are scalable, _while retaining provable performance guarantees_ (as established in Theorem 1).
>
> **Weakness 1 (novel theoretical contributions):** We argue that computing finite-population policies in the MFTGs realm itself is a contribution. Our theoretical proofs (Theorems 2 and 3) showing that this finite-population identical policy is $\epsilon$-optimal for larger population sizes, and proving that the finite-population computed gradient converges to the infinite-population policy gradient are indeed novel contributions. Furthermore, extending the problem to a partially observable setting via gradient regularization along with convergence proofs, is a simple yet novel extension to the existing literature. We kindly request the reviewer to cite the specific prior results they had in mind for theoretical novelties so that we may compare our approach with them directly.
>
> **Weakness 2 (novelty/capability of MF-MAPPO algorithm):** Our design choices deliberately exploit the information structure inherent in MFTGs. As noted in common response #1, MF-MAPPO is not a naïve instance of parameter sharing or a centralized-critic variant. The **minimally informed critic** enables training without access to local observations—thereby using minimal information, while preserving performance. It further serves as an important feature for privacy-preserving settings. Moreover, the use of identical team policies is theoretically justified by the mean-field structure rather than an ad-hoc simplification.
> Tables 1–2 and Figures 2–5 empirically validate these claims and clearly distinguish MF-MAPPO from MAPPO and parameter-sharing baselines (also see common response #1).
>
> **Weakness 3 (demonstrating the capability of MF-MAPPO):** The benchmarking environments with properly constructed weak-interaction dynamics and reward are sparse. We are not aware of environments that follow a mean-field _team_ game structure. In lieu of this, we have also extended MF-Env to include a new environment (see common response #2), further demonstrating the generality and applicability of our framework. We emphasize that these environments entail complex interactions between agents, unlike prior works that merely assume agent dynamics and policies to be independent of both other agents and other population distributions [2]. In particular, Figure 4 (I–II) shows that identical policies can generate heterogeneous team behaviors under such complex dynamics.
> We kindly request the reviewer to cite the specific prior results they had in mind that demonstrate emergent behavior in mixed competitive-collaborative settings so that we can compare MF-MAPPO with them.
>
> **Weakness 4 (novelty in D-PC):**
> Although D-PC draws from classical control-theoretic ideas, applying it to MARL—and specifically to ZS-MFTGs—is non-trivial. Smoothness of the ensuing policies cannot be guaranteed without gradient regularization, and deriving asymptotic convergence results for mean-field estimation algorithms has not been done previously, to the best of our knowledge.

---

> > ### Author Response · Authors · 2025-11-20
> > **Response to Reviewer vvaB (contd)**
> >
> > **Answer to question**:
> > As mentioned earlier, our domain of analysis is not MFGs but ZS-MFTGs. The solution framework incorporating the minimally informed critic and shared team actor is a more fundamental contribution. The structure of MFTGs does not directly go into PPO but into minimally informed critic and shared team actor, both of which are generic architectural innovations that can interface with multiple learning backbones.
> > The reason we chose PPO is its resulting mixed strategy that results in heterogeneous behaviors as seen in Figure 4 (I-II). PPO has been a long-standing algorithm for constraining update sizes and has been utilized in multi-agent tasks effectively and successfully. Its use in MFC (e.g., [4]) further validates its suitability in mean-field contexts. We further establish the merits of MF-MAPPO through Theorems 2 and 3.
> >
> > In summary, the goal of our paper is not to establish a theoretical justification for PPO within the MFTG framework, but rather to demonstrate how mean-field structure enables scalable and information-efficient MARL architectures. Developing TRPO-style performance guarantees for mean-field team games is certainly an interesting and valuable direction, but we believe it lies beyond the scope of the present work.
> >
> > **References**
> >
> > [1] Kai Shao, Jiacheng Shen, Chijie An, and Mathieu Laurière. Reinforcement learning for finite space mean-field type games. arXiv preprint arXiv:2409.18152, 2024a.
> >
> > [2] Julien Perolat, Sarah Perrin, Romuald Elie, Mathieu Laurière, Georgios Piliouras, Matthieu Geist, Karl Tuyls, and Olivier Pietquin. Scaling up mean field games with online mirror descent, 2021. URL https://arxiv.org/abs/2103.00623.
> >
> > [3] Ren´e Carmona, Mathieu Lauri`ere, and Zongjun Tan. Model-free mean-field reinforcement learning: Mean-field MDP and mean-field Q-learning, 2021. URL https://arxiv.org/abs/1910.12802.
> >
> > [4] Kai Cui, Sascha Hauck, Christian Fabian, and Heinz Koeppl. Learning decentralized partially observable mean field control for artificial collective behavior, 2024. URL https://arxiv.org/abs/2307.06175.

---

### Official Review · Reviewer_MyrG · 2025-11-05

**Soundness:** 4
**Presentation:** 3
**Contribution:** 3
**Rating:** 6
**Confidence:** 3

**Summary:**

This paper introduces MF-MAPPO, a mean-field extension of PPO, to address the scalability issues of multi-agent reinforcement learning in large-scale mixed cooperative-competitive tasks. The main contributions include: 1. Proposing the base version of MF-MAPPO, a scalable shared actor-critic algorithm for large-scale zero-sum MFTGs. 2. Extending the MF-MAPPO by gradient-regularized training and a decentralized mean-field estimation framework D-PC for partially observable MFTGs. 3. Constructing novel MFTG benchmarking environments named MFEnv for scalability. 4. Demonstrating performance and efficiency of method above through numerical experiments.

**Strengths:**

S1. This paper aims to addresse mixed collaborative-competitive mean-field game, rather than purely non-cooperative or fully cooperative settings, with broad applicability to real-world domains.
S2. It innovatively proposes MF-MAPPO, a PPO-based algorithm with a minimally-informed critic and shared-team actor, enabling effective scaling to large populations of agents, while guaranteeing convergence.
S3. It extends the algorithm to partially observable settings through a gradient-regularized training, and is coupled with the decentralized mean-field estimation framework D-PC, further broadening the application scope.
S4. The author provides a large number of rigorous theoretical analyses, and conducts numerical experiments to enhance persuasiveness.
S5. The paper is well-structured, with clear organization and rigorous logic.

**Weaknesses:**

W1. The setup of experiments is simple, only contains a cRPS game and a grid-based battlefield game, it maybe need to be further extended to other benchmark to better demonstrate generality and robustness.
W2. The paper does not analyze the cost of computation and decision-making overhead in such large-scale agent scenarios.
W3. As mentioned in the paper's conclusion section, the method struggles to extend to large state or action dimensionality, which to some extent limits its application in realistic scenarios.

**Questions:**

Q1. The MFenv benchmark only contains two tasks and is developed by yourselves, how does the method perform in other benchmarks?
Q2. How about the cost when the number of agents is extremely large?
I think the paper and the method described in it are good enough, and it could be better if the questions above are solved.

---

> ### Author Response · Authors · 2025-11-20
> **Response to Reviewer MyrG**
>
> We thank the reviewer for their feedback. In particular, we are grateful to the reviewer for recognizing the novelty in the problem formulation itself, along with its “broad applicability in real-world domains” which are furthered by the introduction of the estimator.
>
> **W1 and Q1:** [see common response #2]
>
> **W2 and Q2:** The cost of computation in terms of sample efficiency and training times has been added in the revised version of the paper (Refer to Tables 1 and E.2 in the supplementary material). MF-MAPPO and the other baselines update every $T_{\text{rollout}}$ steps, unlike DDPG-MFTG, which updates per time step, resulting in larger training durations. Additionally, we would like to remark that the cost of computation of these policies for MF-MAPPO does not increase with the number of agents, as one can train it for a relatively small population size and deploy the policy on a much larger scale, with retained optimality as guaranteed by Theorem 3 – making it free of further computation cost.
> Figure 2(b) highlights this result in the cRPS framework. We also have Figure E.9 in the appendix that preserves the performance of the policy when trained on 100v100 and deployed on 1500v1500 and 1000v1000.
>
> **W3:** [see common response #4]

---

### Official Review · Reviewer_Hdbj · 2025-11-07

**Soundness:** 3
**Presentation:** 4
**Contribution:** 3
**Rating:** 8
**Confidence:** 2

**Summary:**

This paper introduces a scalable multi-agent reinforcement learning algorithm designed for large mixed collaborative–competitive team games. MF-MAPPO employs shared team actors and shared critics that depend only on team-level mean-field distributions, enabling efficient training without full agent-level observability. The framework provides theoretical guarantees of near-optimality, showing that identical team policies learned in finite populations approximate the infinite-population limit. To address partial observability, the authors propose a Dynamic Projected Consensus estimator that allows agents to infer opponent distributions with limited communication. Experiments trained directly on finite-population simulators demonstrate the algorithm’s effectiveness in rock-paper-scissors and battlefield environments.

**Strengths:**

- It demonstrates scalability in large population environments. The shared actor–critic framework that depends only on mean-field (team-level) distributions allows the algorithm to scale to thousands of agents.
- Its finite-population guarantees ensure that policies trained on small populations generalize effectively to larger teams without additional retraining.
- The authors do include a degradation comparison between fully observable and partially observable settings, showing only minor performance loss under limited communication (battlefield - Fig 4).

**Weaknesses:**

- As the authors note, despite the progress, the algorithm scalability is still limited as the number of state variables increases, and learning accurate mean-field distributions becomes computationally expensive, motivating the need for dimensionality reduction techniques.
- The empirical evaluation is limited to tasks that are discrete, low-dimensional, and noise-free, which is a common limitation. However, for MF-MAPPO this restriction is particularly significant because both the mean-field formulation and the Dynamic Projected Consensus estimator rely on assumptions that do not easily extend to continuous or noisy environments.

**Questions:**

- Despite it might be a bit out of scope, how sensitive is MF-MAPPO to deviations from the mean-field assumption?

---

> ### Author Response · Authors · 2025-11-20
> **Response to Reviewer Hdbj**
>
> We thank the reviewer for their positive review.
>
> **Weakness 1 (Extending to large state-action spaces):**  [See common response #4]
>
> **Weakness 2 (Performance under noisy/continuous environments):**
> In order to claim generality and robustness of both MF-MAPPO and D-PC, we perform the 8x8 battlefield task in the presence of noisy communication and show that the performance degradation is minimal (Figure 4, V). Continuous time formulations of the work have been studied in the past [1] but without learning based techniques to compute policies. Continuous time mean-field theory requires stochastic differential equations, which are incompatible with standard RL without heavy discretization. Furthermore, finite state-action space MFTGs inherit well-developed tools for analysis like MDP-type formulations. Future work can involve extending the current developed RL framework into continuous time/state-action space.
>
> **Answer to question (Sensitivity of MF-MAPPO to mean-field assumptions):**
> Consistent with mean-field literature, we assume a Lipschitz model (Assumption 1), ensuring rewards and dynamics vary smoothly with the mean field and remain tractable. Our minimally informed critic leverages this structure, enabling training independent of local observations; the shared team policy relies on the same assumption. Applying MF-MAPPO outside this setting would neglect essential local-interaction effects and could therefore degrade performance. Without such structure, the problem collapses into a standard large-population MARL setting, which is intractable due to the curse of dimensionality. Thus, additional assumptions are essential for tractability.
> Future directions can include such deviated settings where additional noise or perturbations affect weak or local interactions, and we can aim to derive bounds for cases in which such noise is ignored under standard MF analysis. Another promising direction is to incorporate graphon mean-field models [2–3] to move beyond homogeneous weak interaction. Extending graphon-based interactions to the MFTG framework presents an intriguing avenue for further exploration.
>
> **References**
>
> [1] Sina Sanjari, Naci Saldi, and Serdar Yüksel. Nash equilibria for exchangeable team against team games, their mean field limit, and the role of common randomness. arXiv preprint arXiv:2210.07339, 2022.
>
> [2] Caines, Peter E., and Minyi Huang. "Graphon mean field games and their equations." SIAM Journal on Control and Optimization 59.6 (2021): 4373-4399.
>
> [3] Fabian, Christian, Kai Cui, and Heinz Koeppl. "Learning sparse graphon mean field games." International Conference on Artificial Intelligence and Statistics. PMLR, 2023.

---

### Author Response · Authors · 2025-11-20
**Common Response**

We would like to thank the reviewers for their constructive feedback and positive assessment of our work. Their suggestions have helped us improve and build upon the quality of our paper. We thank the reviewers for recognizing the merit and novelty in applying a PPO-style algorithm to a new class of problems: Mean-field team games, which were seen as “timely”, entailing “broad applicability” and executed “cleanly”. We appreciate the reviewers’ positive feedback on the paper’s soundness, clarity, and organization, and are glad our presentation “made complex ideas accessible”. We have addressed all identified weaknesses and questions as thoroughly as possible.

**1. Absence of baseline algorithms**

In lieu of the feedback, we have conducted more extensive and comprehensive benchmarking of MF-MAPPO against standard PPO variants, along with DDPG-MFTG (as done earlier). This evaluation highlights that MF-MAPPO is not merely a naïve combination of parameter sharing and MAPPO, but instead incorporates key architectural innovations by exploiting the mean-field information sharing structure—most notably the “minimally informed critic,” which does not require any local information (unlike centralized critics or regular parameter sharing approaches; see discussion in Section 3). This yields substantial dimensionality reduction, faster updates, and improved sample efficiency, as evidenced in Table 1 and Table E.2 (Appendix E) of the revised paper.
We evaluated all baselines on both cRPS and the 4×4 battlefield environment. In cRPS, MF-MAPPO achieves returns closest to the Nash value and converges to the center fastest (Table 1). In the battlefield scenario, MF-MAPPO outperforms all baselines as both attacker and defender (Table 2), and requires significantly fewer interactions to converge, consistent with its sample-efficiency advantages (Table E.2).

**2. Absence of benchmark environments**

Mean-field team game environments with weak-interaction dynamics are scarce, and we are not aware of existing benchmarks that follow the MFTG structure. Therefore, we constructed our own environments. To further demonstrate generality, we added a mean-field epidemiology game and trained MF-MAPPO within it (Section 5). The supplementary material also includes a single-team navigation task, illustrating that MF-MAPPO—while grounded in zero-sum theory—extends naturally to general MFTGs. For thematic coherence, however, the main paper focuses on competitive settings.

**3. Performance of the estimator under bandwidth limitations and noise**

Regarding the reviewers’ concerns on D-PC’s performance under bandwidth limitations, a comparison of D-PC under limited communication rounds has already been carried out in Section 5, Figure 4: III and IV, where we see that the % error cumulative rewards (metric defined in Eq. 10) when compared against the fully-observable scenario is minimal even under a few communication rounds. As the bandwidth increases, we see this error decrease, in line with Theorem 4 and intuition.
Following the reviewers’ suggestion, we added results for noisy estimates (Figure 4: V) on the 8×8 battlefield. Injecting zero-mean Gaussian noise of increasing variance into the estimates yields <5% error even at high noise levels, thanks to the projection step that keeps estimates on the simplex.

**4. Extending MF-MAPPO to large state-action spaces**

As we stated in the conclusion of our original submission, applying MF-MAPPO under large state-action dimensionality is a challenge, which aligns with the reviewers' observation. In MARL, large state-action spaces and large population sizes represent two distinct axes of difficulty. Our work primarily addresses the latter by developing a scalable foundation for mixed competitive–collaborative large-population games. Integrating MF-MAPPO with principled dimensionality-reduction techniques is an important direction for future work.

---

### Author Response · Authors · 2025-12-02
**Summary of Reviews, Author Rebuttal, and Paper (Revisions) for the Area Chair**

The zero-sum mean-field team game (ZS-MFTG) framework models competition between large cooperating teams, but existing methods struggle to compute scalable policies. Prior work typically relies on infinite-population oracles that provide exact mean-field trajectories  [1-3], whereas in practice **only stochastic finite-population samples are observable**. Our method is explicitly designed for this realistic setting, and we theoretically show that policy gradients estimated from finite-population rollouts converge to their infinite-population limits. To improve efficiency, we introduce a **minimally informed critic** that leverages the mean-field information structure without requiring access to all agents’ observations, leading to significantly faster and more stable learning. Finally, unlike classical MFTG approaches that assume global knowledge of team-level distributions, our dynamic projected-consensus (D-PC) mechanism enables agents to form accurate mean-field estimates without centralized coordination. Our design choices deliberately exploit the information structure inherent in MFTGs. Together, these components relax restrictive assumptions and bridge the gap between theoretical mean-field models and practical large-scale multi-agent systems.

All reviewers recognized the merit and novelty of applying a PPO-style algorithm to large-population competitive team games, describing our framework as a “tidy fit,” “cleanly executed,” and addressing a “timely” problem with “broad applicability.” They also appreciated the clarity and organization of the paper in making “complex ideas accessible”. Their comments have significantly strengthened our work.

In response to the constructive feedback, we conducted more comprehensive benchmarking of MF-MAPPO against standard PPO variants—independent PPO (MF-IPPO), MAPPO with parameter sharing (MAPPO-PS) and MAPPO with a centralized critic (MAPPO-CC) [5]—and DDPG-MFTG [1] on both the constrained Rock–Paper–Scissors (cRPS) and battlefield environments. These experiments further underscore the advantages of our network design—particularly the _minimally informed critic_, which relies solely on local information rather than aggregating observations from all other agents. This avoids overwhelming the networks, achieves substantial dimensionality reduction, and results in faster updates and improved sample efficiency, directly addressing the reviewers’ concerns (see comparisons in Tables 1 and 2 of the main text and Table E.2 in the appendix). In cRPS, MF-MAPPO achieved performance closest to the theoretical equilibrium value and converged the fastest. In the battlefield setting, it outperformed all baselines as both attacker and defender, while requiring far fewer interactions with the environments. Importantly, the computational cost of MF-MAPPO does not grow with the number of agents, and an MF-MAPPO policy trained on small populations can be transferred to a much larger population team, with optimality preserved as guaranteed by our theory.

To the best of our knowledge no existing benchmarks directly target MFTGs. Thus, we adopted the epidemiology (SIS-model [6]) game to the MFTG framework and performed additional numerical experiments to address the reviewers’ concerns about generality. The results demonstrated that MF-MAPPO performs effectively in this domain as well. Our environments all involve nontrivial agent interactions—unlike prior work assuming independence of agent dynamics and population distributions [2]—and we are not aware of existing benchmarks tailored to mean-field team games.

To broaden the applicability of our algorithm, we introduced a framework wherein a team can estimate its opponent’s mean-field. In addition to the results of the estimator’s performance under bandwidth limitations (measured against the fully-observable scenario), following the reviewers’ suggestion, we also added results for performance under noisy estimates on the 8×8 battlefield.  Even with high-variance zero-mean Gaussian noise, estimation error remained under 5%, demonstrating the robustness of our proposed estimator.

---

> ### Author Response · Authors · 2025-12-02
> **Summary of Reviews, Author Rebuttal, and Paper (Revisions) for the Area Chair (contd)**
>
> Reviewer vvaB appeared to conflate mean-field team games with mean-field games (MFGs). We clarify that MFTGs differ fundamentally from both non-cooperative MFGs and cooperative mean-field control (MFC): MFTGs involve mixed competitive–collaborative interactions between **two opposing infinite-population systems**, forming a distinct class of problems. Existing MFG/MFC algorithms cannot be directly applied because MFTGs require closed-loop mean-field-dependent policies to react to opponents’ behaviors. Although multi-population MFGs have been studied, they typically assume independence of dynamics and policies across populations, which is a restrictive assumption that we do not have in our formulations.
>
> Reviewer vvaB, although appreciative of the use of a PPO-style algorithm, raised questions about the theoretical justification for PPO. However, our primary contribution is the architectural framework itself—particularly the minimally informed critic and team-shared actor—which can interface with many learning backbones. We selected PPO due to its stability, its natural production of mixed strategies, and its strong track record in multi-agent tasks [5]. Moreover, many of the baseline algorithms against which we compare rely on PPO, making it a natural and fair choice. However, PPO is not the only choice. Our aim is not to justify PPO theoretically for MFTGs, but to demonstrate how mean-field structure enables scalable, information-efficient MARL architectures.
>
> We reiterate, consistent with the reviewers’ observations, that applying MF-MAPPO in environments with very large state–action dimensionality remains challenging. In MARL, population size and state–action dimensionality pose distinct difficulties, and our work focuses on the former by providing a scalable foundation for mixed competitive–collaborative large-population games. An important direction for future research is combining MF-MAPPO with principled dimensionality-reduction techniques. We also note that our finite state–action formulation is deliberate: discrete MFTGs naturally align with MDP-style analysis, whereas continuous-time mean-field models rely on stochastic differential equations that are not directly compatible with standard RL without significant discretization. Extending our framework to continuous time or continuous spaces—and exploring settings with perturbations or additional noise in local interactions, as suggested by Reviewer Hdbj—are compelling avenues for future work.
>
> **References**
>
> [1] Kai Shao, Jiacheng Shen, Chijie An, and Mathieu Laurière. Reinforcement learning for finite space mean-field type games. arXiv preprint arXiv:2409.18152, 2024a.
>
> [2] Julien Perolat, Sarah Perrin, Romuald Elie, Mathieu Laurière, Georgios Piliouras, Matthieu Geist, Karl Tuyls, and Olivier Pietquin. Scaling up mean field games with online mirror descent, 2021. URL https://arxiv.org/abs/2103.00623.
>
> [3] René Carmona, Mathieu Laurière, and Zongjun Tan. Model-free mean-field reinforcement learning: Mean-field MDP and mean-field Q-learning, 2021. URL https://arxiv.org/abs/1910.12802.
>
> [4] Yue Guan, Mohammad Afshari, and Panagiotis Tsiotras. Zero-sum games between mean-field teams: Reachability-based analysis under mean-field sharing. Proceedings of the AAAI Conference on Artificial Intelligence, 38(9):9731–9739, Mar. 2024a. doi: 10.1609/aaai.v38i9.28831.  URL https://ojs.aaai.org/index.php/AAAI/article/view/28831.
>
> [5] Yu, Chao, et al. "The surprising effectiveness of ppo in cooperative multi-agent games." Advances in neural information processing systems 35 (2022): 24611-24624.
>
> [6] Eich, Yannick, et al. "Bounded Rationality Equilibrium Learning in Mean Field Games." Proceedings of the AAAI Conference on Artificial Intelligence. Vol. 39. No. 13. 2025.

---

### Meta-Review · Area_Chair_j4vm · 2025-12-21

**Summary:**

The paper proposes new mean-field RL techniques for mixed-motive multi-agent systems with partial observability.  The reviewers expressed the following concerns:

1. Limited scalability with respect to the number os state variables
2. The empirical evaluation is limited to two tasks that are discrete, low-dimensional, and noise-free
3. No analysis of computational cost and decision-making overhead in large-scale agent scenarios
4. The paper lacks significantly novel theoretical contributions.
5. The proposed MF-MAPPO algorithm is a relatively straightforward adaptation of PPO to a mean-field setting
6. The D-PC estimation method is a modest variation on existing consensus algorithms and does not introduce fundamentally new ideas.
7. The set of baselines is limited

**Reviewer Concerns:**

Concern 1 is not a deal breaker and in fact was acknowledged as a limitation by the paper.  The authors added new experiments that address concerns 2 and 3.  The rebuttal addressed concerns 4, 5 and 6 in a satisfactory manner.  The theoretical contributions and the proposed algorithms are novel and non-trivial.  The revised paper partly addressed concern 7 by adding some baselines.  Related to concern 7 and in addition to the reviewers' concerns, let me point out three papers that should have been cited and compared to.

Subramanian, S. G., Taylor, M. E., Crowley, M., & Poupart, P. (2022, June). Decentralized mean field games. In Proceedings of the AAAI Conference on Artificial Intelligence (Vol. 36, No. 9, pp. 9439-9447).

Ganapathi Subramanian, S., Taylor, M. E., Crowley, M., & Poupart, P. (2021, May). Partially Observable Mean Field Reinforcement Learning. In Proceedings of the 20th International Conference on Autonomous Agents and MultiAgent Systems (pp. 537-545).

Ganapathi Subramanian, S., Poupart, P., Taylor, M. E., & Hegde, N. (2020, May). Multi Type Mean Field Reinforcement Learning. In Proceedings of the 19th International Conference on Autonomous Agents and MultiAgent Systems (pp. 411-419).

The above papers describe alternative RL techniques for mean field games with partial observability (i.e., agents can only observe the actions of a finite number of other agents), multiple teams (modelled as agents with different types) and decentralized mean field estimation.

**Reviewer Scores:**

I expect reviewers Hdbj and MyrG to maintain their scores since they recommended acceptance and their concerns were addressed.

I expect reviewers vvaB and B2tA to maintain their scores and recommend rejection since the rebuttal did not fully address their concerns.  The theoretical contributions and the proposed algorithms are novel, but their significance is not clear due to missing related work and baselines that can be found in the three papers listed above.  Since this missing work provides alternative solutions to the problems tackled by this paper, the work is not ready for publication.

---

### Decision · Program_Chairs · 2026-01-26

Reject